# Ensuring Fair Comparisons in Time Series Forecasting: Addressing Quality Issues in Three Benchmark Datasets

## Abstract

Time series forecasting (TSF) is critical in numerous applications; however, unlike other AI domains where benchmark datasets are meticulously standardized, TSF datasets often suffer from data inconsistencies, missing values, and improper temporal splits. These issues have an impact on model performance and evaluation. This paper addresses these challenges by proposing inconsistency-free versions of three well-known TSF datasets. Our methodology involves identifying and correcting data inconsistencies using a combination of linear interpolation and context-aware imputation strategies. Additionally, we introduce a novel cycle-inclusive data splitting method, which respects the longest cycle in each dataset, ensuring that models are evaluated over meaningful temporal patterns. Through extensive testing of multiple transformer-based models, we demonstrate that our revised datasets and cycle-inclusive splitting lead to more accurate and interpretable forecasting results, as well as fairer comparison of TSF models. Finally, our findings highlight the need for proper dataset refinement and tailored data splitting strategies in TSF tasks, and pave the way for future work in the development of more robust forecasting benchmarks.

## 1 Introduction

Time series forecasting (TSF) is essential for numerous applications across diverse domains, including weather prediction, energy management, and traffic forecasting (Lim & Zohren, 2021). Benchmark datasets play a crucial role for evaluating machine learning (ML) models on specific tasks. Their accuracy heavily depends on the task(s) for which the dataset was designed (Koch et al., 2021) and its quality over both training and evaluation sets. However, time series data, typically collected through monitoring devices or by human input, are prone to errors such as missing values and faulty sensor readings. These issues can significantly distort model performance evaluations, leading to unreliable comparisons across different models.

In other ML fields, significant efforts have been put into designing standardized and cleaned datasets, exempt of biases (Dixon et al., 2018). These datasets provide a controlled environment where differences in model performance are solely attributed to the models themselves, rather than inconsistencies in the data. This practice has been crucial for advancing the field, enabling researchers to make fair comparisons and draw meaningful conclusions (Sim et al., 2003). Unfortunately, the TSF field lacks such standardized datasets, which leads to fragmented and often misleading evaluations of forecasting models. Furthermore, as Liu & Wang (2024) emphasize, the distribution shift between training and testing sets presents additional challenges in TSF.

We hypothesize that there is an urgent need for cleaner and standardized TSF datasets to ensure fair benchmarking of models. Errors inherent in current TSF datasets—-such as incorrect temperature conversions, arbitrary imputation of missing data, and improper temporal splits—-obscure the true capabilities of forecasting models. These inconsistencies not only hinder accurate performance assessments but also stall progress in the field. Our research confirms that such errors impact model evaluation, particularly when comparing results (1) before and after data corrections, and (2) without or with data splitting strategy revision.

Our study focuses on three widely-used multivariate time series forecasting (MTSF) datasets that may pose challenges for researchers attempting to compare literature with them: the weather datasets from the Informer and Autoformer papers, which, despite being different, share the same name, and the UCI electricity load diagrams (ELD) dataset, which has multiple variants with differing features and time coverage. The existence of these dataset variants, coupled with each dataset specific inconsistencies, complicates accurate comparison across models. To address these issues, we conduct an extensive analysis of these datasets, identify the underlying issues and propose cleaned versions using techniques such as linear interpolation and context-aware imputation. Additionally, we introduce a novel cycle-inclusive data splitting strategy, which ensures that models are evaluated over entire temporal cycles, rather than arbitrary time slices, offering a more reliable basis for comparison.

By providing cleaner versions of these datasets and introducing standardized approaches to temporal splitting, we lay the groundwork for more accurate and fair evaluations of TSF models. These refined datasets not only serve as a reliable baseline for benchmarking but also allow researchers to explore model robustness through more controlled and progressive evaluations. Our contributions therefore address a critical gap in TSF research and offer a path toward more reliable and meaningful advancements in the field. Our contributions are the following:

- Comprehensive and thorough analysis of three well-known TSF datasets and their variants, if any, providing researchers with detailed descriptions to better understand the data and develop novel architectures.

- Identification of dataset inconsistencies, allowing researchers to evaluate model behavior in the presence of errors in input or ground-truth data.

- Creation of new, cleaned versions of these datasets, enabling researchers to assess model performance in the absence of errors and then evaluate robustness under controlled noisy conditions.

- Comparison of model performance on corrected datasets using both transformer- and linear-based models.

- Introduction of cycle-inclusive data splitting, ensuring performance evaluation over the longest temporal cycle of the data.

- Comparison of model performance on cycle-inclusive splits offering insights into how different data splitting approaches affect results.

## 2 BACKGROUND AND RELATED WORKS

### 2.1 IMPORTANCE OF DATASET IN BENCHMARKING TASK

Standardized datasets offer several key advantages: (i) **Controlled Environment**: Cleaned datasets enable researchers to assess models in a controlled configuration where the effects of preprocessing are minimized. This is crucial for understanding the inherent strengths and weaknesses of different models. It allows differences in performance to be attributed to the models themselves rather than data inconsistencies, ensuring consistency across studies and enabling fair comparisons between different models and approaches (Sun et al., 2022). (ii) **Progressive Evaluation**: Starting with clean data establishes a reliable baseline for model performance. Researchers can then introduce controlled errors or inconsistencies in a structured manner to evaluate model robustness comprehensively. (iii) **Real-World Applicability**: While real-world data are rarely perfect, insights gained from benchmarking on cleaned datasets can guide the development of more robust models. Moreover, techniques used to clean these datasets can be applied to real-world data, improving their quality and facilitating model performance. (iv) **Community Standards**: The practice of using standardized datasets promotes reproducibility, transparency, and collaboration within the ML community, advancing the field as a whole (Sim et al., 2003).

For example, in computer vision (CV), benchmark datasets are often curated to ensure consistency: samples are free of corrupted or missing pixels, and all images are resized to a fixed dimension (Xiao et al., 2017). Although real-world data may contain imperfections, such as missing pixels or varying image sizes, these cleaned datasets serve an important purpose: establishing a baseline allowing researchers to isolate the effects of model design without interference from data inconsistencies.

Once this baseline is established, controlled errors can be introduced to assess model robustness to missing data, as shown in studies like (Collier et al., 2020).

Standardized datasets provide a clear foundation for understanding model behavior, comparing results across studies, and ensuring the robustness of models in real-world applications. While they are critical for benchmarking, the focus of research papers remains on model performance, neglecting potential issues within the datasets themselves. Such issues can significantly influence outcomes, either positively or negatively (Sambasivan et al., 2021). TSF is no exception. Although initiatives like GluonTS (Alexandrov et al., 2020) or TFB (Qiu et al., 2024) aim to promote fairer benchmarking by standardizing experimental settings and pipelines, their effectiveness can be compromised if the underlying datasets contain errors or inconsistencies.

## 2.2 Multivariate Time Series Datasets

MTSF has gained significant attention following the introduction of attention mechanisms (Vaswani et al., 2017) and their successful applications in both CV and natural language processing (NLP). Consequently, in this study, we focus on the most cited papers addressing MTSF from 2019 to 2023 to identify the most commonly used datasets in the domain.

LogTrans (Li et al., 2019) was among the first model to leverage the Transformer principle to address the computational cost of capturing long-term dependencies in MTSF. However, the model that arguably had the greatest impact on the community was Informer (Zhou et al., 2021), which introduced the *probspare* attention mechanism, significantly advancing MTSF. The number of publications addressing MTSF has then increased exponentially, leading to several variants: time series decomposition (Autoformer (Wu et al., 2021)), usage of frequency domain (FEDformer (Zhou et al., 2022)), reversible instance normalization (RevIn (Kim et al., 2022)), cross-dimension dependency (Crossformer (Zhang & Yan, 2023)), patch attention and channel-independent structure (PatchTST (Nie et al., 2023)), and recently, inverse architecture (iTransformer Liu et al. (2024)).

The supremacy of Transformer-based model was questioned in 2022 when Linear-based models demonstrated comparable, and sometimes superior, performance. Zeng et al. (2023) introduced both normalized (NLinear) and time series decomposition (DLinear) versions. Following this, Li et al. (2024) proposed reversible models (RLinear, RMLP, etc.).

All these models were benchmarked on well-known and publicly available datasets, which are not only used in MTSF tasks but in numerous other studies such as (Liu et al., 2022) and (Wu et al., 2023). Table 1 in the supplementary material lists the datasets found in these cited papers. However, despite their widespread use, to the best of our knowledge, these datasets have not been analyzed in depth or described in terms of the different variations that might exist. Detailed and fine-grained analysis of these datasets should enable researchers to better understand their characteristics and design models that are more tailored to the data, ultimately leading to improved MTSF performance.

## 2.3 Existing Practices

### 2.3.1 Pre-processing

In the reviewed literature, datasets are standardized using $StandardScaler()$, which removes the mean and scales each feature to unit variance based on the training set. While widely adopted, this practice can lead to problems when datasets include values that significantly deviate from the rest of the data–particularly if these outliers are unintended anomalies. Such deviations can distort the standardization process, resulting in skewed data representation that may negatively affect model performance, especially the process is reversed. Although recent efforts (Alexandrov et al., 2020; Qiu et al., 2024) have provided TSF datasets from various domains, they do not address the inconsistencies inherent in these datasets.

Another issue arises when datasets contain excessive missing time steps that are filled with a fixed value **within the range** of real data. For instance, Alexandrov et al. (2020) in GluonTS describe a method for identifying missing time steps, and suggest filling them with a "desired value". However, this practice can distort the natural distribution of the dataset, further impacting model accuracy during training and evaluation.

In addition, if there is a substantial distribution shift between the training, validation, and evaluation sets, standardization can exacerbate these discrepancies, leading to unreliable model performance.

### 2.3.2 DATA SPLITTING

In most studies, datasets are commonly split chronologically, either using a fixed ratio (e.g., *7:1:2* for training, validation, and evaluation) (Wu et al., 2021) or based on temporal periods (e.g., *15/3/4* months) (Zhou et al., 2021). However, such splitting methods can lead to suboptimal results, particularly when datasets exhibit cyclic patterns. Training models on a subset that does not fully represent the entire temporal cycle can compromise their ability to generalize effectively across the full cycle (Botache et al., 2023).

Such non-cycle-based splitting approaches may result in models being evaluated on different parts of the cycle, resulting in unfair comparisons and reducing their relevance for real-world forecasting tasks, where capturing the entire temporal cycle is essential for long-term accuracy. While frameworks like GluonTS (Alexandrov et al., 2020) offer flexible options for defining splitting strategies, the impact of these methods on model performance remains underexplored in the literature.

## 3 PROBLEM DEFINITION

### 3.1 INCONSISTENCIES IN TIME SERIES FORECASTING DATASETS

Time series data are prone to errors due to various factors such as device failures, recording issues, or human errors, resulting in inconsistencies like missing values, irregular observations, or data gaps. While some research has developed models that are robust to such inconsistencies (Chen et al., 2024; Niu et al., 2023; Barazani & Tolpin, 2022), these approaches typically rely on datasets where errors are clearly identified, allowing for accurate assessments of model performance under impaired conditions. Similarly, in research focused on imputing missing data, a ground truth dataset free of errors or inconsistencies is essential for benchmarking the effectiveness of imputation methods.

In TSF, models are not just tasked with predicting future values–they also need to capture complex patterns such as auto-correlation, trends, and seasonality. As such, datasets with unidentified inconsistencies cannot be used as-is for fair and accurate model evaluations. This is especially true for MTSF, where models must analyze multiple channels and determine interdependencies between them, adding to the complexity of the forecasting task. As a consequence, addressing these inconsistencies is crucial to ensure that models are evaluated on datasets that accurately represent the underlying phenomena. Proper pre-processing is essential for evaluating models fairly and preventing distortions that can mislead conclusions about their performance.

### 3.2 DATA SPLITTING ISSUES IN TIME SERIES FORECASTING

Time series data often exhibit periodic fluctuations driven by natural cycles–such as seasonal changes due to the Earth's revolution–or human-made cycles like weekly and monthly patterns. These cyclical variations are fundamental to the data and must be appropriately accounted for in the design of forecasting models and the splitting of datasets.

For artificial intelligence (AI) models that follow the typical training-validation-evaluation paradigm, model weights are updated using the training data while the score on the validation set guides hyperparameter optimization. For data exhibiting cyclical patterns, however, if the validation set does not represent all cycles present in the dataset, the model risks being over-optimized for the specific period covered by the validation set. Similarly, if the evaluation set does not encompass all cycles, the reported performance metrics will only reflect the model's capabilities for the considered period, potentially resulting in misleading conclusions about its overall performance. Compounding this issue, MTSF studies often use ratio-based chronological splits. This approach exacerbates the risk of misrepresenting model performance by disregarding the cyclical nature of the data.

As a result, cycle-inclusive data-splitting method, ensuring that each split captures the longest present cycle, is needed to ensure that models are evaluated on data that reflects the full range of temporal patterns, improving the reliability and generalizability of performance metrics.

## 4 PROPOSAL FOR FAIRER EVALUATION

### 4.1 INCONSISTENCIES DETECTION

Given the diverse nature of datasets used in MTSF–spanning domains such as electricity consumption, traffic flow, finance, and health–detecting inconsistencies is a complex and domain-specific challenge. A universal approach to identifying errors across all datasets requires careful design, as certain values may carry different meanings depending on the context. For example, the value "0" is frequently used to replace missing data, but its interpretation varies significantly between datasets. In weather datasets, a humidity value of 0 is physically impossible on Earth and therefore could likely indicate an error. However, in solar energy datasets, a value of 0 indicates a lack of energy generation, typically occurring at night or during severe weather conditions. Such a value should not immediately be flagged as an error without further investigation. For instance, encountering a zero value during a period of otherwise high energy generation could indicate a potential error, as sudden changes in weather conditions are uncommon. In such cases, cross-referencing the observation with weather data for that specific time and location, or consulting other external sources, is necessary to validate the observation's validity.

To address these challenges, we adopted a combination of domain knowledge and visual inspection to detect inconsistencies. First, we plotted the time series data to visually examine any anomalies, irregular patterns, or outliers. Additionally, where domain-specific knowledge is readily available, we leveraged it to cross-check known relationships and ensure the recorded data adhered to expected patterns. While this approach relies heavily on expert knowledge and is resource-intensive, it effectively demonstrates the existence of inconsistencies in these datasets. The results underscore the need for more sophisticated, and potentially automated, methods in future analyses to ensure that datasets used in MTSF are accurate and reliable for model training and evaluation.

### 4.2 DATA CORRECTION AND IMPUTATION

To correct the inconsistencies identified in the datasets, we implemented a systematic approach. First, for each type of inconsistency discovered, we added a new column to the dataset to mark time steps with such inconsistencies, before replacing the corresponding values with "NaN". We then categorized the inconsistencies into two scenarios: (i) in isolated time steps (i.e., preceding and following time steps are consistent) and (ii) in consecutive time steps. For each scenario, we observed two cases: (a) inconsistencies affecting a subset of the dataset channels and (b) inconsistencies affecting all channels (e.g., missing time steps). In the first scenario (i), we employed *linear interpolation* to fill in the gaps, this solution is suitable for both cases (a) and (b).

In the second scenario (ii), we treated case (a) by first using *context-aware imputation*. For that, we created a new dataset consisting of all time steps without marked inconsistencies, referred to as the "correct data". For each time step with inconsistencies, we filtered the "correct data" to include only entries that shared the same time, month and values contained in the channels without inconsistencies. We then replaced the inconsistent values of the target channels with the average of these filtered values. If no similar data points were available, we resorted to *linear interpolation* to fill in the remaining inconsistencies. For case (b) of the second scenario (ii), we again used *linear interpolation* to avoid introducing out-of-context information. Although these correction strategies may not be optimal, they provide a foundational approach to **assess the impact of a corrected version** of the datasets. Future research can build upon this work by proposing more sophisticated methods for data correction.

### 4.3 CYCLE-INCLUSIVE DATA SPLITTING

To ensure unbiased model evaluation, the length of the sets after splitting must be at least equal to the cycle length most represented in the dataset, considering all channels. We define the overall longest cycle (OLC) as a cycle that is shared by a substantial number of the channels under consideration. Specifically, if the longest cycle is only observed in one or fewer than ten channels, we do not classify it as the OLC.

To determine the OLC, we propose focusing on the frequency analysis of the dataset. By applying the fast Fourier transform (FFT), we can transpose the data into the frequency domain. In this

representation, we identify the fundamental frequency for each channel and subsequently determine the corresponding fundamental period. By combining this frequency analysis outputs with domain knowledge, as well as distribution and correlation analyses, when available, we can define the OLC and establish more appropriate lengths for the validation and evaluation sets.

This cycle-inclusive splitting method ensures that models are tested over the most representative and longest cycle present in the data.

## 5 CASE STUDY: DATASET INCONSISTENCIES

This study examines three well-known real-world datasets critical appropriate comparison of MTSF models. Due to space constraints, comprehensive analyses of these datasets are included in the supplementary material. Additionally, we provide the revised versions of these datasets, along with the corresponding code for their generation and a Python data loader for implementing cycle-inclusive splitting, are available in the following Git repository: `https://anonymous.4open.science/r/2392-NDBT-2AED/`.

### 5.1 WEATHER FROM INFORMER

This dataset is a subset of the local climatological data (LCD) dataset, originally comprising weather observations across 20 indicators from various global stations. Specifically, the version referenced by (Zhou et al., 2021) focuses on 12 weather indicators from a single station in the U.S. spanning over a four-year period. Unfortunately, the authors did not disclose the name or ID of this weather station, complicating the assessment of data processing methods applied to generate this dataset.

**Inconsistencies:** Our detailed analysis (see supplementary material) identified several critical inconsistencies within this dataset. For instance, indicators provided in both *Celsius* and *Fahrenheit* should exhibit a linear relationship, yet discrepancies were noted in the wet bulb temperature data. Specifically, for a Fahrenheit value of $32°F$, the corresponding Celsius values ranged from $-9.5°C$ to $9.9°C$, which is inconsistent. Moreover, our plots of Celsius against Fahrenheit for all temperature indicators revealed instances where both values were equal to zero, suggesting that missing data had been incorrectly replaced with zeros. These errors, present in both the validation and evaluation sets, can hinder the learning process and accurate evaluation of models.

**Revised versions:** To create a more appropriate version for MTSF, we addressed the identified errors. Conversion discrepancies were corrected using the formula $T_C = (T_F - 32) * 5/9$, where $T_C$ and $T_F$ denote temperatures in Celsius and Fahrenheit, respectively. Any time step where both Celsius and Fahrenheit values were zero was considered corrupted. For these instances, any other channels for which the observation was also equal to zero were treated as invalid. The replacement process followed the methodology outlined in Section 4. Our first corrected revision, referred to as "**LCDWf_1H_4Y_USUNK**", is the same dataset with inconsistencies corrected. However, we noticed that *WetBulbCelsius*, the target channel for univariate prediction, is the only temperature indicator in this dataset represented as a "float"; the others are integers. Consequently, we propose a second version, "**LCDWi_1H_4Y_USUNK**", where the target channel data are rounded to integers.

**Overall longest cycle:** Our frequency analysis (detailed in the supplementary material) reveals that, aside from *Relative Humidity*, *Wind Speed* and *Direction*– which exhibit a fundamental period of 24 hours–most channels display a fundamental frequency corresponding to 365.25 days. Furthermore, our data distribution study indicates that the yearly cycle significantly influences the value ranges across channels. Specifically, channels with a fundamental period of 365.25 days demonstrate substantial seasonal fluctuations in data distribution. Additionally, examining the data distribution per set, based on ratio splitting, further highlights significant distribution shifts. On the contrary, **our cycle-inclusive split significantly mitigate this shift between sets**.

These findings are supported by our correlation analysis, which illustrates that correlations between channels vary markedly by season, particularly between *Winter* and *Summer*. Collectively, these analyses affirm that the minimum length for validation and evaluation sets should be of one year to ensure robust model evaluation.

**Experiment setting:** We conducted performance tests on both our corrected versions and the original Informer dataset. These tests also used two distinct splitting strategies: the conventional chrono-

logical ratio splitting of *7:1:2* (commonly found in the literature), as well as a chronological cycle-inclusive splitting of *2/1/1* year, which we consider more appropriate for MTSF tasks.

## 5.2 WEATHER FROM AUTOFORMER

This dataset is a subset of Max-Planck-Institute (MPI) dataset, originally providing weather observations of 21 indicators from three weather stations in Germany. The Autoformer version focuses on data collected from the roof of the Max-Planck Biogeochemistry Institute during the year 2020.

**Inconsistencies:** Our detailed analysis revealed several inconsistencies within this dataset. We identified missing time steps and duplicated rows, alongside unaddressed errors. Notably, *Wind velocity* and $CO_2$ *concentration* exhibited "$-9999$" values, indicating sensor failures as confirmed by the authors. These errors, present in the training set, compromise pre-processing and hinder the model's learning capabilities. Moreover, the dataset's one-year duration is insufficient for capturing the full range of seasonal patterns and inter-channel relationships. As a result, this dataset is better suited for testing models' abilities to understand fine-grained relationships between indicators rather than long-cycle dependencies.

**Revised versions:** To create a more suitable version for MTSF, we rectified the identified inconsistencies following the procedures outlined in Section 4. Our corrected version, named "**MPIW_10T_1Y_R**", is then the same dataset with inconsistencies corrected. To facilitate performance testing with a cycle-inclusive split, we extended this dataset to four years, covering the period from "2020-01-01 00:10:00" to "2024-01-01 00:00:00" (included). We employed the same methods to correct sensor failures and fill in missing time steps. This extended dataset is available in two versions: "**MPIW_10T_4Y_R**" with a 10-minute resolution and "**MPIW_1H_4Y_R**" for hourly observations (further details on the process are provided in the supplementary material).

**Overall longest cycle:** Our frequency analysis (refer to the supplementary material for further details) indicates that, with the exception of *Surface Shortwave Downward Radiation*, *Photosynthetic Active Radiation (PAR)* and *max. PAR*–which have a fundamental period of 6 days– all other channels exhibit a fundamental period of 365.25 days.

In addition, our data distribution study (detailed in the supplementary material) highlights the significant impact of the yearly cycle on value ranges. Channels with a 365.25-day fundamental period demonstrate substantial seasonal fluctuations in their data distribution. The conventional ratio splitting also reflects this strong seasonality, as evidenced by notable shifts in the evaluation set's data distribution. On the contrary, our cycle-inclusive split significantly reduces data distribution differences between sets.

These findings are reinforced by our correlation analysis, which reveals that channel correlations vary significantly by season, particularly between *Winter* and *Summer*. Consequently, our analyses confirm that the minimum length for validation and evaluation sets should be one year.

**Experiment setting:** We assessed the performance of models using both our corrected versions and the Autoformer dataset with the conventional chronological ratio split of *7:1:2*, as commonly found in the literature. Additionally, we evaluated the performance of the four-year dataset using a chronological cycle-based split of *2/1/1* year, which we consider more suitable for a fair and comprehensive evaluation of MTSF tasks.

## 5.3 ELECTRICITY LOAD DIAGRAMS

The UCI ELD dataset is one of the most "confusing" for researchers tackling MTSF. Several variants of the dataset have been used in the literature, creating challenges for reproducibility and model comparison. For instance, Li et al. (2019) appear to have used the full dataset available on UCI, named *electricity-f*, though they did not describe their splitting strategy. Zhou et al. (2021), on the other hand, used a trimmed version of the dataset, both in terms of channels and the covered period, which they named *ECL*. In the Informer paper, the dataset was supposedly two years long with a *15/3/4* month split, but a closer inspection of their code revealed that they used three years (2012-2014) with a *7/1/2* ratio split.

In addition, *ECL* excluded clients with continuous zero consumption at the beginning of the selected period, resulting in a dataset with 321 channels. This decision likely aimed to prevent models

from misinterpreting prolonged zeros, which likely reflect unrecorded data rather than actual zero consumption. However, this assumption is inferred from our observations, as no explanation was provided by the authors. Wu et al. (2021) also used this reduced version of the dataset with a *7/1/2* ratio split, but they shifted the timestamps by more than four years and modified client IDs for unknown reasons, to finally name it *Electricity*. These uncertainties complicate the task of comparing models and ensuring the reproducibility of published results, **underscoring the need for a standardized benchmark dataset**.

**Inconsistencies:** Our analysis of the *ECL* dataset identified several inconsistencies. Some clients show anomalies or have extended periods of missing data. For instance, "MT_106" shows no consumption between March 18$^{th}$ and May 8$^{th}$, 2012, while "MT_298" experiences several prolonged zero-consumption periods and, when active, exhibits nearly constant consumption. Besides, "MT_182" ceases consumption after September 23$^{rd}$, 2012. These inconsistencies, present in the validation and evaluation sets, could distort model accuracy and introduce bias during MTSF tasks.

**Revised version:** Building on the revisions made to obtain *ECL*, we propose further refining the dataset by removing clients with prolonged zero-consumption periods and irregular consumption patterns (detailed in the supplementary material). To forecast effectively, models would need external information about these unusual patterns to differentiate them from normal behavior–information not provided in the dataset. Our revised version, "**PELD_1H_3Y_308**", consists of hourly observations over three years, with 308 clients remaining after filtering.

**Overall longest cycle:** Given the large number of channels in this dataset, we conducted a frequency analysis across all channels and studied the data distribution of a random sample. The frequency analysis shows that most channel have a fundamental period of 1 day, very few of them exhibit a fundamental period larger than one year but the other majority have a fundamental period of 365.33 days. In addition, the distribution analysis revealed significant seasonal variations in some channels, indicating a yearly dependency. Similarly to the weather datasets, the conventional ratio splitting also reflects this strong seasonality, as evidenced by notable shifts in the evaluation set's data distribution, while our cycle-inclusive split significantly mitigate data distribution differences between sets. As a result, we concluded that the minimum duration for validation and evaluation sets should be at least one year.

**Experiment setting:** We evaluated model performance on both the *ECL* and our revised *PELD* dataset using two splitting strategies: the conventional chronological ratio split of *7:1:2*, commonly found in the literature, and a chronological cycle-inclusive split of *1/1/1* year. We believe the latter provides a more suitable framework for fair and comprehensive evaluation of MTSF models.

# 6 EXPERIMENTS

## 6.1 BASELINES

Our objective is to demonstrate the impact of the corrections applied as well as the use of different data splits on models' performances. iTransformer, one of the SOTA transformer-based model for MTSF, is tested with all datasets. Among the linear-based models, we tested DLinear and NLinear which has shown the best overall performance for MPI as well as LCD and *ECL*, respectively. With *ECL* and LCD datasets, we also tested Informer, while for MPI, we also selected Autoformer, as the first paper to introduce this dataset.

## 6.2 PREDICTION SCENARIO

Considering that this paper does not aim to produce an extensive evaluation of the forecasting performances of various models released for TSF, but rather demonstrate the importance of clean dataset and cycle-inclusive split, we focus on the most represented prediction scenario in the literature. This scenario involves forecasting the next $F$ time steps ($F \in \{96, 192, 336, 720\}$), referred to as the prediction horizon, given $H$ time steps, $H = 96$. For each dataset and each model, we ran this scenario three times and reported the average and standard deviation of the error computed using both mean average error (MAE) and mean squared error (MSE). Experiments are conducted for both the usual ratio and the proposed cycle-inclusive splits.

In the ratio splits case, we included the results published in the corresponding paper to verify that our experiments with the original dataset produced similar results, thereby confirming the results with the proposed revised version and/or splitting method. Except for Portuguese electricity load diagrams (PELD), both univariate-to-univariate (U2U) and multivariate-to-multivariate (M2M) predictions are performed.

| | | 7:1:2 (i.e., 24544.8 / 3506.4 / 7012.8 days) | | | | | | Splitting |
| | | Original | | LCDWf_1H_4Y_USUNK | | LCDWi_1H_4Y_USUNK | | Dataset |
| | F | MSE | MAE | MSE | MAE | MSE | MAE | Metric |
|---|---|---|---|---|---|---|---|---|
| NLinear | 96 | 0.520±0.0012 | 0.498±0.0006 | **0.504**±0.0015 | **0.491**±0.0009 | 0.504±0.0015 | 0.492±0.0009 | |
| | 192 | 0.589±0.0002 | 0.542±0.0001 | **0.572**±0.0013 | **0.535**±0.0006 | **0.572**±0.0013 | 0.536±0.0006 | |
| | 336 | 0.624±0.0006 | 0.565±0.0001 | **0.606**±0.0006 | **0.558**±0.0001 | **0.606**±0.0006 | **0.558**±0.0001 | |
| | 720 | 0.688±0.0002 | 0.601±0.0001 | **0.669**±0.0002 | **0.595**±0.0000 | **0.669**±0.0002 | **0.595**±0.0000 | |
| Informer | 96 | 0.482±0.0030 | 0.490±0.0022 | 0.472±0.0031 | 0.483±0.0048 | **0.466**±0.0021 | **0.483**±0.0043 | |
| | 192 | 0.586±0.0104 | 0.548±0.0116 | 0.567±0.0025 | 0.540±0.0044 | **0.562**±0.0076 | **0.533**±0.0044 | |
| | 336 | 0.627±0.0067 | 0.586±0.0086 | **0.610**±0.0102 | **0.579**±0.0102 | 0.610±0.0103 | 0.580±0.0106 | |
| | 720 | 0.623±0.0137 | 0.586±0.0091 | **0.598**±0.0103 | **0.575**±0.0067 | 0.599±0.0094 | 0.576±0.0059 | |
| iTrans. | 96 | 0.509±0.0041 | 0.487±0.0022 | **0.492**±0.0044 | **0.480**±0.0022 | **0.492**±0.0043 | 0.481±0.0021 | |
| | 192 | 0.577±0.0026 | 0.533±0.0008 | **0.559**±0.0024 | **0.526**±0.0005 | **0.559**±0.0024 | **0.526**±0.0005 | |
| | 336 | 0.609±0.0029 | 0.555±0.0034 | 0.591±0.0026 | 0.548±0.0032 | **0.591**±0.0025 | **0.548**±0.0031 | |
| | 720 | 0.655±0.0033 | 0.583±0.0026 | **0.636**±0.0041 | **0.576**±0.0029 | 0.636±0.0042 | 0.576±0.0029 | |

Table 1: Results with LCD (informer weather dataset) for multivariate-to-multivariate predictions and a ratio splitting (*7:1:2*). Our experiments are run three times, both the average error and standard deviation are reported in this table.

## 6.3 MAIN RESULTS

Table 1 and Table 2 presents the experiment results obtained for LCD and M2M predictions with ratio and cycle-inclusive splits, respectively. Other results are available in the supplementary material. In these tables, **bold and underline** metric indicates the best performance (lowest value) for a given model and prediction horizon across all tested datasets. Values in blue and purple denote the best and second-best results for a given dataset and prediction horizon among all considered models.

### 6.3.1 COMPARATIVE PERFORMANCE: ORIGINAL VS. PROPOSED DATASETS

For PELD and LCD, the proposed datasets show better performance across the three considered models for both U2U and M2M predictions, as illustrated in Table 1. Despite the improvements, the ranking of models remains unchanged: iTransformer consistently outperforms others. Considering PELD, DLinear consistently provides the second-best results. Regarding LCD and M2M predictions, Informer shows competitive results, and especially excelling at prediction horizons 96 and 720. For LCD and U2U predictions, the second-best model depends on the metric and the prediction horizon, although Informer still leads in performances for horizon 720.

A different behavior is observed for MPI, where the original dataset yields better results, likely due to the presence of failure values ($-9999$). U2U predictions indicate very accurate forecasts with low MSE and MAE, suggesting that the failure values may interfere with the scaling process and, consequently, metrics computation (performed before inverse normalization). Testing an alternate version where failure values were replaced by zeros also showed worse performance compared to the original dataset, supporting this assumption and further advocating for clean datasets.

### 6.3.2 ASSESSMENT OF CYCLE-INCLUSIVE SPLITS

Using cycle-inclusive splits, models have better performance with corrected datasets compared to the original ones, as shown in Table 2. However, performance with this strategy are worse than with ratio splitting, likely due to fewer training samples or increased number of evaluation samples.

For PELD, cycle-inclusive splits slightly increase errors for iTransformer and DLinear, though iTransformer still achieves lower metrics. However, Informer's results worsen significantly, suggesting that it might have limitation when dealing with spatiotemporal MTSF datasets. Considering LCD, both M2M and U2U predictions show that this strategy shakes iTransformer's superiority, with Informer excelling on all prediction horizons for MSE. These results could imply that Informer's architecture might prevail when direct relationships exist between variables, like with Fahrenheit and Celsius. Regarding MPI, Autoformer continues to perform the worst, indicating that treating variables separately might be the appropriate architecture for MTSF lacking clear inter-relationships.

| | | 24/12/12 months (i.e., 17520 / 8784 / 8760 days) | | | | | | Splitting |
| | | Original | | LCDWf_1H_4Y_USUNK | | LCDWi_1H_4Y_USUNK | | Dataset |
| | F | MSE | MAE | MSE | MAE | MSE | MAE | Metric |
|---|---|---|---|---|---|---|---|---|
| **NLinear** | 96 | 0.582±0.0000 | 0.535±0.0000 | **0.566**±0.0001 | **0.528**±0.0000 | **0.566**±0.0001 | **0.528**±0.0000 | |
| | 192 | 0.660±0.0001 | 0.581±0.0001 | **0.644**±0.0001 | **0.575**±0.0001 | **0.644**±0.0001 | **0.575**±0.0001 | |
| | 336 | 0.680±0.0001 | 0.597±0.0001 | **0.663**±0.0001 | **0.591**±0.0001 | **0.663**±0.0001 | **0.591**±0.0001 | |
| | 720 | 0.741±0.0000 | 0.634±0.0000 | **0.725**±0.0000 | **0.628**±0.0000 | **0.725**±0.0000 | **0.628**±0.0000 | |
| **Informer** | 96 | 0.545±0.0064 | 0.532±0.0050 | **0.535**±0.0099 | 0.529±0.0063 | 0.544±0.0170 | **0.529**±0.0058 | |
| | 192 | 0.624±0.0013 | 0.570±0.0027 | 0.622±0.0047 | 0.571±0.0021 | **0.620**±0.0004 | **0.571**±0.0016 | |
| | 336 | 0.661±0.0030 | 0.611±0.0056 | **0.639**±0.0004 | **0.600**±0.0046 | 0.639±0.0011 | 0.601±0.0042 | |
| | 720 | 0.673±0.0128 | 0.619±0.0105 | 0.650±0.0084 | 0.609±0.0085 | **0.648**±0.0100 | **0.608**±0.0092 | |
| **iTrans.** | 96 | 0.562±0.0004 | 0.520±0.0015 | **0.546**±0.0002 | **0.513**±0.0012 | **0.546**±0.0002 | **0.513**±0.0012 | |
| | 192 | 0.644±0.0014 | 0.572±0.0028 | 0.627±0.0016 | **0.565**±0.0025 | **0.627**±0.0015 | **0.565**±0.0025 | |
| | 336 | 0.669±0.0010 | 0.590±0.0010 | **0.652**±0.0012 | **0.584**±0.0010 | 0.652±0.0014 | **0.584**±0.0010 | |
| | 720 | 0.723±0.0030 | 0.623±0.0029 | **0.702**±0.0031 | **0.612**±0.0025 | 0.703±0.0029 | **0.612**±0.0025 | |

Table 2: Results with LCD (informer weather dataset) for multivariate-to-multivariate predictions and a cycle splitting (*24/12/12* months). Our experiments are run three times, both the average error and standard deviation are reported in this table.

# 7 DISCUSSION AND LIMITATIONS

These results highlight the crucial importance of cleaning datasets used for TSF, as inconsistencies and improper data splitting can hinder both the learning process and the evaluation of models. This study contributes to a better understanding of how model performance varies with the nature of multivariate time series (MTS) datasets, though analysis of other TSF datasets is needed to confirm these findings. This study opens up to several directions for future research, such as training models to identify and label inconsistencies or paving the way for more sophisticated imputation methods, potentially using large language models (LLMs).

However, this study has limitations that need to be addressed. Firstly, we posit that some model architectures may be better suited to specific types of datasets. For instance, Informer excels with MTS datasets having clear inter-variable relationships but struggles with spatiotemporal MTS datasets. This observation suggests that Informer, which focuses on temporal tokens, may not be ideal for spatiotemporal datasets, unlike inverse architectures such as iTransformer, which focus on variate tokens. Further investigation could involve creating a cleaned four-year version of ELD (increasing training samples but significantly reducing the number of clients) to compare transformer-based models with their inverse versions using different data splitting. Moreover, the imputation quality remains uncertain without ground truth. Future work should include different metrics: 1. for overall performance, 2. for evaluating the performance for prediction horizons involving replaced values, and 3. for evaluating those with only original values. Finally, rather than relying on generic temporal embeddings, future models should be tailored to capture the most important temporal cycles present in the data. This could significantly improve the performance of models on datasets with strong seasonalities or long-term cycles.

# 8 CONCLUSION

This study underscores the critical importance of redefining and refining multivariate time series forecasting (MTSF) datasets to ensure accurate and reliable model performance. By systematically addressing inconsistencies and implementing cycle-aware splitting methods, our revised datasets provide a more robust foundation for fair comparison of MTSF models, paving the way for more reliable forecasting outcomes. While our work provides significant insights, it also opens avenues for further research. Expanding the analysis to other datasets will be essential to verify the generalizability of our findings. Additionally, the development of more advanced imputation techniques, possibly leveraging the power of large language models (LLMs), offers substantial potential to further enhance dataset accuracy and quality.

Though challenges remain–such as the need for ground truth validation and refinement of temporal embeddings–this work lays a solid foundation for future advances in MTSF. By addressing these challenges head-on, we can significantly improve the reliability of forecasting models, enabling better decision-making across a wide range of industries, from energy management to finance and beyond. Our efforts contribute to establishing stronger benchmarks for TSF tasks and highlight the importance of data integrity in advancing the field.

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
