# [Supplementary Material] Ensuring Fair Comparisons in Time Series Forecasting: Addressing Quality Issues in Three Benchmark Datasets

## A  Multivariate time series (MTS) datasets found in literature

Table 1 lists the multivariate time series (MTS) datasets used in the following time series forecasting (TSF) papers: [1] LogTrans (Li et al., 2019), [2] Informer (Zhou et al., 2021), [3] Autoformer (Wu et al., 2021), [4] Pyraformer (Liu et al., 2022), [5] FEDformer (Zhou et al., 2022), [6] Triformer (Cirstea et al., 2022), [7] RevIn (Kim et al., 2022), [8] Preformer (Du et al., 2023), [9] ETSformer (Woo et al., 2023), [10] Crossformer (Zhang & Yan, 2023), [11] D·NLinear (Zeng et al., 2023), [12] TimesNet (Wu et al., 2023), [13] PatchTST (Nie et al., 2023) [14] RLinear (Li et al., 2024) and [15] iTransformer (Liu et al., 2024).

| | electricity-f (fine) | electricity-c (coarse) | traffic-f (fine) | traffic-c (coarse) | Solar | Wind | M4-Hourly | ETTh1 & m1 | ETTh2 | ETTm2 | ECL | Weather | App Flow | Electricity | Electricity | Weather | Exchange | ILI | Traffic | PeMS-Bay | Market |
|---|---|---|---|---|---|---|---|---|---|---|---|---|---|---|---|---|---|---|---|---|---|
| [1] | ✓ | ✓ | ✓ | ✓ | ✓ | ✓ | ✓ | | | | | | | | | | | | | | |
| [2] | | | | | | | | ✓ | ✓ | | ✓ | ✓ | | | | | | | | | |
| [3] | | | | | | | | ✓ | ✓ | ✓ | | | | | | ✓ | ✓ | ✓ | ✓ | | |
| [6] | | | | | | | | ✓ | | | ✓ | ✓ | | | | | | | | | |
| [4] | | | | | | ✓ | | | ✓ | | | | ✓ | ✓ | | | | | | | |
| [5] | | | | | | | | ✓ | ✓ | ✓ | | | | | | ✓ | ✓ | ✓ | ✓ | | |
| [7] | | | | | | | | ✓ | ✓ | | ✓ | | | | | | | | | | |
| [8] | | | | | | | | ✓ | ✓ | ✓ | | | | | | ✓ | ✓ | ✓ | ✓ | | |
| [9] | | | | | | | | | ✓ | ✓ | | | | | | ✓ | ✓ | ✓ | ✓ | | |
| [10] | | | | | | | | ✓ | | | ✓ | ✓ | | | | | | ✓ | ✓ | | |
| [11] | | | | | | | | ✓ | ✓ | ✓ | | | | | | ✓ | ✓ | ✓ | ✓ | | |
| [12] | | | | | | | ✓ | ✓ | ✓ | ✓ | | | | | | ✓ | ✓ | ✓ | | | |
| [13] | | | | | | | | ✓ | ✓ | | | | | | | ✓ | | ✓ | ✓ | | |
| [14] | | | | | | | | ✓ | ✓ | ✓ | ✓ | | | | | ✓ | | | | | |
| [15] | | | | | ✓ | | | ✓ | ✓ | ✓ | ✓ | | | | | ✓ | ✓ | | ✓ | ✓ | ✓ |

Table 1: Table listing all the datasets used in the selected papers.

## B  Selected Dataset Descriptions

In the previous table, it is evident that some datasets share the same name, such as *Electricity* and *Weather*. However, these datasets are actually different either when looking at the number of features or looking at the splitting. Furthermore, all "electricity" datasets– *electricity-f*, *electricity-c*, *ECL* and both *Electricity*– are actually variants of the same dataset: UCI electricity load diagrams (ELD).

In this study, we examined three real-world datasets for inconsistencies: (1) Weather from Informer (Zhou et al., 2021), which includes 12 meteorological indicators collected hourly at a Surface Weather Station in the U.S. from 2010 to 2013; (2) Weather from Autoformer (Wu et al., 2021) that comprises 21 meteorological variables collected every 10 minutes in 2020 from one of the Weather Station at the Max-Planck-Institute of Biogeochemistry; (3) ELD from UCI first introduced in (Li et al., 2019), which records the hourly electricity consumption of 370 clients from 2011 to 2014. These datasets were selected to clarify potential confusion in the multivariate time series forecasting (MTSF) literature.

To align with existing discussions (Han et al., 2024; Zhao & Shen, 2024), dataset variables or features (i.e., weather indicators or electricity clients) is referred as *channels* throughout this document.

## B.1 WEATHER FROM INFORMER

This dataset is derived from the local climatological data (LCD) dataset, which originally includes weather observations of 20 indicators from multiple worldwide stations. The Informer subset represents data from a single U.S. station collected between 2010 and 2013 (a more detailed description is provided in Appendix J).

We propose three revised versions of this dataset: (a) **LCDWf_1H_4Y_USUNK**, where identified inconsistencies have been corrected; (b) **LCDWi_1H_4Y_USUNK**, where inconsistencies have been corrected and the usual target channel has been rounded to "integer" values, consistent with other temperature channels in the dataset; and (c) **LCDWr_1H_4Y_USUNK**, where inconsistencies have been corrected and the usual target channel is the actual Fahrenheit value converted from the Celsius value using the known relation.

The latter two versions aim to evaluate whether rounding to integer or direct conversion impacts predictive performance.

| | From Informer (Zhou et al., 2021) | Corrected Proposal | | |
|---|---|---|---|---|
| Dataset | Weather | LCDWf_1H_4Y_USUNK | LCDWi_1H_4Y_USUNK | LCDWr_1H_4Y_USUNK |
| Granularity | 1H | 1H | | |
| Number of time steps | 35 064 | 35 064 | | |
| Dataset Start Date | 2010-1-1 00:00 | 2010-1-1 00:00 | | |
| Dataset End Date | 2013-12-31 23:00 | 2013-12-31 23:00 | | |
| Number of channels | 12 | 12 + 6 inconsistency identifier | | |
| Target | WetBulbCelsius | WetBulbCelsius | WetBulbCelsiusInt | RealWetBulbCelsius |

Table 2: Details of the Weather Dataset from Informer against the proposed corrected version.

## B.2 WEATHER FROM AUTOFORMER

This dataset is derived from the Max-Planck-Institute (MPI) dataset, initially comprising weather observations from three stations at the Max-Planck-Institute of Biogeochemistry in Germany. The Autoformer subset uses data collected from the station located on the roof of the building during 2020 (a more detailed description is provided in Appendix K).

We propose three revised versions of this dataset: (a) **MPIW_10T_1Y_R** where identified inconsistencies have been corrected; (b) **MPIW_10T_4Y_R** which extends the dataset to four years with a 10-minute resolutions and where identified inconsistencies have been corrected; and (c) **MPIW_1H_4Y_R** which is the hourly resolution version of our extended revision.

| | From Autoformer (Wu et al., 2021) | Corrected Proposal | | |
|---|---|---|---|---|
| Dataset | Weather | MPIW_10T_1Y_R | MPIW_10T_4Y_R | MPIW_1H_4Y_R |
| Granularity | 10T | 10T | 10T | 1H |
| Number of time steps | 52 696 | 52 705 | 210 284 | 35 064 |
| Dataset Start Date | 2020-1-1 00:10 | 2020-1-1 00:10 | 2020-1-1 00:10 | 2020-1-1 00:00 |
| Dataset End Date | 2021-1-1 00:00 | 2021-1-1 00:00 | 2024-1-1 00:00 | 2023-12-31 23:00 |
| Number of channels | 21 | 21 + 5 inconsistency identifier | | |
| Target | OT | CO2 (ppm) | | |

Table 3: Details of the Weather Dataset from Autoformer against the proposed corrected version.

## B.3 ECL

This dataset is derived from the ELD dataset, originally providing 15-minute electricity consumption data for 370 clients collected between 2011 and 2014. The version used in many paper aggregates this to an hourly resolution and focuses on 321 clients from 2012 to 2014 – excluding clients with excessive zero data in the first year (a more detailed description is provided in Appendix L).

|  | **From Zhou et al. (2021)** | **Corrected Proposal** |
|---|---|---|
| **Dataset** | ECL | PELD_1H_3Y_308 |
| **Granularity** | 1H | 1H |
| **Number of time steps** | 35 064 | 35 064 |
| **Dataset Start Date** | 2011-1-1 00:00 | 2011-1-1 00:00 |
| **Dataset End Date** | 2013-12-31 23:00 | 2013-12-31 23:00 |
| **Number of channels** | 321 | 308 |
| **Target** | MT_320 | MT_320 |

Table 4: Details of the ECL from Informer against the proposed corrected version.

We propose a revised version of this dataset: **PELD_1H_3Y_308**, which further reduces the dataset to 308 clients by removing those with unusual profiles and the remaining clients with excessive missing data.

## C    EXPERIMENT DETAILS

### C.1    SETUP

We utilized the ADAM optimizer with an initial learning rate of 0.0001 and L2 loss for model optimization. Each experiment is run three times, with a total of 25 epochs and an early stopping patience set to 5.

### C.2    IMPLEMENTATION

We used the original PyTorch implementations of Informer [1], Autoformer [2], NLinear and Dlinear [3] as well as iTransformer [4]. All experiments were conducted using the default parameter values outlined in Table 5. Each iteration used a unique seed selected from the following set: $\{24, 1024, 2024\}$.

| Parameters | Informer | Autoformer | iTransformer | xLinear |
|---|---|---|---|---|
| d_model | 512 | | | |
| n_heads | 8 | | | - |
| e_layers | 2 | | | - |
| d_layers | 1 | | | - |
| s_layers | "3,2,1" | - | | |
| d_ff | 2048 | | | - |
| moving_avg | - | 25 | | - |
| factor | 5 | 3 | | - |
| padding | 0 | - | | |
| distil | True | | | - |
| dropout | 0.05 | | 0.1 | - |
| attn | "prob" | - | | |
| embed | "timeF" | | | |
| activation | gelu | | | |
| output_attention | False | | | |
| mix | "store_false" | - | | |
| num_workers | 0 | 10 | | |
| batch_size | 32 | | | |
| learning_rate | 0.0001 | | | |
| des | "Exp" | | | - |
| loss | mse | | | |
| lradj | "type1" | | | |
| channel_independence | - | | False | - |
| class_strategy | - | | Projection | - |

Table 5: List of the default parameters used in our experiments

### C.3    PLATFORM

All experiments were executed on one NVIDIA DGX-1 system equipped with Tesla P100 GPUs.

---

[1] https://github.com/zhouhaoyi/Informer2020

[2] https://github.com/thuml/Autoformer/tree/main

[3] https://github.com/honeywell21/DLinear

[4] https://github.com/thuml/iTransformer/tree/main

# D  RESULTS (COMPLETE)

| Dataset | Horizon | Zhou et al. (2021) | Wu et al. (2021) | Zeng et al. (2023) | | Liu et al. (2024) | | |
|---|---|---|---|---|---|---|---|---|
| | | InF | AutoF | NL | DL | DL | InF | iTransF |
| **LCD (Informer weather)** | 96 | - | - | - | - | - | - | - |
| | 192 | - | - | - | - | - | - | - |
| | 336 | 0.297 | - | - | - | - | - | - |
| | 720 | 0.359 | - | - | - | - | - | - |
| **MPI (Autoformer weather)** | 96 | - | 0.266 | 0.182 | 0.176 | 0.196 | 0.300 | 0.174 |
| | 192 | - | 0.307 | 0.225 | 0.220 | 0.237 | 0.598 | 0.221 |
| | 336 | - | 0.359 | 0.271 | 0.265 | 0.283 | 0.578 | 0.278 |
| | 720 | - | 0.419 | 0.338 | 0.323 | 0.345 | 1.059 | 0.358 |
| **ECL** | 96 | - | 0.201 | 0.141 | 0.140 | 0.197 | 0.274 | 0.148 |
| | 192 | - | 0.222 | 0.154 | 0.153 | 0.196 | 0.296 | 0.162 |
| | 336 | 0.489 | 0.231 | 0.171 | 0.169 | 0.209 | 0.300 | 0.178 |
| | 720 | 0.540 | 0.254 | 0.210 | 0.203 | 0.245 | 0.373 | 0.225 |

Table 6: MSE performances for multivariate-to-multivariate (M2M) predictions reported in the different literature papers.

| Dataset | Horizon | Zhou et al. (2021) | Wu et al. (2021) | Zeng et al. (2023) | | Liu et al. (2024) | | |
|---|---|---|---|---|---|---|---|---|
| | | InF | AutoF | NL | DL | DL | InF | iTransF |
| **LCD (Informer weather)** | 96 | - | - | - | - | - | - | - |
| | 192 | - | - | - | - | - | - | - |
| | 336 | 0.416 | - | - | - | - | - | - |
| | 720 | 0.466 | - | - | - | - | - | - |
| **MPI (Autoformer weather)** | 96 | - | 0.336 | 0.232 | 0.237 | 0.255 | 0.384 | 0.214 |
| | 192 | - | 0.367 | 0.269 | 0.282 | 0.296 | 0.544 | 0.254 |
| | 336 | - | 0.395 | 0.301 | 0.319 | 0.335 | 0.523 | 0.296 |
| | 720 | - | 0.428 | 0.348 | 0.362 | 0.381 | 0.741 | 0.349 |
| **ECL** | 96 | - | 0.317 | 0.237 | 0.237 | 0.282 | 0.368 | 0.240 |
| | 192 | - | 0.334 | 0.248 | 0.249 | 0.285 | 0.386 | 0.253 |
| | 336 | 0.528 | 0.338 | 0.265 | 0.267 | 0.301 | 0.394 | 0.269 |
| | 720 | 0.571 | 0.361 | 0.297 | 0.301 | 0.333 | 0.439 | 0.317 |

Table 7: MAE performances for M2M predictions reported in the different literature papers.

Table 6 and 7 list the M2M prediction performance reported in associated papers. Notably, the reproduced results for Informer and DLinear by the iTransformer authors deviate from the originally published results, which could potentially be attributed to differences in the random seed used. However, despite running each experiment three times, our reproduced results, presented in Tables 8 through 16, closely align with those reported by the iTransformer authors. This consistency suggests that our findings are in line with recent literature and that variations in performance might stem from differences in datasets or data-splitting strategies.

In these tables, **bold and underline** values represent the best performance (lowest value per row) for each model across different prediction horizons and datasets. Values highlighted in blue [resp. purple] denote the best [resp. second-best] performances (lowest value per column and prediction horizon) obtained for a given dataset among all considered models.

## D.1  PORTUGUESE ELECTRICITY LOAD DIAGRAMS

Table 8 presents predictions results for the ECL dataset using both the Informer version (ECL 321) and our revised version (PELD_1H_3Y_308), evaluated with a 7:1:2 ratio split. At first glance, our revised version appears to perform better than the Informer version –as depicted by the underlined values. However, this apparent improvement should be viewed with caution. The reduced number of channels in our revised dataset may contribute for the lower error rates. Specifically, for DLinear and iTransformer, the mean average error (MAE) for some prediction horizons shows similar average errors and standard deviations for both datasets, with iTransformer occasionally performing worse . These observations could be interpreted in two ways: (i) fewer channels lead to worse predictions overall, or (ii) the removed channels helped lower the error (i.e., removing some complexity and making prediction easier). Despite this uncertainty, we argue that our revised version offers a fairer comparison of model performance. Ultimately, with our revised dataset, the ranking trend remains consistent with the literature: iTransformer outperforms DLinear, which in turn surpasses Informer.

Table 9 reports model performance with cycle-inclusive splitting, where each training, validation, and evaluation set covers one year of data chronologically. With this splitting strategy, our revised dataset continues to yield the best performances. However, errors for Informer with this strategy

| | | 7:1:2 (767.2/109.6/219.2 days) | | | | | | Splitting |
| | | Published | | Produced | | | | Results |
| | | | | Reduced (ECL 321) | | Revised (PELD_1H_3Y_308) | | Dataset |
| | F | MSE | MAE | MSE | MAE | MSE | MAE | Metric |
|---|---|---|---|---|---|---|---|---|
| DLinear | 96 | 0.140 | 0.237 | 0.195±0.0001 | 0.278±0.0001 | **0.192**±0.0001 | **0.277**±0.0001 | |
| | 192 | 0.153 | 0.249 | 0.194±0.0000 | **0.280**±0.0000 | **0.191**±0.0000 | 0.280±0.0000 | |
| | 336 | 0.169 | 0.267 | 0.207±0.0001 | 0.296±0.0004 | **0.202**±0.0000 | **0.295**±0.0000 | |
| | 720 | 0.203 | 0.301 | 0.242±0.0001 | 0.329±0.0003 | **0.235**±0.0002 | **0.326**±0.0003 | |
| Informer | 96 | - | - | 0.286±0.0037 | 0.381±0.0023 | **0.249**±0.0022 | **0.355**±0.0019 | |
| | 192 | - | - | 0.293±0.0012 | 0.385±0.0026 | **0.250**±0.0049 | **0.356**±0.0044 | |
| | 336 | 0.489 | 0.528 | 0.305±0.0091 | 0.396±0.0071 | **0.274**±0.0084 | **0.376**±0.0046 | |
| | 720 | 0.540 | 0.571 | 0.332±0.0151 | 0.410±0.0102 | **0.279**±0.0087 | **0.381**±0.0078 | |
| iTrans. | 96 | 0.148 | 0.240 | **0.163**±0.0003 | **0.253**±0.0002 | **0.161**±0.0003 | 0.254±0.0002 | |
| | 192 | 0.162 | 0.253 | 0.175±0.0002 | **0.263**±0.0001 | **0.172**±0.0001 | 0.264±0.0001 | |
| | 336 | 0.178 | 0.269 | 0.192±0.0001 | **0.280**±0.0001 | **0.188**±0.0003 | 0.280±0.0002 | |
| | 720 | 0.225 | 0.317 | / | / | / | / | |

Table 8: Results with PELD (electricity dataset) for multivariate-to-multivariate predictions and a ratio splitting (*7:1:2*). Our experiments are run three times, both the average error and standard deviation are reported in this table.

increased significantly compared to the ratio splitting. This result may be due to the lack of samples in the training set or the larger number of samples in the evaluation set. A similar trend is observed with DLinear and iTransformer, though the impact is less severe, suggesting that these models are less sensitive to the training and evaluation set size. Conversely, it may indicate that Informer, which focuses on *temporal tokens*, struggles to capture channel relationships effectively, while iTransformer, which uses *variate tokens* and processes each channel independently, delivers more robust performance. To further validate this assumption, it would be beneficial to create a four-year version of the dataset. This version would increase the training sample size while significantly reducing the number of clients, allowing for a more comprehensive comparison between ratio-based and cycle-inclusive splitting strategies.

Overall, these experiments suggest that iTransformer and DLinear outperform Informer for spatiotemporal MTS datasets, with iTransformer achieving the best performance. These preliminary findings should be extended to other spatiotemporal MTS datasets, such as Traffic or Weather (datasets using multiple monitoring locations but focusing on only one observation).

| | | 1/1/1 year (366/365/365 days) | | | | Splitting |
| | | Reduced (ECL 321) | | Revised (PELD_1H_3Y_308) | | Dataset |
| | F | MSE | MAE | MSE | MAE | Metric |
|---|---|---|---|---|---|---|
| DLinear | 96 | 0.208±0.0004 | 0.288±0.0006 | **0.197**±0.0000 | **0.283**±0.0001 | |
| | 192 | 0.207±0.0004 | 0.290±0.0007 | **0.194**±0.0000 | **0.284**±0.0005 | |
| | 336 | 0.226±0.0002 | 0.310±0.0002 | **0.207**±0.0001 | **0.301**±0.0001 | |
| | 720 | 0.277±0.0023 | 0.350±0.0019 | **0.246**±0.0009 | **0.337**±0.0010 | |
| Informer | 96 | 0.624±0.0151 | 0.542±0.0087 | **0.480**±0.0257 | **0.510**±0.0192 | |
| | 192 | 0.639±0.0166 | **0.563**±0.0108 | **0.555**±0.0450 | 0.558±0.0299 | |
| | 336 | 0.680±0.0544 | 0.583±0.0284 | **0.520**±0.0204 | **0.535**±0.0144 | |
| | 720 | 0.879±0.1037 | 0.662±0.0500 | **0.589**±0.0121 | **0.571**±0.0068 | |
| iTrans. | 96 | 0.179±0.0003 | 0.260±0.0001 | **0.171**±0.0002 | **0.259**±0.0001 | |
| | 192 | 0.189±0.0002 | 0.269±0.0001 | **0.180**±0.0001 | **0.268**±0.0001 | |
| | 336 | 0.208±0.0001 | 0.287±0.0001 | **0.197**±0.0001 | **0.285**±0.0001 | |
| | 720 | / | / | / | / | |

Table 9: Results with PELD (electricity dataset) for multivariate-to-multivariate predictions and a cycle splitting (*1/1/1* year). Our experiments are run three times, both the average error and standard deviation are reported in this table.

## D.2 LOCAL CLIMATOLOGICAL DATA

### D.2.1 M2M

Table 10 presents the results for LCD and M2M predictions using the Informer version of the dataset, our corrected version (LCDWf_1H_4Y_USUNK), and our corrected version with the target in "integer" form (LCDWi_1H_4Y_USUNK) using a ratio splitting (*7:1:2*). Our corrected versions exhibit better performance across all the models used (as depicted by underlined results). The float version (LCDWf_1H_4Y_USUNK version), which keeps *WetBulbCelsius* as a *float*, generally performs better than the integer version than the integer version (LCDWi_1H_4Y_USUNK), suggesting that our corrections enhance model performance and dataset understanding. Therefore, such corrected versions should be preferred for fairer TSF model comparisons. iTransformer consistently outperforms

| | | Published | | Produced | | | | | | Results |
| | | | | 7:1:2 (i.e., 24544.8 / 3506.4 / 7012.8 days) | | | | | | Splitting |
| | | | | Original | | LCDWf_1H_4Y_USUNK | | LCDWi_1H_4Y_USUNK | | Dataset |
| | F | MSE | MAE | MSE | MAE | MSE | MAE | MSE | MAE | Metric |
| NLinear | 96 | - | - | 0.520±0.0012 | 0.498±0.0006 | **0.504**±0.0015 | **0.491**±0.0009 | 0.504±0.0015 | 0.492±0.0009 | |
| | 192 | - | - | 0.589±0.0002 | 0.542±0.0001 | **0.572**±0.0013 | **0.535**±0.0006 | **0.572**±0.0013 | 0.536±0.0006 | |
| | 336 | - | - | 0.624±0.0006 | 0.565±0.0006 | **0.606**±0.0006 | **0.558**±0.0001 | **0.606**±0.0006 | **0.558**±0.0001 | |
| | 720 | - | - | 0.688±0.0002 | 0.601±0.0001 | **0.669**±0.0002 | **0.595**±0.0000 | **0.669**±0.0002 | **0.595**±0.0000 | |
| Informer | 96 | - | - | 0.482±0.0030 | 0.490±0.0022 | 0.472±0.0031 | 0.483±0.0048 | **0.466**±0.0021 | **0.483**±0.0043 | |
| | 192 | - | - | 0.586±0.0104 | 0.548±0.0116 | 0.567±0.0025 | 0.540±0.0044 | **0.562**±0.0076 | **0.533**±0.0044 | |
| | 336 | 0.702 | 0.620 | 0.627±0.0067 | 0.586±0.0086 | **0.610**±0.0102 | **0.579**±0.0102 | 0.610±0.0103 | 0.580±0.0106 | |
| | 720 | 0.831 | 0.731 | 0.623±0.0137 | 0.586±0.0091 | **0.598**±0.0103 | **0.575**±0.0067 | 0.599±0.0094 | 0.576±0.0059 | |
| iTrans. | 96 | - | - | 0.509±0.0041 | 0.487±0.0022 | **0.492**±0.0044 | **0.480**±0.0022 | **0.492**±0.0043 | 0.481±0.0021 | |
| | 192 | - | - | 0.577±0.0026 | 0.533±0.0008 | **0.559**±0.0024 | **0.526**±0.0005 | **0.559**±0.0024 | **0.526**±0.0005 | |
| | 336 | - | - | 0.609±0.0029 | 0.555±0.0034 | 0.591±0.0026 | 0.548±0.0032 | **0.591**±0.0025 | **0.548**±0.0031 | |
| | 720 | - | - | 0.655±0.0033 | 0.583±0.0026 | **0.636**±0.0041 | **0.576**±0.0029 | 0.636±0.0042 | **0.576**±0.0029 | |

Table 10: Results with LCD (informer weather dataset) for multivariate-to-multivariate predictions and a ratio splitting (*7:1:2*). Our experiments are run three times, both the average error and standard deviation are reported in this table.

other models, with Informer often achieving the second-best results and occasionally surpassing iTransformer for specific prediction horizons.

| | | 24/12/12 months (i.e., 17520 / 8784 / 8760 days) | | | | | | Splitting |
| | | Original | | LCDWf_1H_4Y_USUNK | | LCDWi_1H_4Y_USUNK | | Dataset |
| | F | MSE | MAE | MSE | MAE | MSE | MAE | Metric |
| NLinear | 96 | 0.582±0.0000 | 0.535±0.0000 | **0.566**±0.0001 | **0.528**±0.0000 | **0.566**±0.0001 | **0.528**±0.0000 | |
| | 192 | 0.660±0.0001 | 0.581±0.0001 | **0.644**±0.0001 | **0.575**±0.0001 | **0.644**±0.0001 | **0.575**±0.0001 | |
| | 336 | 0.680±0.0001 | 0.597±0.0001 | **0.663**±0.0001 | **0.591**±0.0001 | **0.663**±0.0001 | **0.591**±0.0001 | |
| | 720 | 0.741±0.0000 | 0.634±0.0000 | **0.725**±0.0000 | **0.628**±0.0000 | **0.725**±0.0000 | **0.628**±0.0000 | |
| Informer | 96 | 0.545±0.0064 | 0.532±0.0050 | **0.535**±0.0099 | 0.529±0.0063 | 0.544±0.0170 | **0.529**±0.0058 | |
| | 192 | 0.624±0.0013 | 0.570±0.0027 | 0.622±0.0047 | 0.571±0.0021 | **0.620**±0.0004 | **0.571**±0.0016 | |
| | 336 | 0.661±0.0030 | 0.611±0.0056 | **0.639**±0.0004 | **0.600**±0.0046 | 0.639±0.0011 | 0.601±0.0042 | |
| | 720 | 0.673±0.0128 | 0.619±0.0105 | 0.650±0.0084 | 0.609±0.0085 | **0.648**±0.0100 | **0.608**±0.0092 | |
| iTrans. | 96 | 0.562±0.0004 | 0.520±0.0015 | **0.546**±0.0002 | **0.513**±0.0012 | **0.546**±0.0002 | **0.513**±0.0012 | |
| | 192 | 0.644±0.0014 | 0.572±0.0028 | **0.627**±0.0016 | **0.565**±0.0025 | 0.627±0.0015 | **0.565**±0.0025 | |
| | 336 | 0.669±0.0010 | 0.590±0.0010 | **0.652**±0.0012 | **0.584**±0.0010 | 0.652±0.0014 | **0.584**±0.0010 | |
| | 720 | 0.723±0.0030 | 0.623±0.0029 | **0.702**±0.0031 | **0.612**±0.0025 | 0.703±0.0029 | **0.612**±0.0025 | |

Table 11: Results with LCD (informer weather dataset) for multivariate-to-multivariate predictions and a cycle splitting (*24/12/12* months). Our experiments are run three times, both the average error and standard deviation are reported in this table.

Table 11 lists the models' performance on M2M predictions with cycle-inclusive splitting. Here, the training set spans approximately two years, while validation and evaluation sets each cover one year chronologically. Although the metrics are worse compared to ratio splitting –likely due to reduced training samples and increased evaluation samples– the corrected dataset versions still perform better. With the cycle-inclusive splitting, Informer consistently offers lower mean squared error (MSE), while iTransformer provides the second-best results. For MAE, both models show competitive performance, but iTransformer generally performs better.

These findings suggest that Informer might be more suitable for MTS datasets with direct relations among channels, such as the electricity transformer temperature (ETT) dataset. Moreover, using a cycle-inclusive split challenges iTransformer previous superiority.

D.2.2 UNIVARIATE-TO-UNIVARIATE (U2U)

Table 12 demonstrates that for LCD with ratio splitting and for U2U predictions, iTransformer clearly takes the lead over the other models. However, for the longest prediction horizon (i.e., 720), Informer achieves the best performance.

Table 13 indicates that cycle-inclusive splitting for U2U predictions also challenges iTransformer's superiority. On average, Informer performs better, although iTransformer shows the best performance on our corrected dataset (LCDWf_1H_4Y_USUN) in terms ofMAE.

As a conclusion, these experiments suggest that with ratio splits, iTransformer is the leading model for both M2M and U2U predictions. Contrary to previous studies, Informer outperforms NLinear and even surpasses iTransformer for the 720 prediction horizon, suggesting that *Probsparse* attention may be particularly beneficial for long prediction horizons. Further experiments comparing iTransformer, Informer, and inverse Informer for very large prediction horizons ($\geq$ 720) are re-

| | | Published | | Produced | | | | | | Results |
| | | | | 7:1:2 (i.e., 24544.8 / 3506.4 / 7012.8 days) | | | | | | Splitting |
| | | | | Original | | LCDWf_1H_4Y_USUNK | | LCDWi_1H_4Y_USUNK | | Dataset |
| | F | MSE | MAE | MSE | MAE | MSE | MAE | MSE | MAE | Metric |
|---|---|---|---|---|---|---|---|---|---|---|
| NLinear | 96 | - | - | 0.196±0.0003 | 0.316±0.0004 | **0.180**±0.0003 | **0.304**±0.0004 | 0.183±0.0004 | 0.309±0.0005 | |
| | 192 | - | - | 0.252±0.0001 | 0.363±0.0000 | **0.235**±0.0002 | **0.350**±0.0002 | 0.238±0.0002 | 0.354±0.0002 | |
| | 336 | - | - | 0.297±0.0003 | 0.398±0.0002 | **0.279**±0.0002 | **0.386**±0.0002 | 0.283±0.0002 | 0.390±0.0002 | |
| | 720 | - | - | 0.393±0.0001 | 0.465±0.0001 | **0.377**±0.0002 | **0.455**±0.0001 | 0.380±0.0002 | 0.458±0.0001 | |
| Informer | 96 | - | - | 0.206±0.0087 | 0.340±0.0182 | **0.184**±0.0123 | **0.321**±0.0200 | 0.193±0.0071 | 0.328±0.0147 | |
| | 192 | - | - | 0.246±0.0095 | 0.370±0.0059 | **0.221**±0.0136 | **0.349**±0.0164 | 0.223±0.0157 | 0.351±0.0169 | |
| | 336 | 0.297 | 0.416 | 0.268±0.0206 | 0.398±0.0172 | **0.258**±0.0195 | **0.390**±0.0170 | 0.259±0.0154 | 0.391±0.0128 | |
| | 720 | 0.359 | 0.466 | 0.247±0.0081 | 0.385±0.0110 | **0.237**±0.0073 | **0.378**±0.0108 | 0.238±0.0078 | 0.379±0.0109 | |
| iTrans. | 96 | - | - | 0.191±0.0012 | 0.310±0.0016 | **0.176**±0.0018 | **0.295**±0.0015 | 0.178±0.0014 | 0.299±0.0013 | |
| | 192 | - | - | 0.235±0.0017 | 0.351±0.0025 | **0.217**±0.0012 | **0.337**±0.0018 | 0.221±0.0013 | 0.341±0.0019 | |
| | 336 | - | - | 0.265±0.0032 | 0.373±0.0005 | **0.245**±0.0042 | **0.358**±0.0014 | 0.248±0.0039 | 0.362±0.0017 | |
| | 720 | - | - | 0.292±0.0043 | 0.394±0.0021 | **0.263**±0.0030 | **0.378**±0.0025 | 0.266±0.0034 | 0.382±0.0012 | |

Table 12: Results with LCD (informer weather dataset) for univariate-to-univariate predictions and a ratio splitting (*7:1:2*). Our experiments are run three times, both the average error and standard deviation are reported in this table.

| | | 24/12/12 months (i.e., 17520 / 8784 / 8760 days) | | | | | | Splitting |
| | | Original | | LCDWf_1H_4Y_USUNK | | LCDWi_1H_4Y_USUNK | | Dataset |
| | F | MSE | MAE | MSE | MAE | MSE | MAE | Metric |
|---|---|---|---|---|---|---|---|---|
| NLinear | 96 | 0.251±0.0000 | 0.358±0.0000 | **0.233**±0.0004 | **0.345**±0.0004 | 0.237±0.0004 | 0.350±0.0004 | |
| | 192 | 0.310±0.0001 | 0.407±0.0001 | **0.292**±0.0001 | **0.396**±0.0001 | 0.296±0.0001 | 0.400±0.0001 | |
| | 336 | 0.352±0.0000 | 0.438±0.0000 | **0.334**±0.0000 | **0.427**±0.0000 | 0.339±0.0000 | 0.431±0.0000 | |
| | 720 | 0.442±0.0000 | 0.505±0.0000 | **0.426**±0.0000 | **0.495**±0.0000 | 0.430±0.0000 | 0.498±0.0000 | |
| Informer | 96 | 0.258±0.0041 | 0.367±0.0018 | **0.231**±0.0028 | **0.345**±0.0012 | 0.236±0.0030 | 0.351±0.0025 | |
| | 192 | 0.299±0.0005 | 0.408±0.0035 | **0.283**±0.0041 | **0.397**±0.0052 | 0.287±0.0027 | 0.401±0.0024 | |
| | 336 | 0.317±0.0067 | 0.425±0.0040 | **0.305**±0.0047 | **0.416**±0.0024 | 0.308±0.0040 | 0.418±0.0035 | |
| | 720 | 0.305±0.0182 | 0.419±0.0128 | **0.283**±0.0064 | **0.408**±0.0078 | 0.289±0.0083 | 0.411±0.0085 | |
| iTrans. | 96 | 0.257±0.0037 | 0.361±0.0033 | **0.237**±0.0016 | **0.344**±0.0016 | 0.241±0.0016 | 0.349±0.0016 | |
| | 192 | 0.304±0.0030 | 0.401±0.0023 | **0.286**±0.0019 | **0.387**±0.0021 | 0.290±0.0018 | 0.391±0.0020 | |
| | 336 | 0.346±0.0041 | 0.430±0.0023 | **0.324**±0.0023 | **0.414**±0.0009 | 0.336±0.0117 | 0.422±0.0052 | |
| | 720 | 0.362±0.0059 | 0.445±0.0041 | **0.337**±0.0037 | **0.431**±0.0032 | 0.340±0.0054 | 0.434±0.0041 | |

Table 13: Results with LCD (informer weather dataset) for univariate-to-univariate predictions and a cycle splitting (*24/12/12* months). Our experiments are run three times, both the average error and standard deviation are reported in this table.

quired to investigate this finding. In addition, the results indicate that cycle-inclusive splits can re-define model rankings, with iTransformer being second-best to Informer for both M2M and U2U predictions. To confirm these observations, extending the study to other MTS datasets with direct relations among channels, such as ETT, is recommended.

### D.3 MAX-PLANCK-INSTITUTE

#### D.3.1 M2M

| | | 7:1:2 (8.4 / 1.2 / 2.4 months) | | | | | | | | Splitting |
| | | Original | | | | Simple | | Corrected | | Dataset |
| | | Published | | Produced | | | | | | Results |
| | F | MSE | MAE | MSE | MAE | MSE | MAE | MSE | MAE | Metric |
|---|---|---|---|---|---|---|---|---|---|---|
| DLinear | 96 | 0.176 | 0.237 | **0.195**±0.0002 | **0.255**±0.0020 | 0.242±0.0005 | 0.299±0.0012 | 0.252±0.0006 | 0.303±0.0009 | |
| | 192 | 0.220 | 0.282 | **0.237**±0.0008 | **0.296**±0.0013 | 0.293±0.0048 | 0.350±0.0082 | 0.306±0.0048 | 0.357±0.0092 | |
| | 336 | 0.265 | 0.319 | **0.285**±0.0015 | **0.336**±0.0024 | 0.341±0.0013 | 0.387±0.0026 | 0.356±0.0032 | 0.396±0.0032 | |
| | 720 | 0.323 | 0.362 | **0.349**±0.0027 | **0.387**±0.0045 | 0.412±0.0011 | 0.446±0.0010 | 0.424±0.0021 | 0.445±0.0023 | |
| Auto. | 96 | 0.266±0.007 | 0.336±0.006 | **0.262**±0.0094 | **0.340**±0.0094 | NA | NA | 0.328±0.0107 | 0.389±0.0116 | |
| | 192 | 0.307±0.024 | 0.367±0.022 | **0.341**±0.0154 | **0.396**±0.0109 | NA | NA | 0.392±0.0110 | 0.428±0.0099 | |
| | 336 | 0.359±0.035 | 0.395±0.031 | **0.375**±0.0275 | **0.413**±0.0259 | NA | NA | 0.461±0.0220 | 0.476±0.0254 | |
| | 720 | 0.419±0.017 | 0.428±0.014 | **0.501**±0.0350 | **0.492**±0.0245 | NA | NA | 0.568±0.0312 | 0.542±0.0222 | |
| iTrans. | 96 | 0.174 | 0.214 | **0.174**±0.0005 | **0.215**±0.0015 | 0.218±0.0021 | 0.258±0.0015 | 0.227±0.0028 | 0.263±0.0022 | |
| | 192 | 0.221 | 0.254 | **0.225**±0.0014 | **0.257**±0.0008 | 0.278±0.0002 | 0.306±0.0003 | 0.291±0.0012 | 0.313±0.0008 | |
| | 336 | 0.278 | 0.296 | **0.281**±0.0014 | **0.299**±0.0006 | 0.340±0.0010 | 0.351±0.0014 | 0.351±0.0012 | 0.357±0.0012 | |
| | 720 | 0.358 | 0.349 | **0.360**±0.0003 | **0.351**±0.0004 | 0.426±0.0004 | 0.407±0.0006 | 0.441±0.0024 | 0.414±0.0013 | |

Table 14: Results with **mpiw!** (autoformer weather dataset) for multivariate-to-multivariate predictions and a ratio splitting (*7:1:2*). Our experiments are run three times, both the average error and standard deviation are reported in this table.

Table 14 presents the M2M prediction results for MPI using ratio splitting (*7:1:2*). We compare three versions of the dataset: the original version from Autoformer, a simple version where failure values ($-9999$) are replaced by $0$ (Simple), and our corrected version using linear interpolation or context-aware imputation (MPIW_10T_1Y_R). Our corrected version performs the worst among

these datasets, and even the Simple version underperforms compared to the original dataset, which retains the failure values. iTransformer outperforms other models for both the original and corrected datasets, with DLinear providing the second-best results.

| | F | 24/12/12 months | | | | Splitting |
| --- | --- | --- | --- | --- | --- | --- |
| | | MPIW_10T_4Y_R | | MPIW_1H_4Y_R | | Granularity |
| | | MSE | MAE | MSE | MAE | Metric |
| DLinear | 96 | **0.417**±0.0001 | **0.392**±0.0002 | 0.504±0.0000 | 0.472±0.0000 | |
| | 192 | **0.478**±0.0000 | **0.436**±0.0001 | 0.562±0.0000 | 0.507±0.0000 | |
| | 336 | **0.542**±0.0000 | **0.479**±0.0002 | 0.601±0.0000 | 0.534±0.0001 | |
| | 720 | **0.615**±0.0001 | **0.525**±0.0001 | 0.664±0.0000 | 0.570±0.0000 | |
| Auto. | 96 | **0.416**±0.0056 | **0.409**±0.0030 | 0.598±0.0097 | 0.537±0.0072 | |
| | 192 | **0.551**±0.0069 | **0.500**±0.0047 | 0.642±0.0190 | 0.561±0.0091 | |
| | 336 | **0.613**±0.0288 | **0.534**±0.0150 | 0.669±0.0241 | 0.578±0.0105 | |
| | 720 | **0.668**±0.0055 | **0.568**±0.0028 | 0.713±0.0234 | 0.599±0.0099 | |
| iTrans. | 96 | **0.363**±0.0005 | **0.336**±0.0004 | 0.521±0.0015 | 0.470±0.0014 | |
| | 192 | **0.443**±0.0002 | **0.394**±0.0004 | 0.591±0.0002 | 0.510±0.0005 | |
| | 336 | **0.531**±0.0020 | **0.449**±0.0016 | 0.638±0.0012 | 0.540±0.0011 | |
| | 720 | **0.637**±0.0021 | **0.512**±0.0014 | 0.716±0.0004 | 0.578±0.0003 | |

Table 15: Results with MPI (autoformer weather dataset) for multivariate-to-multivariate predictions and a cycle splitting (*24/12/12* months). Our experiments are run three times, both the average error and standard deviation are reported in this table.

Table 15 shows the performance with cycle-inclusive splitting and extended dataset versions: MPIW_10T_4Y_R (with a 10-minute resolution) and MPIW_1H_4Y_R (with an hourly resolution). Here, the training set spans approximately two years, while validation and evaluation sets each cover one year chronologically. Results with cycle-inclusive splitting are significantly worse than with ratio splitting, likely due to the significant increased sample size in the evaluation set and its comprehensive coverage of all seasons. This suggests potential overfitting in models trained on only 8.5 months and evaluated on 2.5 months. We note that DLinear performs better with the hourly dataset, whereas iTransformer excels with the 10-minute resolution, indicating DLinear's difficulty with lower resolution cycles. Future work should verify if model performance varies across different seasons within the evaluation set. Notably, the hourly dataset performs worse than the 10-minute version, implying that our process for creating the hourly dataset may need revision.

Overall, these experiments suggest that iTransformer is the best model for MTS datasets. Extending these preliminary results to similar MTS datasets like Exchange would be valuable.

### D.3.2 U2U

| | F | 7:1:2 (8.4 / 1.2 / 2.4 months) | | | | | | | | Splitting |
| --- | --- | --- | --- | --- | --- | --- | --- | --- | --- | --- |
| | | Original | | | | Simple | | Corrected | | Dataset |
| | | Published | | Produced | | | | | | Results |
| | | MSE | MAE | MSE | MAE | MSE | MAE | MSE | MAE | Metric |
| DLinear | 96 | - | - | **0.005**±0.0003 | **0.056**±0.0027 | 0.387±0.0145 | 0.429±0.0071 | 0.555±0.0089 | 0.514±0.0029 | |
| | 192 | - | - | **0.006**±0.0001 | **0.064**±0.0006 | 0.476±0.0033 | 0.484±0.0021 | 0.651±0.0027 | 0.567±0.0012 | |
| | 336 | - | - | **0.006**±0.0002 | **0.064**±0.0019 | 0.527±0.0039 | 0.510±0.0025 | 0.743±0.0007 | 0.604±0.0002 | |
| | 720 | - | - | **0.006**±0.0002 | **0.066**±0.0021 | 0.595±0.0024 | 0.548±0.0012 | 0.947±0.0223 | 0.690±0.0093 | |
| Auto. | 96 | - | - | **0.003**±0.0002 | **0.041**±0.0017 | NA | NA | 0.767±0.0347 | 0.674±0.0182 | |
| | 192 | - | - | **0.004**±0.0009 | **0.047**±0.0063 | NA | NA | 0.767±0.0347 | 0.674±0.0182 | |
| | 336 | - | - | **0.004**±0.0002 | **0.050**±0.0016 | NA | NA | 0.940±0.0796 | 0.756±0.0353 | |
| | 720 | - | - | **0.004**±0.0005 | **0.052**±0.0030 | NA | NA | 1.205±0.0670 | 0.861±0.0259 | |
| iTrans. | 96 | - | - | **0.001**±0.0000 | **0.027**±0.0002 | 0.266±0.0020 | 0.360±0.0016 | 0.440±0.0103 | 0.456±0.0029 | |
| | 192 | - | - | **0.002**±0.0000 | **0.029**±0.0002 | 0.339±0.0007 | 0.414±0.0005 | 0.571±0.0043 | 0.532±0.0025 | |
| | 336 | - | - | **0.002**±0.0000 | **0.031**±0.0002 | 0.377±0.0028 | 0.444±0.0024 | 0.641±0.0157 | 0.573±0.0054 | |
| | 720 | - | - | **0.002**±0.0000 | **0.035**±0.0001 | 0.499±0.0046 | 0.516±0.0005 | 0.857±0.0124 | 0.671±0.0050 | |

Table 16: Results with MPI (autoformer weather dataset) for univariate-to-univariate predictions and a ratio splitting (*7:1:2*). Our experiments are run three times, both the average error and standard deviation are reported in this table.

For U2U predictions using ratio splitting, performance trends mirror those of M2M predictions: the corrected dataset yields worse results. The performance gap between the original and corrected datasets is significant, with the original dataset showing surprisingly low error values. This discrepancy likely arises from the impact of the failure value ($-9999$) on data normalization. Such an extreme value may distort z-normalization, affecting metrics calculated before reversing the normalization. Despite these issues, the corrected dataset maintains the same model ranking, with iTransformer performing best and DLinear second.

We believe our corrected dataset provides more accurate metric values, enabling fairer model comparisons.

| | | 24/12/12 months | | | | Splitting |
| | | MPIW_10T_4Y_R | | MPIW_1H_4Y_R | | Granularity |
| | F | MSE | MAE | MSE | MAE | Metric |
|---|---|---|---|---|---|---|
| DLinear | 96 | **0.393**±0.0001 | **0.411**±0.0002 | 0.425±0.0001 | 0.443±0.0001 | |
| | 192 | **0.436**±0.0001 | **0.441**±0.0001 | 0.476±0.0001 | 0.471±0.0000 | |
| | 336 | **0.473**±0.0000 | **0.465**±0.0001 | 0.514±0.0005 | 0.490±0.0002 | |
| | 720 | **0.523**±0.0001 | **0.493**±0.0001 | 0.568±0.0005 | 0.518±0.0003 | |
| Auto. | 96 | **0.498**±0.0181 | **0.493**±0.0116 | 0.501±0.0130 | 0.509±0.0118 | |
| | 192 | 0.695±0.0636 | 0.594±0.0324 | **0.563**±0.0100 | **0.540**±0.0042 | |
| | 336 | 0.715±0.0021 | 0.607±0.0032 | **0.657**±0.0373 | **0.578**±0.0047 | |
| | 720 | 0.717±0.0079 | 0.607±0.0031 | **0.653**±0.0268 | **0.590**±0.0127 | |
| iTrans. | 96 | **0.330**±0.0030 | **0.363**±0.0015 | 0.443±0.0032 | 0.455±0.0012 | |
| | 192 | **0.402**±0.0007 | **0.414**±0.0008 | 0.527±0.0020 | 0.502±0.0025 | |
| | 336 | **0.461**±0.0031 | **0.451**±0.0015 | 0.582±0.0031 | 0.528±0.0008 | |
| | 720 | **0.530**±0.0036 | **0.490**±0.0009 | 0.618±0.0040 | 0.551±0.0031 | |

Table 17: Results with MPI (autoformer weather dataset) for univariate-to-univariate predictions and a cycle splitting (*24/12/12* months). Our experiments are run three times, both the average error and standard deviation are reported in this table.

Contrary to M2M predictions, U2U models trained with cycle-inclusive splitting outperform those using ratio splitting, as shown in Table 17. However, similarly to M2M predictions, DLinear excels with the hourly version, while iTransformer performs best with the 10-minute resolution dataset.

These findings suggest the need to revisit the generation of the hourly dataset and the temporal embedding implementation, which may influence model performance.

# E  ADDITIONAL DISCUSSIONS

Our findings highlight the critical importance of clean datasets for improving model learning and ensuring fair model comparisons across TSF models. Notably, our proposed cycle-inclusive splitting strategy suggests that evaluating models over the longest temporal cycle offers a more complete assessment of TSF model efficiency. However, this outcome warrants further validation through experiments involving diverse datasets and alternative splitting strategies to fully assess the impact of varying sizes in training, validation, and evaluation sets.

Furthermore, our results suggest that no single model consistently excels in MTS forecasting. Instead, the optimal model or architecture may depend on the dataset's characteristics. Models focusing on *variate tokens* tend to perform better on datasets lacking explicit inter-channel relationships (e.g., datasets monitoring different physical quantities that are not directly intertwined or spatiotemporal datasets where delays between channels may occur). In contrast, architectures based on *temporal tokens* appear more efficient when clear and direct relationships exist between channels. For example, despite both being weather datasets, the key difference between LCD and MPI is the explicitness of the relationships between weather indicators. The LCD dataset includes both Celsius and Fahrenheit temperature readings, providing explicit interdependencies between channels. Conversely, The MPI dataset may exhibit less explicit relationships, where changes in one channel may influence others only after a delay. Consequently, models that effectively capture these relationships excel on datasets like LCD. Therefore, Informer, which prioritizes temporal tokenization, might captures "direct" inter-channel relationships more effectively, explaining its effectiveness with LCD. On the other hand, iTransformer, which focuses on variate tokens, and linear-based models that treat each channel independently, deliver superior performance on datasets like MPI and ELD, where inter-channel relationships are more complex.

These insights highlight the need for further experiments involving a broader range of transformer-based models and their variants. Such studies could refine our understanding of model suitability for different dataset types, potentially guiding the development of tailored architectures for specific MTS forecasting tasks.

# F  ADDITIONAL LIMITATIONS AND PERSPECTIVES

Beyond the limitations discussed in the main paper, additional issues must be addressed in the future.

Firstly, the current approach for identifying errors on a per-time-step basis is inadequate, particularly when only a subset of channels is affected by errors. To improve this point, we plan to create separate error masks that pinpoint error positions per time step and channel. Additionally, a dedicated file containing only the proposed corrections should be created. This approach would simplify the use of multiple correction versions, eliminating the need to manage multiple files and reducing storage complexity and space requirements.

Secondly, we believe that the temporal embedding implementation, inherited from the Informer model, also requires revision. Our experiments, particularly with the hourly and 10-minute resolution versions of the MPI dataset, reveal inconsistencies in its performance. We suggest revising the encoding scheme to better capture cyclical patterns, which are prevalent in TSF datasets. A more robust implementation could enhance the ability of models to represent and leverage temporal dynamics effectively.

## G  SOCIETAL IMPACTS

Time series forecasting (TSF) plays a pivotal role in optimizing resource management and facilitating strategic economic planning across various sectors. Accurate TSF contributes to (i) Enhanced resource utilization, (ii) Reduced service disruptions and operational costs, and (iii) Better-informed decision-making in domains such as energy, healthcare, finance, and logistics.

Our research underscores the necessity for clean datasets and rigorous model evaluation methodologies. By advancing these aspects, we aim to improve TSF accuracy, foster a deeper understanding of model strengths and limitations, and contribute to the development of more resilient, efficient systems that benefit society as a whole.

## H  HOSTING AND LICENSING

The following GitHub repository [5] is made available during the reviewing period and contains the following resources:

- Code used for dataset analysis
- Code used for dataset correction
- Implementation of cycle-inclusive splitting dataloader
- CSV files of the revised versions of these datasets
- Experiment results in markdown format

The original dataset used in this study are licensed as follows:

- Electricity load diagrams (ELD) is available from UCI and distributed under a Creative Commons Attribution 4.0 International (CC-BY-4.0) license;
- Local climatological data (LCD) is publicly available and according to the National Oceanic and Atmospheric Administration (NOAA), it is "open and free to use. There are no restrictions.";
- Max-Planck-Institute (MPI) is publicly available and distributed under a Creative Commons CC-BY-4.0 license.

The revised version of ELD and corrected versions of MPI adhere to the licensing terms of their original datasets. The corrected version of LCD will be distributed under a Creative Commons Attribution 4.0 International (CC-BY-4.0) license to ensure consistency with open-access principles.

This repository aims to provide transparency, foster reproducibility, and encourage further research in the field. Upon acceptance of this paper, it would be important to include these dataset versions on platforms such as HuggingFace [6] (as updated version of the existing datasets) or libraries such

---

[5]`https://anonymous.4open.science/r/2392-NDBT-2AED/`
[6]`https://huggingface.co/datasets?sort=trending`

as GluonTS [7]. This will increase their accessibility for future research and support comprehensive benchmarking of existing TSF models.

# I DATASETS PRESENTATION AND ANALYSIS METHOD

For AI models trained on data, presence of errors can severely hinder the learning of correlations and physical relationships, particularly if these errors are pervasive throughout the dataset. Furthermore, including time steps with inconsistencies in the evaluation set can significantly impair model assessment. A model performing well on an evaluation set that includes errors may either (i) excel on correct time steps while performing poorly on erroneous ones, demonstrating its ability to understand the data and its patterns, or (ii) it may perform moderately on both correct and erroneous time steps. However, the latter scenario does not necessarily indicate a robust model.

Benchmark datasets should ideally be free from such errors unless the objective explicitly targets predictions with erroneous data, tests model robustness to errors, or aims at anomaly detection. When evaluating TSF and comparing model performances, it is crucial to use datasets that are free from such errors, especially in the evaluation set. Therefore, it is essential to identify and annotate these problematic time steps, and to correct these errors or at least select appropriate metrics that account for them.

Our approach aims to address these concerns by proposing inconsistency-free dataset versions, accompanied by detailed annotations which will be beneficial for future research. The following sections present the inconsistencies found in each dataset, the method used to identify them and our proposed corrections.

## I.1 FREQUENCY

To investigate the dominant frequencies in each dataset channel, we applied fast Fourier transform (FFT) using the following method: (1) compute the trend of the considered channel, (2) apply the scipy FFT to the detrended channel, and (3) select the top K frequencies with the highest magnitude. Based on our experimentation, we adopted $k = 3$ in this paper. By combining domain knowledge with frequency analysis, we can determine the overall (considering all channels) longest cycle for each dataset.

## I.2 DISTRIBUTION

For each dataset (original and revised versions), distribution analyses were conducted for: (i) the entire dataset, (ii) per longest cycle (mostly year), and (iii) per data splitting strategy. To better understand the impact of the data splitting in regard of seasonal variations, these distributions were plotted per solar season: *Spring*, *Summer*, *Autumn* and *Winter*. When visually tractable, these distribution plots were performed for each channel. These plots can help researcher understand the impact of splitting strategies that can introduce significant distribution differences between sets.

## I.3 CORRELATION

For each dataset, we performed four correlation analyses using *Pearson*, *Kendall*, *Spearman* and *Cosine similarity* methods. To simplify the interpretation of the resulting heatmaps, we focused on highly correlated channels by ignoring values between $-0.75$ and $0.75$, which are represented as gray areas in the plots. Similarly to distribution plots, correlation analyses were conducted for: (i) the entire dataset, (ii) per longest cycle (considering all seasons), and (iii) per longest cycle and per solar season. Due to the large number of channels in ELD dataset and its variants, correlation plots for these datasets were excluded from the analysis.

---

[7]https://ts.gluon.ai/stable/index.html

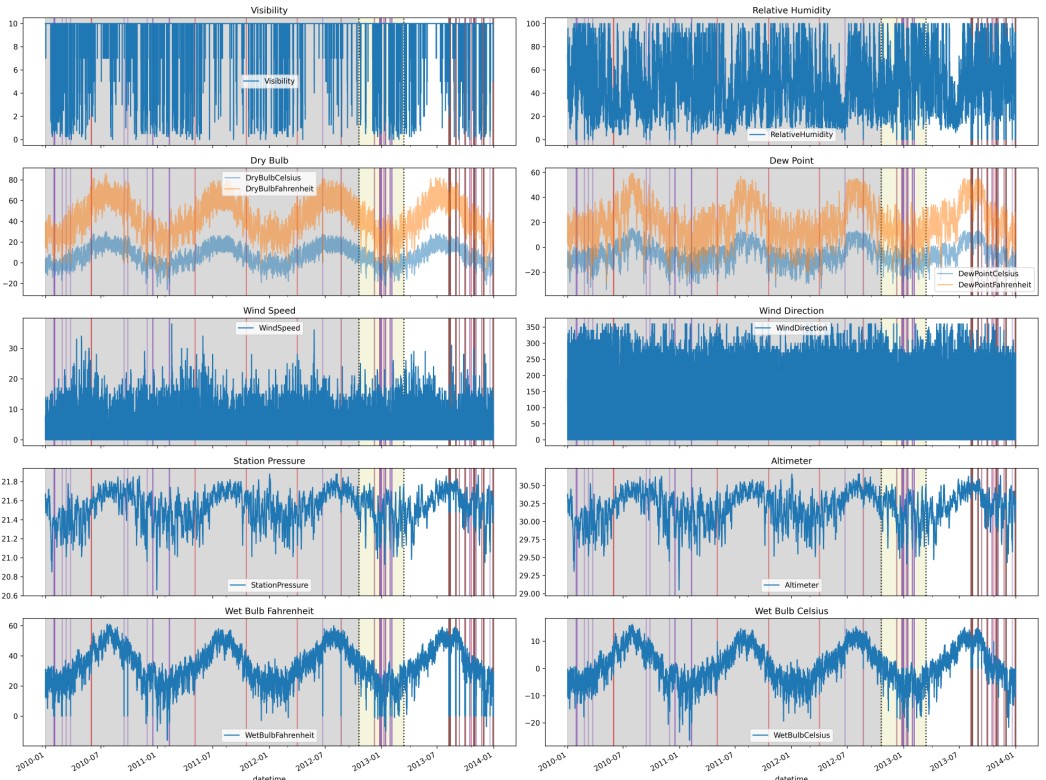

Figure 1: Overview of the weather indicators from the 4-year dataset used in Informer and collected from LCD. The gray background area represents the training period, and the yellow area represents the validation period as defined in the ratio splitting. Colored vertical lines indicate time steps where inconsistencies were found.

# J LOCAL CLIMATOLOGICAL DATA DATASET

## J.1 DESCRIPTION

The local climatological data (LCD) [8] dataset archives climatological data from approximately $20,000$ stations worldwide, of which around $14,000$ are active. For each station, surface observations are collected from various sources, including both manual and automated methods, and are managed by the National Centers for Environmental Information's Integrated Surface Data (ISD). The dataset includes records of 20 weather indicators, such as dry bulb temperature in both Celsius and Fahrenheit, relative humidity, and more. Data in the archive spans from January 1st, 1901, to the present day, although the availability of data may vary significantly by station.

## J.2 ANALYSIS

The LCD dataset is a multi-variable spatiotemporal dataset consisting of observations from various weather stations. Researchers can utilize this dataset to explore the spatiotemporal relationships among the monitored physical quantities, investigating how different weather indicators interact over time. Additionally, it offers opportunities to analyze how artificial intelligence (AI) models learn and interpret fundamental unit conversions, such as the relationship between Celsius and Fahrenheit temperatures. Ultimately, the dataset facilitates studies aimed at predicting future values of individual or multiple weather indicators based on historical observations.

---

[8]https://www.ncei.noaa.gov/data/local-climatological-data/

## J.3 ORIGINAL VERSION

The version of the dataset selected by Zhou et al. (2021) introduce in the Informer paper provides hourly weather observations from a U.S. station over a 4-year period. It includes the following 12 weather indicators:

- Visibility (float)
- Dry bulb Temperature: Fahrenheit (integer), Celsius (integer)
- Wet bulb Temperature Fahrenheit (integer)
- Dew point Temperature: Fahrenheit (integer), Celsius (integer)
- Relative humidity (integer)
- Wind speed (integer)
- Wind direction (integer)
- Pressure (float)
- Altimeter (float)
- Wet bulb Temperature Celsius (float)

The dataset's timestamp is unspecified regarding the time zone and spans from "2010-01-01 00:00:00" to "2013-12-31 23:00:00" (included).

### J.3.1 OVERALL ANALYSIS

Figure 1 displays grouped plots of the 12 weather indicators over the 4-year period. The gray and yellow areas represent the training and validation periods, respectively, as defined by the ratio splitting. At first glance, the dataset appears consistent; however, the vertical lines mark time steps where inconsistencies were identified.

### J.3.2 FREQUENCY ANALYSIS

|  | Fundamental | 2nd | 3rd |
|---|---|---|---|
| Visibility | 8766.0 (365.25) | 389.6 (16.23) | 313.1 (13.04) |
| DryBulbFahrenheit | 8766.0 (365.25) | 24.0 (1.00) | 17532.0 (730.50) |
| DryBulbCelsius | 8766.0 (365.25) | 24.0 (1.00) | 17532.0 (730.50) |
| WetBulbFahrenheit | 8766.0 (365.25) | 24.0 (1.00) | 4383.0 (182.62) |
| DewPointFahrenheit | 8766.0 (365.25) | 4383.0 (182.62) | 2922.0 (121.75) |
| DewPointCelsius | 8766.0 (365.25) | 4383.0 (182.62) | 2922.0 (121.75) |
| RelativeHumidity | 24.0 (1.00) | 8766.0 (365.25) | 4383.0 (182.62) |
| WindSpeed | 24.0 (1.00) | 12.0 (0.50) | 4383.0 (182.62) |
| WindDirection | 24.0 (1.00) | 12.0 (0.50) | 4383.0 (182.62) |
| StationPressure | 8766.0 (365.25) | 4383.0 (182.62) | 407.7 (16.99) |
| Altimeter | 8766.0 (365.25) | 4383.0 (182.62) | 407.7 (16.99) |
| WetBulbCelsius | 8766.0 (365.25) | 24.0 (1.00) | 4383.0 (182.62) |

Table 18: Frequency analysis of the original Weather dataset from Informer. The first value is the period in number of time steps the value in parentheses is the correspondence in days.

This study reveals that most channels exhibit a primary cycle of one year (8766 time steps, eq. 365.25 days). However, exceptions include *Relative Humidity*, *Wind Speed* and *Wind Direction*, which demonstrate a one-day cycle–an expected outcome for wind due to its inherently chaotic nature. The most prominent cycles identified in this dataset include one year, half a year, two years, one day, half a day, and approximately half a month. As a results, the longest dominant cycle across all channels is one year. Consequently, a cycle-inclusive splitting strategy should ensure that each set (training, validation, and evaluation) covers at least one full year to represent these temporal patterns effectively.

### J.3.3 CORRELATION ANALYSIS

Figures 2 represents the channels correlation of the Weather dataset from Informer using the different methods mentioned in Appendix I.3. For all metrics, we can observe similar patterns:

1. By row: For each year, the correlations remain consistent or show only slight variations;

2. By column: Within a given period, when divided by solar seasons, the correlations between channels can vary significantly. For instance, *Winter* and *Spring* exhibit notable differences compared to *Summer* and *Autumn*. Additionally, while differences between *Winter* and *Spring*, as well as *Summer* and *Autumn*, are less pronounced, they are still evident.

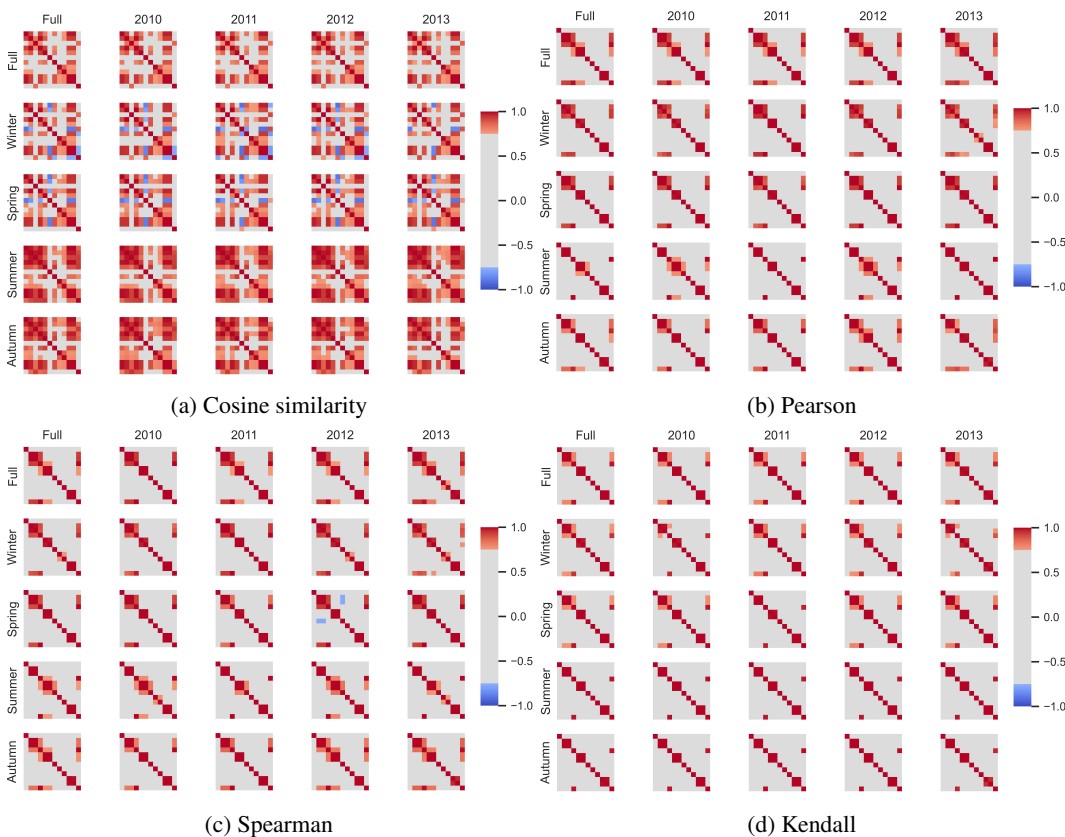

Figure 2: Weather Dataset from Informer - Channels correlation for the full dataset, per year and per season.

An efficient model for MTSF should be able to efficiently capture these seasonal variations and dynamically adapt the dependencies based on the input season.

### J.3.4 DATA DISTRIBUTION ANALYSIS

Figure 3 provides various distribution plots for the original dataset. As expected, most weather indicators exhibit distinct seasonal distributions, with the exception of *Visibility*, *Wind Speed* and *Wind Direction*. These seasonal fluctuations are especially significant for most of the channels. In addition, some variations can be observed across years,such as changes in the distribution of *Visibility* in 2012 and 2013 compared to 2010 and 2011. Any efficient MTSF model should be able to account for such differences and patterns in order to ensure robust performance.

Although the dataset provides enough data to consider a splitting strategy based on the longest cycle, Zhou et al. (2021) opted for a ratio splitting (*7:1:2 ∼ 28/10/10*-month). This approach is not optimal for time series and chronological data because neither the validation nor the evaluation periods encompass a complete ***longest cycle***, which, according to our frequency analysis, is one year.

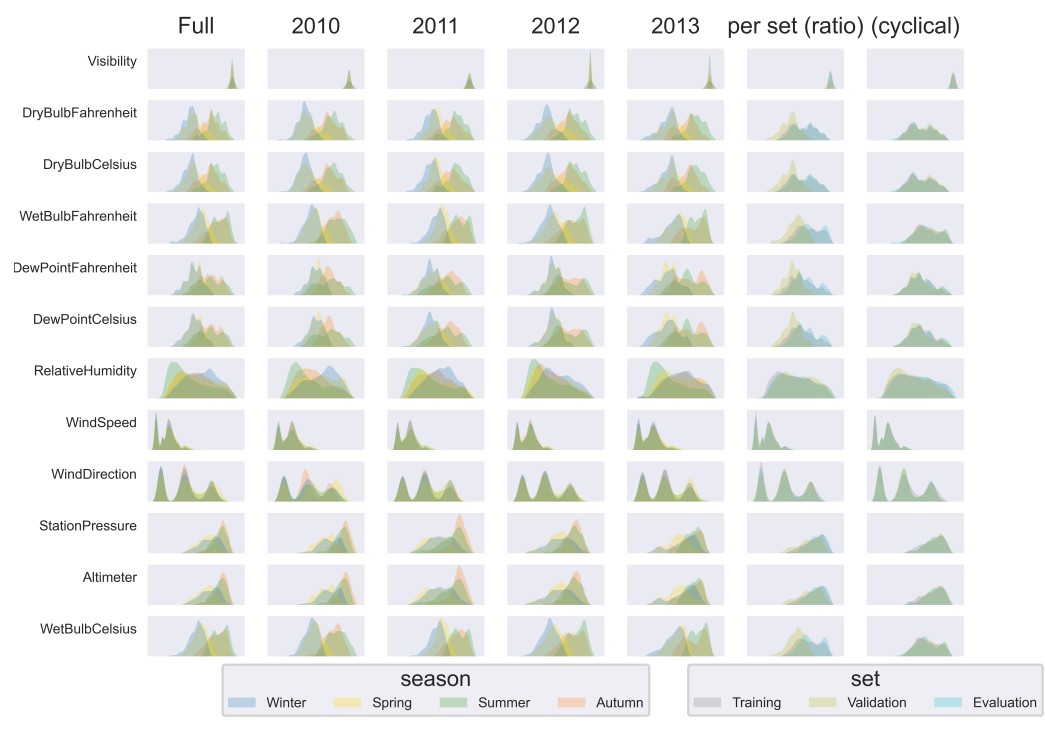

Figure 3: Weather Dataset from Informer - Distribution plots per channel. The last two columns illustrate data distribution per splitting strategy: ratio and our proposal cycle-inclusive. The other columns illustrate the data distribution for the whole datasets and per year, with a differentiation per season.

Consequently, the training process is skewed to optimize performance for the selected validation period (i.e., *Winter*), while the evaluation period (i.e., *Spring*) does not fully test the model's ability to generalize across the full cycle. Our distribution and correlation analyses further highlight notable differences between these periods, reinforcing the limitations of the ratio-based approach. In addition, Figure 3 demonstrates that ratio splitting introduces significant distribution discrepancies between the training, validation, and evaluation sets. In contrast, our cycle-inclusive splitting strategy mitigates these discrepancies, ensuring that the model is trained using a score that reflects the longest cycle and evaluated over a period covering an entire cycle.

### J.3.5 INCONSISTENCIES PRESENTATION

We identified several inconsistencies in the LCD dataset:

1. **Missing Values Set to Zero**: Figure 4 highlights instances where missing values were inappropriately set to zero. For example, it is not plausible for both Fahrenheit and Celsius values of the same indicator (e.g., Dew Point Temperature in the figure) to be zero simultaneously at a given time step. Moreover, having the relative humidity also set to zero at this time step is inconsistent with surrounding values, which are close to $100\%$. Such an example advocates for missing data filled with zero.

2. **Incorrect Fahrenheit to Celsius Conversion**: For the Wet Bulb Temperature feature, while the expected conversion from Fahrenheit to Celsius is affine, we observed significant errors. Figure 5a shows that for a Fahrenheit value of $32°F$, the corresponding Celsius values range between $-9.5°C$ and $9.9°C$, which is unacceptably wide and indicates a problem with the data.

3. **Inconsistent Altimeter and Surface Pressure Relationship**: Figure 5b illustrates a somewhat staircase relationship between Altimeter and Surface Pressure. However, inconsistencies are evident when certain pressure values (e.g., 21.478686), where the altimeter values deviate significantly from the expected pattern. Such inconsistencies hinder the model's ability to learn this relationship accurately.

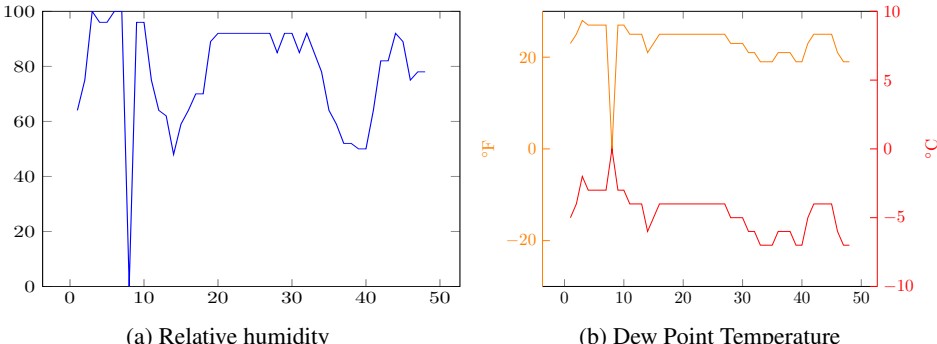

(a) Relative humidity          (b) Dew Point Temperature

Figure 4: Visualization of LCD's Relative Humidity and Dew Point Temperature for January 28-29, 2010. This figure highlights instances of missing values improperly set to zero, with both Relative Humidity and Dew Point Temperature showing simultaneous zero values, which are inconsistent with expected meteorological behavior.

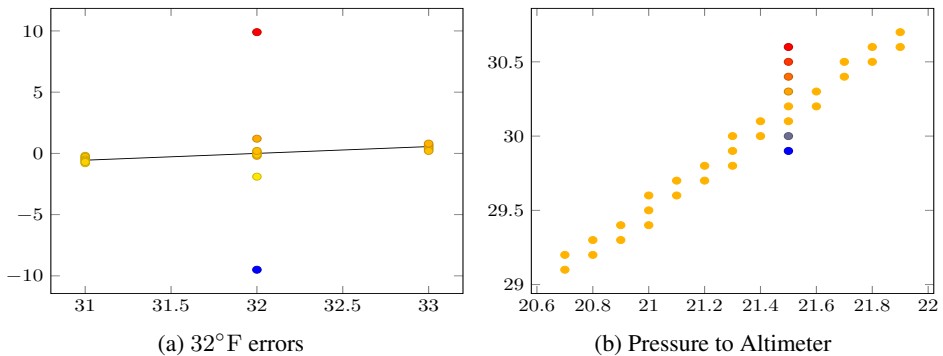

(a) $32°$F errors          (b) Pressure to Altimeter

Figure 5: Visualization of Errors for $32°$F Conversion and Altimeter-to-Pressure Relationship. In the left panel, the black line represents the affine function used for converting Fahrenheit to Celsius using the formula $C = (F - 32) * 5/9$. The red and blue points illustrate discovered inconsistencies in the dataset. In the right panel, the relationship between Altimeter and Surface Pressure is shown, highlighting deviations from the expected "staircase" pattern in Altimeter values for the pressure value 21.478686

### J.3.6 INCONSISTENCIES APPEARANCE

As shown by the colored vertical lines in Figure 1, inconsistencies are widespread throughout this dataset, but particularly present in the evaluation period.

The **red** vertical lines indicate time steps where errors in the $32°$F values were identified, while the **purple** lines highlight time steps where missing data were inaccurately filled with zeros for multiple variables. **Pink** lines mark time steps where errors in Wet Bulb temperature conversions were found, and **brown** lines depict time steps where inconsistencies between pressure and altimeter values occurred.

### J.4 PROPOSED CORRECTION

To address the issues of missing data filled with zero and altimeter-to-pressure errors, we propose the following process outlined in the main paper: (i) replace erroneous values with NaN, (ii) apply

linear interpolation for isolated errors, and (iii) use either context-aware when possible or linear interpolation for consecutive errors.

Regarding the $32°$F errors, we recommend replacing inconsistent values with values computed from the observed data and the well-known affine conversion. Specifically, $32°$F converted values are identified as errors, if they deviate beyond the standard deviation of the correct converted data.

### J.4.1 IDENTIFY INCONSISTENCIES

Six additional columns have been appended to the CSV file in order to identify the time step where inconsistencies were corrected:

- $32F\_errors$: identify time steps with $32°$F errors
- $common\_conversion\_errors$: flags time steps where missing value were filled with zeros for a subset of variables.
- $wet\_conversion\_errors$: marks time steps with other conversion errors on Wet Bulb Temperature features.
- $pressure\_relation\_errors$: highlights time steps where altimeter-to-pressure errors were corrected.
- $is\_ts\_missing$: indicates time steps that were missing in the original dataset.
- $is\_ts\_modified$: logs all time steps where corrections were applied.

### J.4.2 OVERALL ANALYSIS

Figure **??** displays grouped plots of the corrected 12 weather indicators over the 4-year period from the **LCDWf_1H_4Y_USUNK** version. The gray and yellow areas represent the training and validation periods, respectively, as defined by the cycle-inclusive splitting. No data stand out which would imply that there are no errors in this version.

### J.4.3 FREQUENCY ANALYSIS

|  | Fundamental | 2nd | 3rd |
| --- | --- | --- | --- |
| Visibility | 8766.0 (365.25) | 389.6 (16.23) | 313.1 (13.04) |
| DryBulbFahrenheit | 8766.0 (365.25) | 24.0 (1.00) | 17532.0 (730.50) |
| DryBulbCelsius | 8766.0 (365.25) | 24.0 (1.00) | 17532.0 (730.50) |
| WetBulbFahrenheit | 8766.0 (365.25) | 24.0 (1.00) | 4383.0 (182.62) |
| DewPointFahrenheit | 8766.0 (365.25) | 4383.0 (182.62) | 2922.0 (121.75) |
| DewPointCelsius | 8766.0 (365.25) | 4383.0 (182.62) | 2922.0 (121.75) |
| RelativeHumidity | 24.0 (1.00) | 8766.0 (365.25) | 4383.0 (182.62) |
| WindSpeed | 24.0 (1.00) | 12.0 (0.50) | 4383.0 (182.62) |
| WindDirection | 24.0 (1.00) | 12.0 (0.50) | 4383.0 (182.62) |
| StationPressure | 8766.0 (365.25) | 4383.0 (182.62) | 407.7 (16.99) |
| Altimeter | 8766.0 (365.25) | 4383.0 (182.62) | 407.7 (16.99) |
| WetBulbCelsius | 8766.0 (365.25) | 24.0 (1.00) | 4383.0 (182.62) |

Table 19: **LCDWf_1H_4Y_USUNK** - Frequency analysis. The first value is the period in number of time steps the value in parentheses is the equivalent in days.

The revised version does not differ from the original dataset in terms of dominant frequencies. Therefore, the longest cycle remains one year.

### J.4.4 CORRELATION ANALYSIS

The correlation patterns observed in the revised dataset are consistent with those in the original dataset. This observation suggests that models still need to be capable of adapting dependencies based on seasonal variations.

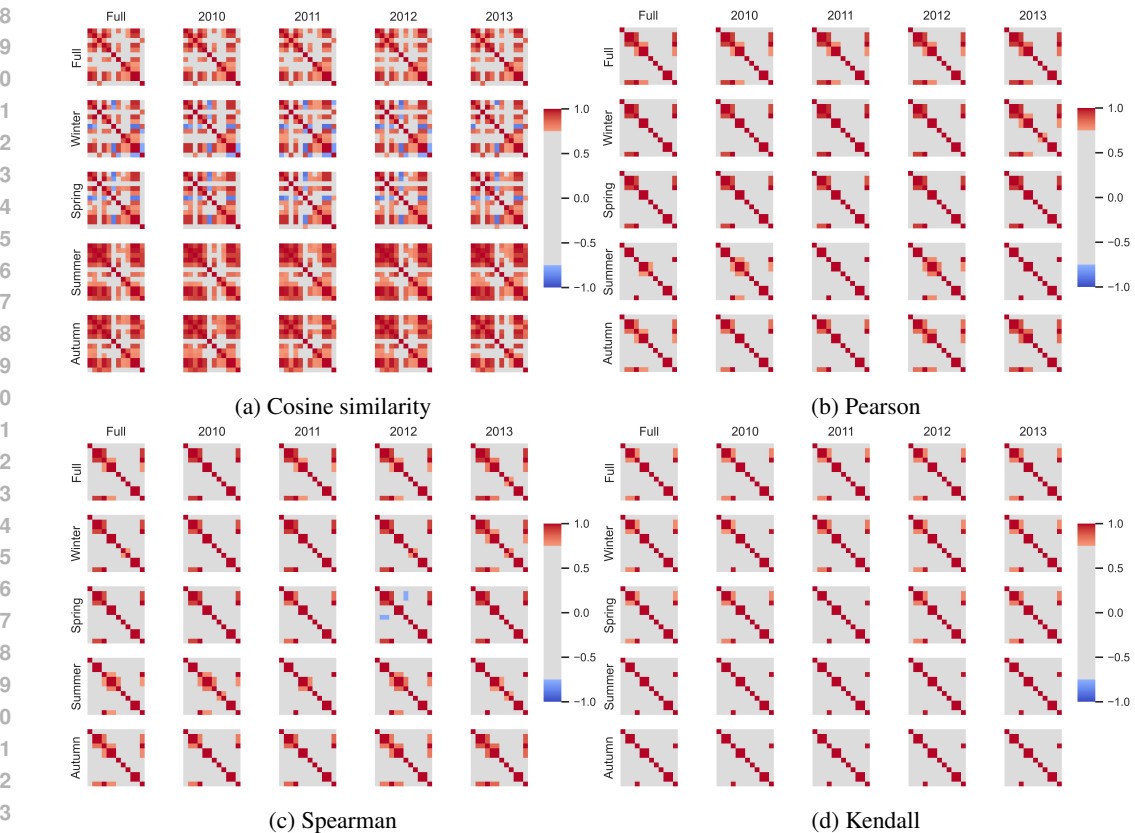

Figure 6: **LCDWf_1H_4Y_USUNK** - Channels correlation for the full dataset, per year and per season.

### J.4.5 DATA DISTRIBUTION ANALYSIS

Figure 7 presents the distribution plots for the revised dataset: **LCDWf_1H_4Y_USUNK**. The corrections applied to address inconsistencies and errors have not altered the dataset's inherent properties. While data distributions continue to vary significantly by season, our cycle-inclusive splitting strategy ensures better distributional similarity between the training, validation, and test sets. This strategy makes the dataset more suitable for benchmarking and facilitates more reliable model evaluations.

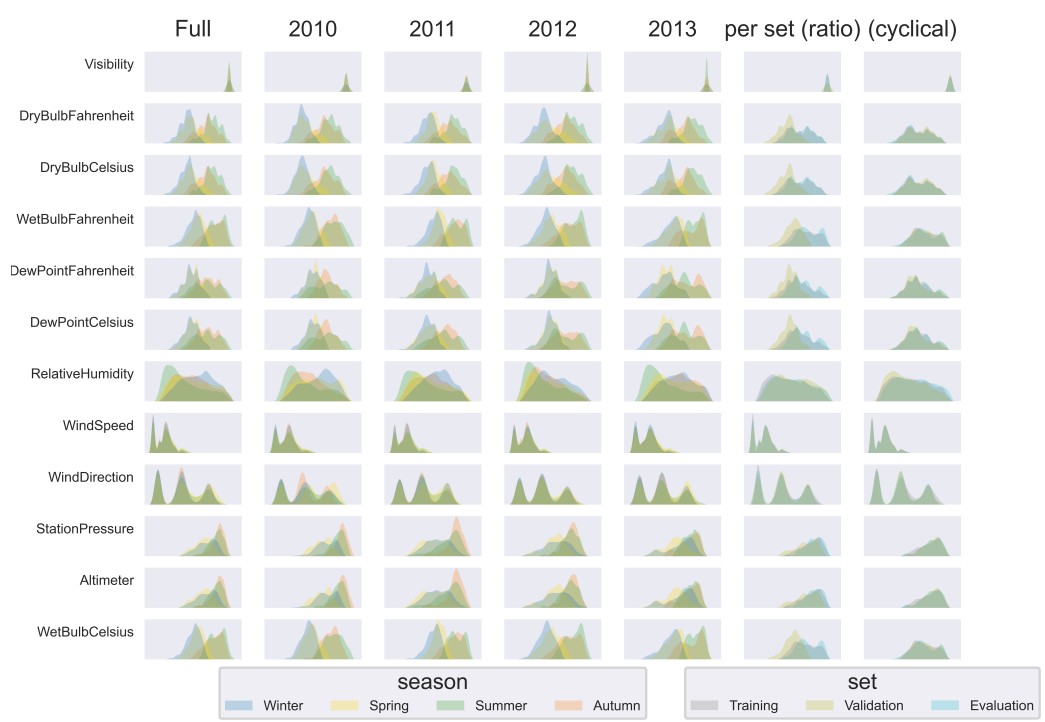

Figure 7: **LCDWf_1H_4Y_USUNK** - Distribution plots per channel. The last two columns illustrate data distribution per splitting strategy: ratio and our proposal cycle-inclusive. The other columns illustrate the data distribution for the whole datasets and per year, with a differentiation per season.

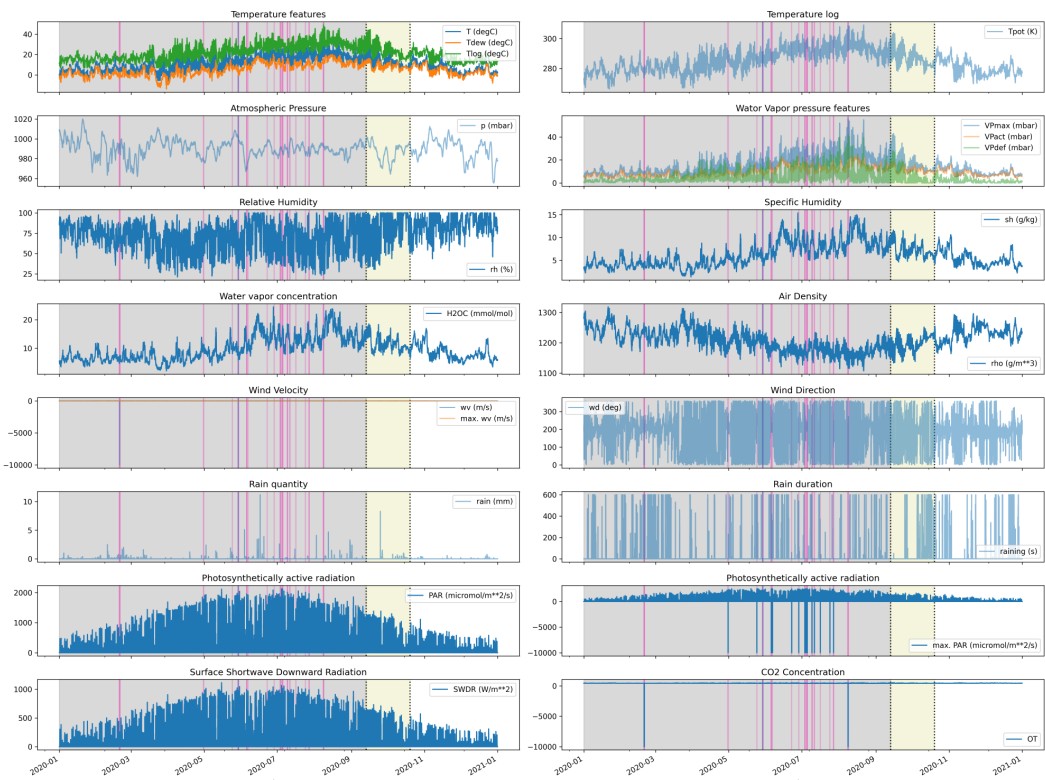

Figure 8: Overview of the weather indicators from the 1-year dataset used in Autoformer and collected from MPI. The gray background area represents the training period, while the yellow area denotes the validation period as defined in the ratio splitting. Colored vertical lines indicate time steps where inconsistencies were identified.

## K  MAX-PLANCK-INSTITUTE DATASET

### K.1  DESCRIPTION

The Max-Planck-Institute (MPI) [9] dataset provides weather measurements collected from three distinct weather stations. One of these stations, *WS Beutenberg*, is located atop the building's roof of the Max-Planck-Institute for Biogeochemistry. It comprises 21 weather indicators, including air temperature and humidity, recorded at 10-minute intervals. This dataset spans from "2003-11-24 16:00:00" to the present days.

### K.2  ANALYSIS

Similarly to LCD, the MPI dataset is a multi-variable spatiotemporal dataset. When focusing on data from a single station, the resulting dataset is a MTS dataset capturing observations from a specific location in Germany via various sensors. These observations exhibit variations intricately linked to Earth's revolution (year, seasons) and rotation (day, hours). Other factors, such as human behavior and global warming, likely contribute to fluctuations in the recorded parameters. This dataset offers the opportunity for models to discern relationships between these indicators and leverage such insights to predict one or multiple variables.

---

[9]https://www.bgc-jena.mpg.de/wetter/

### K.3 ORIGINAL VERSION

Wu et al. (2021) selected a 1-year period–the year 2020–of the data available from *WS Beutenberg*, located on the roof of the Max-Planck-Institute for Biogeochemistry. It has a 10-minute resolution. This datasets includes weather observations of the following 21 indicators:

- Atmospheric Pressure (p (mbar))
- Air Temperature (T (degC))
- Potential Temperature (Tpot (K))
- Dew Point Temperature (Tdew (degC))
- Relative Humidity (rh (%))
- Saturation Water Vapor Pressure (VPmax (mbar))
- Actual Water Vapor Pressure (VPact (mbar))
- Water Vapor Pressure Deficit (VPdef (mbar))
- Specific Humidity (sh (g/kg))
- Water Vapor Concentration ($H_2OC$ ($\mu$mol/mol))
- Air Density (rho (g/m$^3$))
- Wind Velocity (wv (m/s))
- Maximum Wind Velocity (max. wv (m/s))
- Wind Direction (wd (deg))
- Precipitation Amount (rain (mm))
- Precipitation Duration (raining (s))
- Surface Shortwave Downward Radiation (SWDR (W/m$^2$))
- Photosynthetic Active Radiation (PAR ($\mu$mol/m$^2$/s))
- Maximum Photosynthetic Active Radiation (max. PAR ($\mu$mol/m$^2$/s))
- Internal Logger Temperature (Tlog (degC))
- $CO_2$ concentration ($CO_2$ (ppm))

The timestamp are provided without any specific time zone. The dataset spans from "2020-01-01 00:10:00" to "2021-01-01 00:10:00" (included).

#### K.3.1 OVERALL ANALYSIS

Figure 8 presents the plots of the different weather indicators. The gray area represents the training period, while the yellow area indicates the validation period, as defined by the ratio splitting strategy. The presence of errors is particularly noticeable in plots where the y-axis extends to values as extreme as $-10000$, which are clearly unrealistic for any of the weather indicators monitored.

In addition, as this dataset spans only one year, the ratio splitting approach trains on one part of the year and evaluates on another, leading to a highly specific evaluation. This splitting method does not adequately represent the model's ability to produce accurate predictions across the entire year, which poses problem for potential real-world applications.

#### K.3.2 FREQUENCY ANALYSIS

The frequency analysis indicates that 10 channels exhibit a dominant yearly cycle (52696 time steps, approximately 365.94 days). The remaining channels show dominant cycles of one day (7 channels), half a month (1 channel), two months (1 channel), two and a half months (1channel), and four months (1 channel). The most prominent cycles in this dataset are one year, six months, four months, and one day.

| | Fundamental | 2nd | 3rd |
|---|---|---|---|
| p (mbar) | 10539.2 (73.19) | 8782.7 (60.99) | 4391.3 (30.50) |
| T (degC) | 52696.0 (365.94) | 144.0 (1.00) | 26348.0 (182.97) |
| Tpot (K) | 52696.0 (365.94) | 144.0 (1.00) | 26348.0 (182.97) |
| Tdew (degC) | 52696.0 (365.94) | 26348.0 (182.97) | 17565.3 (121.98) |
| rh (%) | 144.0 (1.00) | 52696.0 (365.94) | 17565.3 (121.98) |
| VPmax (mbar) | 52696.0 (365.94) | 144.0 (1.00) | 26348.0 (182.97) |
| VPact (mbar) | 52696.0 (365.94) | 26348.0 (182.97) | 10539.2 (73.19) |
| VPdef (mbar) | 52696.0 (365.94) | 144.0 (1.00) | 143.6 (1.00) |
| sh (g/kg) | 52696.0 (365.94) | 26348.0 (182.97) | 10539.2 (73.19) |
| $H_2OC$ ($\mu$mol/mol) | 52696.0 (365.94) | 26348.0 (182.97) | 10539.2 (73.19) |
| rho (g/m$^3$) | 52696.0 (365.94) | 144.0 (1.00) | 26348.0 (182.97) |
| wv (m/s) | 144.0 (1.00) | 143.6 (1.00) | 72.0 (0.50) |
| max. wv (m/s) | 144.0 (1.00) | 8782.7 (60.99) | 143.6 (1.00) |
| wd (deg) | 8782.7 (60.99) | 52696.0 (365.94) | 3293.5 (22.87) |
| rain (mm) | 2107.8 (14.64) | 17565.3 (121.98) | 258.3 (1.79) |
| raining (s) | 17565.3 (121.98) | 893.2 (6.20) | 958.1 (6.65) |
| SWDR (W/m$^2$) | 144.0 (1.00) | 52696.0 (365.94) | 143.6 (1.00) |
| PAR ($\mu$mol/m$^2$/s) | 144.0 (1.00) | 52696.0 (365.94) | 143.6 (1.00) |
| max. PAR ($\mu$mol/m$^2$/s) | 144.0 (1.00) | 52696.0 (365.94) | 143.6 (1.00) |
| Tlog (degC) | 52696.0 (365.94) | 144.0 (1.00) | 26348.0 (182.97) |
| $CO_2$ (ppm) | 144.0 (1.00) | 4053.5 (28.15) | 521.7 (3.62) |

Table 20: Weather from Autoformer - Frequency analysis. The first value is the period in number of time steps the value in parentheses is the equivalent in days.

### K.3.3 CORRELATION ANALYSIS

Figures 9 represents the channels correlation of the Weather dataset from Autoformer using the different methods mentioned in Appendix I.3. Across all metrics, significant seasonal differences are observed:

- *Winter* and *Spring* exhibit correlations that differ substantially from those of *Summer* and *Autumn*

- Some smaller differences are also observed between *Winter* and *Spring*, as well as between *Summer* and *Autumn*.

An efficient MTSF model must effectively capture these seasonal variations and adapt the dependencies based on the input season.

### K.3.4 DATA DISTRIBUTION ANALYSIS

Similar to LCD, most weather indicators demonstrate distinct seasonal distributions, with significant fluctuations for several channels.

Figure 10 provides two data distribution plots for the original dataset: one per season and one per data splitting set. As expected, channels with inconsistencies or where failure values have been identified appear anomalous. Similarly to LCD, most weather indicators demonstrate distinct seasonal distributions, with significant fluctuations for several channels.

The lack of data spanning multiple years prevent from using a cycle-inclusive splitting strategy with a one year dominant cycle. Instead, Wu et al. (2021) adopted a ratio splitting (*7:1:2 ∼ (8.4/1.2/2.4 months*). This approach implies that neither the validation nor the evaluation periods encompass a complete ***longest cycle***. Consequently, the training process is skewed to optimize performance for the selected validation period (i.e., *Autumn*), while the evaluation (i.e., *Winter*) fails to adequately test the model's ability to generalize across the full cycle. As demonstrated by the distribution and correlation analyses, notable differences exist between these periods. In addition, we observed in Figure 10 that the ratio splitting strategy implies significant distribution difference between training, validation and evaluation sets.

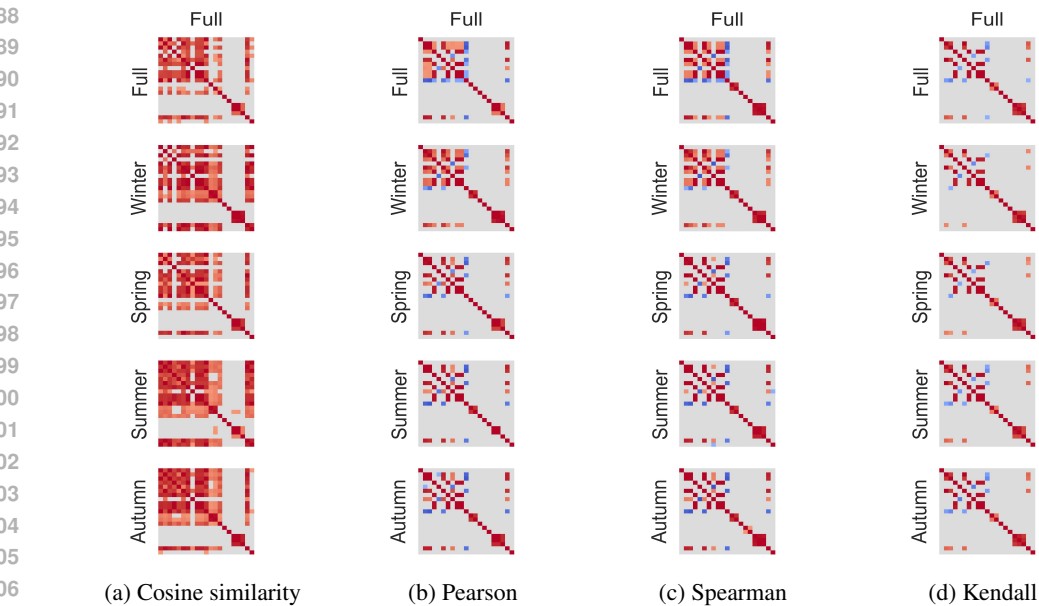

| (a) Cosine similarity | (b) Pearson | (c) Spearman | (d) Kendall |

Figure 9: Weather Dataset from Autoformer - Channels correlation for the full dataset and per season.

### K.3.5 INCONSISTENCIES PRESENTATION

We identified three types of inconsistencies in the MPI dataset:

1. **Failure Values** ($-9999$): Figure 8 shows instances where the value $-9999$ appears throughout the dataset, likely indicating measurement failures or missing observations due to instrument errors.

2. **Duplicated entries**: We found duplicated entries with identical timestamp and values across all variables.

3. **Missing Time Step**: Certain time steps are missing from the original data.

These inconsistencies can present in the file provided by Autoformer as well as the data archives on the original website, as detailed in Table 21.

|  | Duplicated | Missing |
|---|---|---|
| Autoformer | 1 | 9 |
| 2020a | 1 | 9 |
| 2020b | 0 | 0 |
| 2021a | 0 | 0 |
| 2021b | 0 | 0 |
| 2022a | 6 | 0 |
| 2022b | 1 | 82 |
| 2023a | 0 | 3 |
| 2023b | 143 | 0 |

Table 21: Count of duplicated entries and missing time steps found in the Autoformer CSV file and data archives from the Max-Planck-Institute original website.

### K.3.6 INCONSISTENCIES APPEARANCE

In Figure 8, colored vertical lines indicate time steps with inconsistencies. These errors occur only in the training period for the dataset introduced in the Autoformer paper.

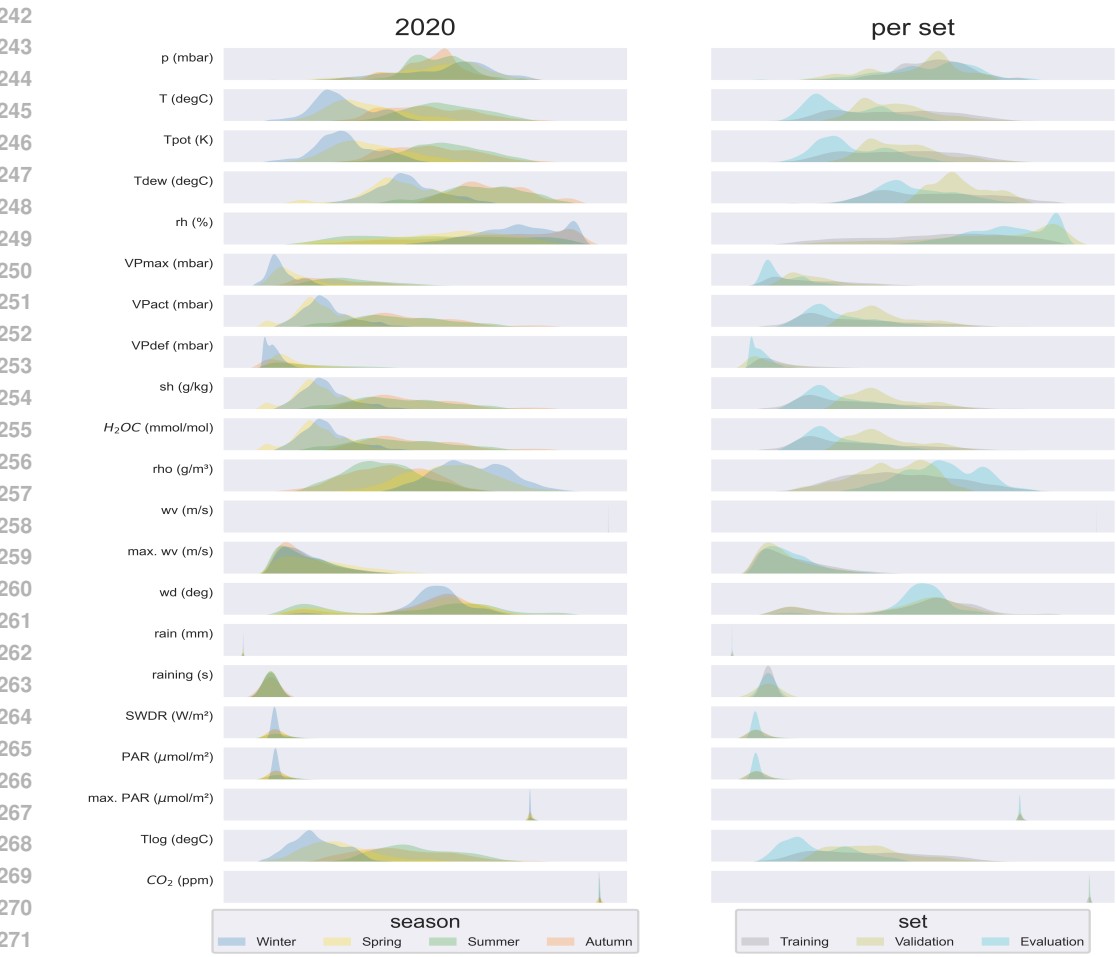

Figure 10: Weather Dataset from Autoformer - Distribution plots per channel. The last column illustrates data distribution with the ratio splitting strategy. The first column illustrates the data distribution for the whole datasets with a differentiation per season.

The **pink** vertical lines mark time steps where failure value appeared, while the **purple** lines denote missing time steps.

### K.4 PROPOSED CORRECTION

To address these errors (failure values and missing time steps), we propose the following correction process as described in the main paper: (i) replace erroneous values with NaN, (ii) apply linear interpolation for isolated errors, and (iii) for consecutive errors, use context-aware when possible or linear interpolation.

The corrected dataset is visualized in Figure 11.

#### K.4.1 IDENTIFY INCONSISTENCIES

Five additional columns have been added to the CSV file in order to identify the time steps where inconsistencies were corrected:

- $is\_wv\_value\_error$: flags time steps where a failure value arose in *Wind Velocity*.

- $is\_maxPAR\_value\_error$: marks time steps where a failure value occurred in the *Maximum Photosynthetic Active Radiation* variable.

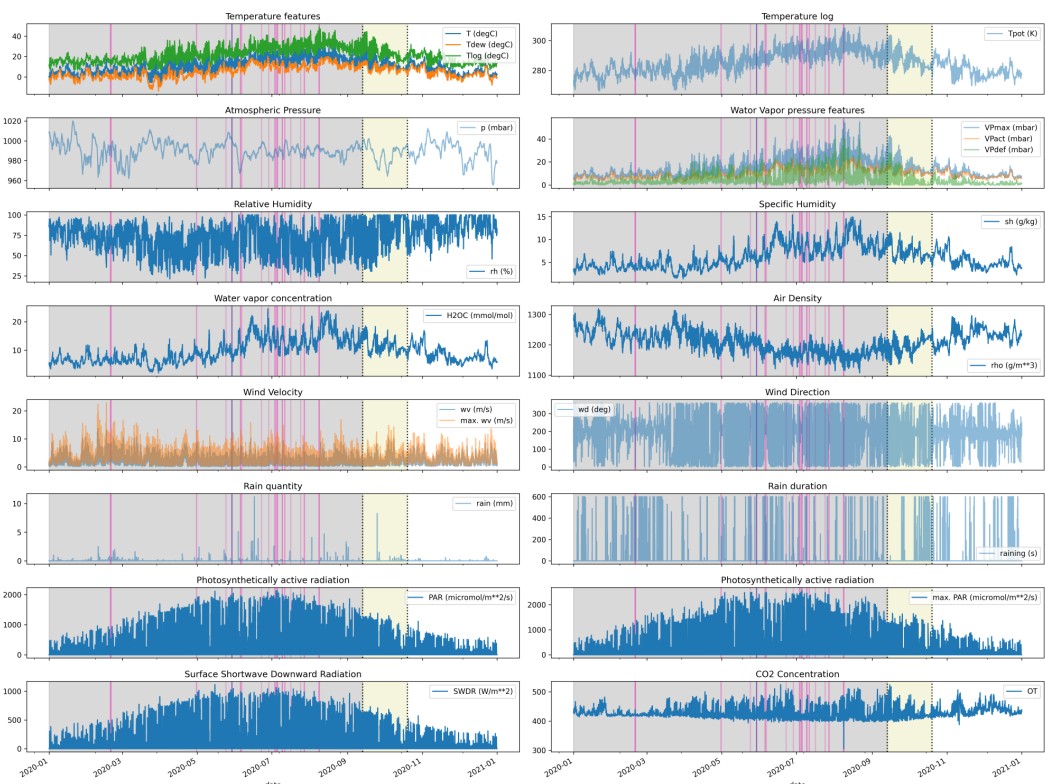

Figure 11: Overview of the weather indicators from the 1-year dataset used in Autoformer **after our correction process**. The gray [resp. yellow] background area denotes the training [resp. validation] period as defined in the ratio splitting. Colored vertical lines indicate time steps where inconsistencies were identified.

- $is\_OT\_value\_error$: identifies time steps where a failure value appeared in the *CO2 concentration* variable.

- $is\_ts\_missing$: indicates time steps that were missing in the original dataset.

- $is\_ts\_modified$: logs all time steps where corrections were applied.

### K.4.2 Overall Analysis

Figure 11 shows the plots of the different weather indicators from the corrected version. The gray area represents the training period, while the yellow area indicates the validation period as defined by the ratio splitting strategy. The corrections appear to have effectively addressed the errors and inconsistencies.

### K.4.3 Frequency Analysis

The frequency analysis of the revised dataset reveals slight differences from the original. While 10 channels still exhibit a dominant yearly frequency, the cycle now spans 52704 time steps (equivalent to 366 days), confirming that time steps were missing in the original dataset. The primary cycles in this dataset are now one year, six months, four months, and one day.

### K.4.4 Correlation Analysis

The correlation analysis for the corrected version closely resembles that of the original Autoformer dataset, with no significant deviations.

| | Fundamental | 2nd | 3rd |
|---|---|---|---|
| p (mbar) | 10540.8 (73.20) | 8784.0 (61.00) | 4392.0 (30.50) |
| T (degC) | 52704.0 (366.00) | 144.0 (1.00) | 26352.0 (183.00) |
| Tpot (K) | 52704.0 (366.00) | 144.0 (1.00) | 26352.0 (183.00) |
| Tdew (degC) | 52704.0 (366.00) | 26352.0 (183.00) | 17568.0 (122.00) |
| rh (%) | 144.0 (1.00) | 52704.0 (366.00) | 17568.0 (122.00) |
| VPmax (mbar) | 52704.0 (366.00) | 144.0 (1.00) | 26352.0 (183.00) |
| VPact (mbar) | 52704.0 (366.00) | 26352.0 (183.00) | 10540.8 (73.20) |
| VPdef (mbar) | 52704.0 (366.00) | 144.0 (1.00) | 143.6 (1.00) |
| sh (g/kg) | 52704.0 (366.00) | 26352.0 (183.00) | 10540.8 (73.20) |
| $H_2$OC ($\mu$mol/mol) | 52704.0 (366.00) | 26352.0 (183.00) | 10540.8 (73.20) |
| rho (g/m$^3$) | 52704.0 (366.00) | 144.0 (1.00) | 26352.0 (183.00) |
| wv (m/s) | 144.0 (1.00) | 52704.0 (366.00) | 8784.0 (61.00) |
| max. wv (m/s) | 144.0 (1.00) | 8784.0 (61.00) | 2773.9 (19.26) |
| wd (deg) | 8784.0 (61.00) | 52704.0 (366.00) | 3294.0 (22.88) |
| rain (mm) | 2108.2 (14.64) | 17568.0 (122.00) | 258.4 (1.79) |
| raining (s) | 17568.0 (122.00) | 893.3 (6.20) | 958.3 (6.65) |
| SWDR (W/m$^2$) | 144.0 (1.00) | 52704.0 (366.00) | 72.0 (0.50) |
| PAR ($\mu$mol/m$^2$/s) | 144.0 (1.00) | 52704.0 (366.00) | 72.0 (0.50) |
| max. PAR ($\mu$mol/m$^2$/s) | 144.0 (1.00) | 52704.0 (366.00) | 72.0 (0.50) |
| Tlog (degC) | 52704.0 (366.00) | 144.0 (1.00) | 26352.0 (183.00) |
| $CO_2$ (ppm) | 144.0 (1.00) | 144.4 (1.00) | 52704.0 (366.00) |

Table 22: **MPIW_10T_1Y_R** - Frequency analysis. The first value is the period in number of time steps the value in parentheses is the equivalent in days.

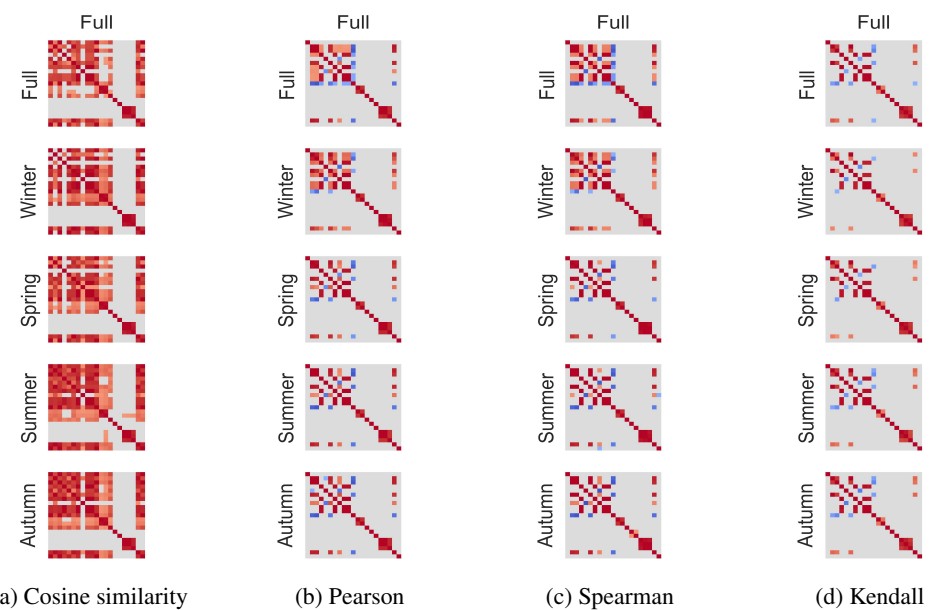

(a) Cosine similarity     (b) Pearson     (c) Spearman     (d) Kendall

Figure 12: **MPIW_10T_1Y_R** - Channels correlation for the full dataset and per season.

### K.4.5 DATA DISTRIBUTION ANALYSIS

Figure 13 provides two distribution plots for our corrected version **MPIW_10T_1Y_R**: one per season and one per splitting strategy set. As expected, the channel for which inconsistencies and especially failure values were uncovered now appears more consistent with the other channels. However, the distribution shift induced by the ratio splitting strategy persists.

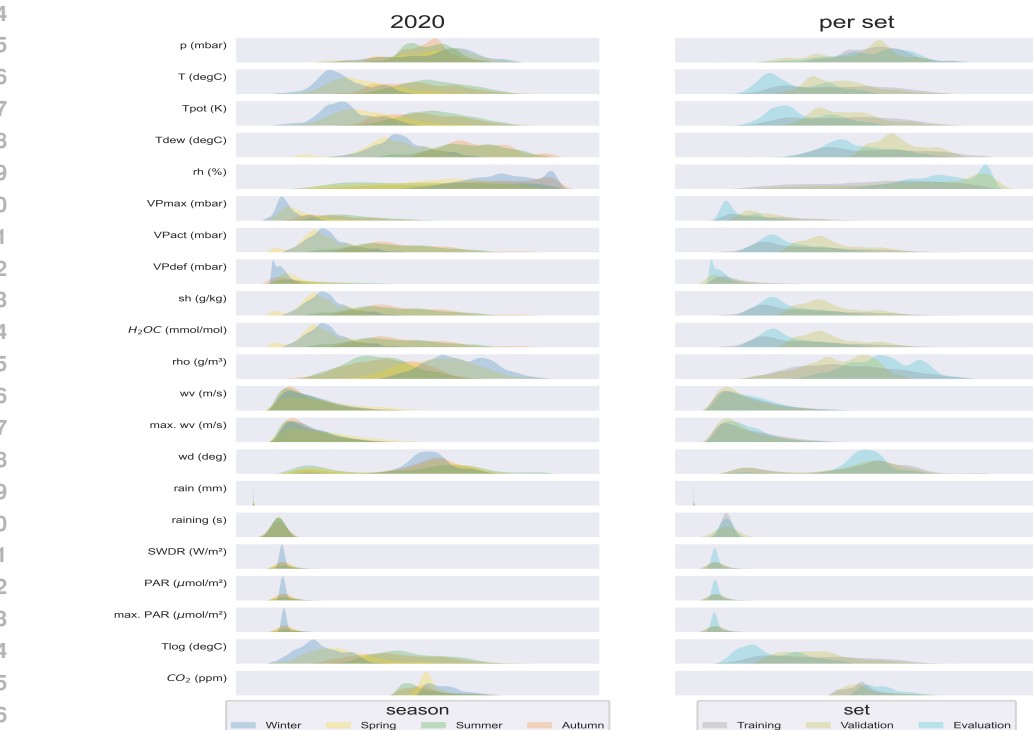

Figure 13: **MPIW_10T_1Y_R** - Distribution plots per channel. The last column illustrates data distribution with the ratio splitting strategy. The first column illustrates the data distribution for the whole datasets with a differentiation per season.

### K.5 EXTENDED VERSIONS

To investigate cycle-inclusive splits, we extended the dataset to cover a 4-year period spanning from "2020-01-01 00:10:00" to "2024-01-01 00:10:00" (included). We collected additional data from the corresponding website and applied our correction process. The corrected dataset is shown in Figure 14. As illustrated, errors primarily appeared in the training and validation periods. However, due to our correction process, their impact should be minimal.

#### K.5.1 OVERALL ANALYSIS

Figure 14 depicts the plots of the different weather indicators for the extended and corrected dataset. The gray area represents the training period, while the yellow area indicates the validation period as defined by the ratio splitting strategy. Errors and inconsistencies are no longer visible, suggesting that the corrections were applied successfully. This four-year dataset further confirms the presence of clear yearly cycles, as indicated by earlier analyses.

#### K.5.2 FREQUENCY ANALYSIS

The frequency analysis of the extended dataset reveals differences from the one-year datasets. Now, 12 channels have a dominant yearly frequency and 7 channels have a dominant daily frequency. The most common cycles are one year, six months, and one day. As a result, the longest dominant cycle across all channels remains one year. However, it is now possible to use a cycle-inclusive splitting strategy that covers at least one full year.

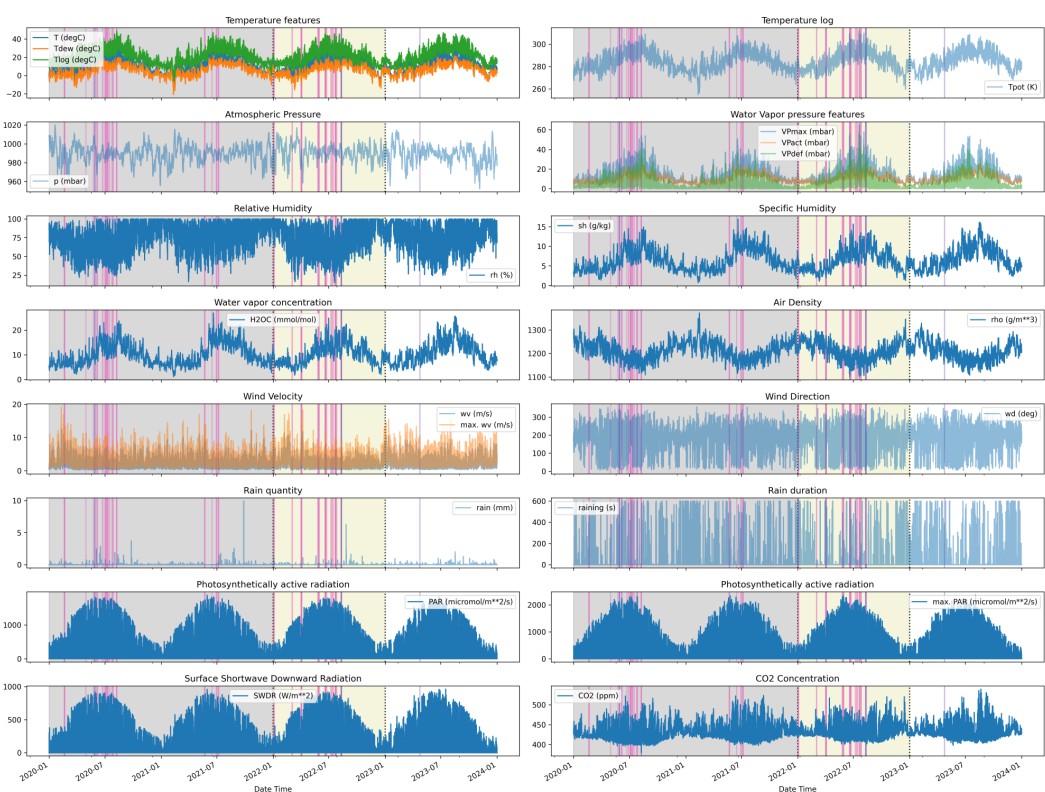

Figure 14: Overview of the weather indicators from our proposed 4-year dataset collected from MPI **after our correction process**. The gray background area represents the training period, while the yellow area denotes the validation period as defined in our proposed cycle-inclusive splitting. Colored vertical lines indicate time steps where inconsistencies were identified.

| | Fundamental | 2nd | 3rd |
|---|---|---|---|
| p (mbar) | 15027.4 (104.36) | 16183.4 (112.38) | 8091.7 (56.19) |
| T (degC) | 52596.0 (365.25) | 144.0 (1.00) | 26298.0 (182.63) |
| Tpot (K) | 52596.0 (365.25) | 144.0 (1.00) | 26298.0 (182.63) |
| Tdew (degC) | 52596.0 (365.25) | 8766.0 (60.88) | 26298.0 (182.63) |
| rh (%) | 144.0 (1.00) | 52596.0 (365.25) | 143.6 (1.00) |
| VPmax (mbar) | 52596.0 (365.25) | 144.0 (1.00) | 26298.0 (182.63) |
| VPact (mbar) | 52596.0 (365.25) | 26298.0 (182.63) | 8766.0 (60.88) |
| VPdef (mbar) | 52596.0 (365.25) | 144.0 (1.00) | 143.6 (1.00) |
| sh (g/kg) | 52596.0 (365.25) | 26298.0 (182.63) | 8766.0 (60.88) |
| $H_2OC$ ($\mu$mol/mol) | 52596.0 (365.25) | 26298.0 (182.63) | 8766.0 (60.88) |
| rho (g/m$^3$) | 52596.0 (365.25) | 144.0 (1.00) | 5686.1 (39.49) |
| wv (m/s) | 144.0 (1.00) | 52596.0 (365.25) | 9562.9 (66.41) |
| max. wv (m/s) | 144.0 (1.00) | 52596.0 (365.25) | 143.6 (1.00) |
| wd (deg) | 52596.0 (365.25) | 21038.4 (146.10) | 144.0 (1.00) |
| rain (mm) | 2805.1 (19.48) | 52596.0 (365.25) | 1290.7 (8.96) |
| raining (s) | 52596.0 (365.25) | 16183.4 (112.38) | 2390.7 (16.60) |
| SWDR (W/m$^2$) | 144.0 (1.00) | 52596.0 (365.25) | 143.6 (1.00) |
| PAR ($\mu$mol/m$^2$/s) | 144.0 (1.00) | 52596.0 (365.25) | 143.6 (1.00) |
| max. PAR ($\mu$mol/m$^2$/s) | 144.0 (1.00) | 52704.0 (366.00) | 72.0 (0.50) |
| Tlog (degC) | 52596.0 (365.25) | 144.0 (1.00) | 144.4 (1.00) |
| $CO_2$ (ppm) | 144.0 (1.00) | 144.4 (1.00) | 143.6 (1.00) |

Table 23: **MPIW_10T_4Y_R** - Frequency analysis. The first value is the period in number of time steps the value in parentheses is the equivalent in days.

### K.5.3  CORRELATION ANALYSIS

Figure 15 displays the channel correlations for the extended dataset **MPIW_10T_4Y_R** using the different methods mentioned in Appendix I.3. Similarly to LCD, for all metrics, the following patterns emerge:

1. By row: Year-to-year correlations remain consistent (with minimal variation);

2. By column: Within a given period, when divided by solar seasons, the correlations can vary significantly. For instance, *Winter* and *Spring* exhibit notable differences compared to *Summer* and *Autumn*. In addition, while differences between *Winter* and *Spring*, as well as *Summer* and *Autumn*, are less pronounced, they are still evident.

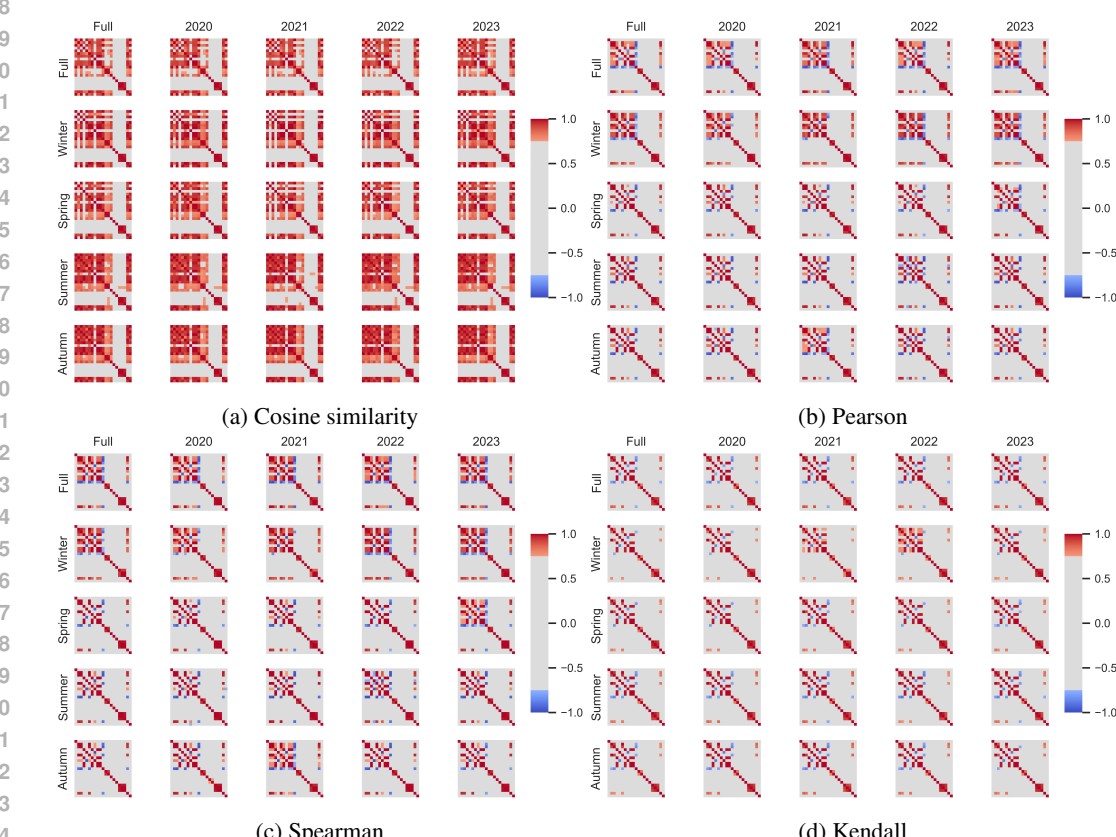

(a) Cosine similarity          (b) Pearson

(c) Spearman          (d) Kendall

Figure 15: **MPIW_10T_4Y_R** - Channels correlation for the full dataset, per year and per season.

### K.5.4 DATA DISTRIBUTION ANALYSIS

Figure 16 provides various distribution plots for the corrected four-year dataset: **MPIW_10T_4Y_R**. Some inter-annual variations are observed, such as differences in *Relative Humidity (rh)* densities between 2021 and 2022 compared to 2020 and 2024. Any efficient MTSF models should account for such variations in order to be considered robust.

In addition, we observed in Figure 16 that our cycle-inclusive splitting strategy significantly reduces distribution shift across sets, ensuring that model performances are evaluated over the longest cycle period.

### K.5.5 IDENTIFY INCONSISTENCIES:

Six additional columns have been appended to the produced CSV files in order to identify the time steps where inconsistencies were corrected:

- $is\_wv\_value\_error$: marks time steps where a failure value appeared in the *Wind Velocity* variable.

- $is\_SWDR\_value\_error$: highlights time steps where a failure value occurred in the *Surface Shortwave Downward Radiation* variable.

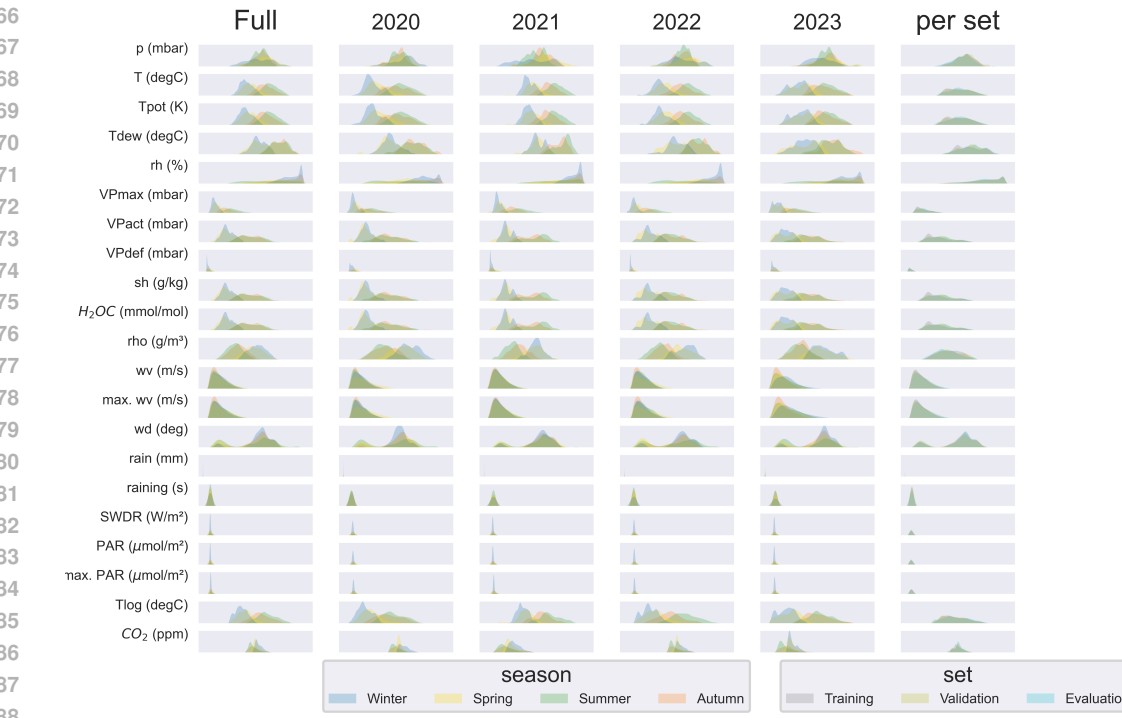

Figure 16: **MPIW_10T_4Y_R** - Distribution plots per channel. The last column illustrates data distribution with the ratio splitting strategy. The first column illustrates the data distribution for the whole datasets with a differentiation per season.

- $is\_maxPAR\_value\_error$: flags time steps where a failure value appeared in the *Maximum Photosynthetic Active Radiation* variable.
- $is\_CO2\_value\_error$: identifies time steps where a failure value occurred in the *CO2 concentration* variable.
- $is\_ts\_missing$: indicates time steps that were missing in the original dataset.
- $is\_ts\_modified$: logs all time steps where corrections were applied.

### K.5.6 HOURLY VERSION:

We propose an hourly version of this 4-year dataset by aggregating data over six consecutive time steps (i.e., from HH:10 to HH+1:00). The following aggregation functions are applied to the corresponding variables:

- **Sum**: *Precipitation Amount*, *Precipitation Duration*, *Surface Shortwave Downward Radiation*, *Photosynthetic Active Radiation* and columns identifying the errors.
- **Maximum**: *Maximum Photosynthetic Active Radiation* and *Maximum Wind Velocity*.
- **Mean**: All other variables.

**Frequency Analysis:** As shown in Table 24, the frequency analysis of the hourly dataset is similar to that of the 10-minute interval dataset.

**Correlation Analysis:** As shown in Figure 17, the correlation analysis of the hourly dataset is similar to that of the 10-minute interval dataset.

The data distribution of the hourly dataset being very similar to the 10-minute interval dataset, the corresponding plots have been omitted.

| | Fundamental | 2nd | 3rd |
|---|---|---|---|
| p (mbar) | 2504.6 (104.36) | 2697.2 (112.38) | 1348.6 (56.19) |
| T (degC) | 8766.0 (365.25) | 24.0 (1.00) | 4383.0 (182.62) |
| Tpot (K) | 8766.0 (365.25) | 24.0 (1.00) | 4383.0 (182.62) |
| Tdew (degC) | 8766.0 (365.25) | 1461.0 (60.88) | 4383.0 (182.62) |
| rh (%) | 24.0 (1.00) | 8766.0 (365.25) | 23.9 (1.00) |
| VPmax (mbar) | 8766.0 (365.25) | 24.0 (1.00) | 4383.0 (182.62) |
| VPact (mbar) | 8766.0 (365.25) | 4383.0 (182.62) | 1461.0 (60.88) |
| VPdef (mbar) | 8766.0 (365.25) | 24.0 (1.00) | 23.9 (1.00) |
| sh (g/kg) | 8766.0 (365.25) | 4383.0 (182.62) | 1461.0 (60.88) |
| $H_2OC$ ($\mu$mol/mol) | 8766.0 (365.25) | 4383.0 (182.62) | 1461.0 (60.88) |
| rho (g/m$^3$) | 8766.0 (365.25) | 24.0 (1.00) | 947.7 (39.49) |
| wv (m/s) | 24.0 (1.00) | 8766.0 (365.25) | 1593.8 (66.41) |
| max. wv (m/s) | 24.0 (1.00) | 8766.0 (365.25) | 23.9 (1.00) |
| wd (deg) | 8766.0 (365.25) | 3506.4 (146.10) | 24.0 (1.00) |
| rain (mm) | 467.5 (19.48) | 8766.0 (365.25) | 215.1 (8.96) |
| raining (s) | 8766.0 (365.25) | 2697.2 (112.38) | 398.5 (16.60) |
| SWDR (W/m$^2$) | 24.0 (1.00) | 8766.0 (365.25) | 23.9 (1.00) |
| PAR ($\mu$mol/m$^2$/s) | 24.0 (1.00) | 8766.0 (365.25) | 23.9 (1.00) |
| max. PAR ($\mu$mol/m$^2$/s) | 24.0 (1.00) | 8766.0 (365.25) | 23.9 (1.00) |
| Tlog (degC) | 8766.0 (365.25) | 24.0 (1.00) | 24.1 (1.00) |
| $CO_2$ (ppm) | 24.0 (1.00) | 24.1 (1.00) | 23.9 (1.00) |

Table 24: **MPIW_1H_4Y_R** - Frequency analysis. The first value is the period in number of time steps the value in parentheses is the equivalent in days.

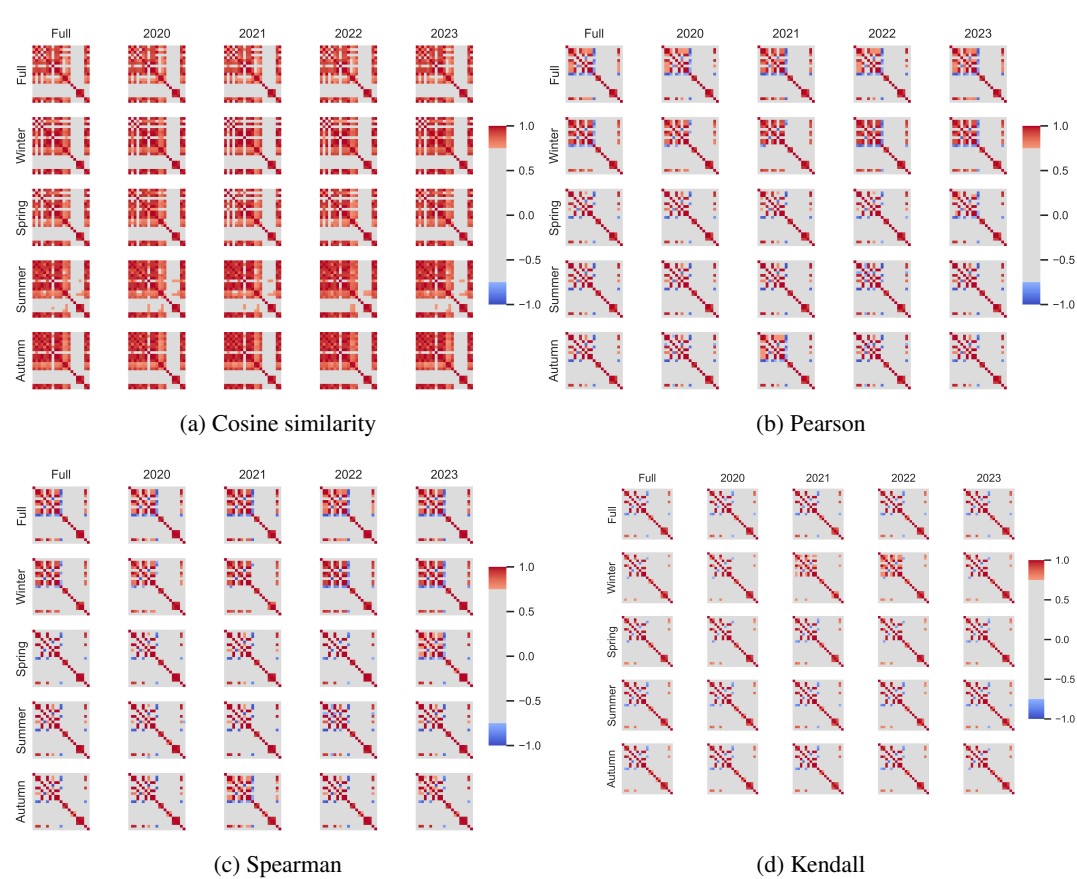

(a) Cosine similarity

(b) Pearson

(c) Spearman

(d) Kendall

Figure 17: **MPIW_1H_4Y_R** - Channels correlation for the full dataset, per year and per season.

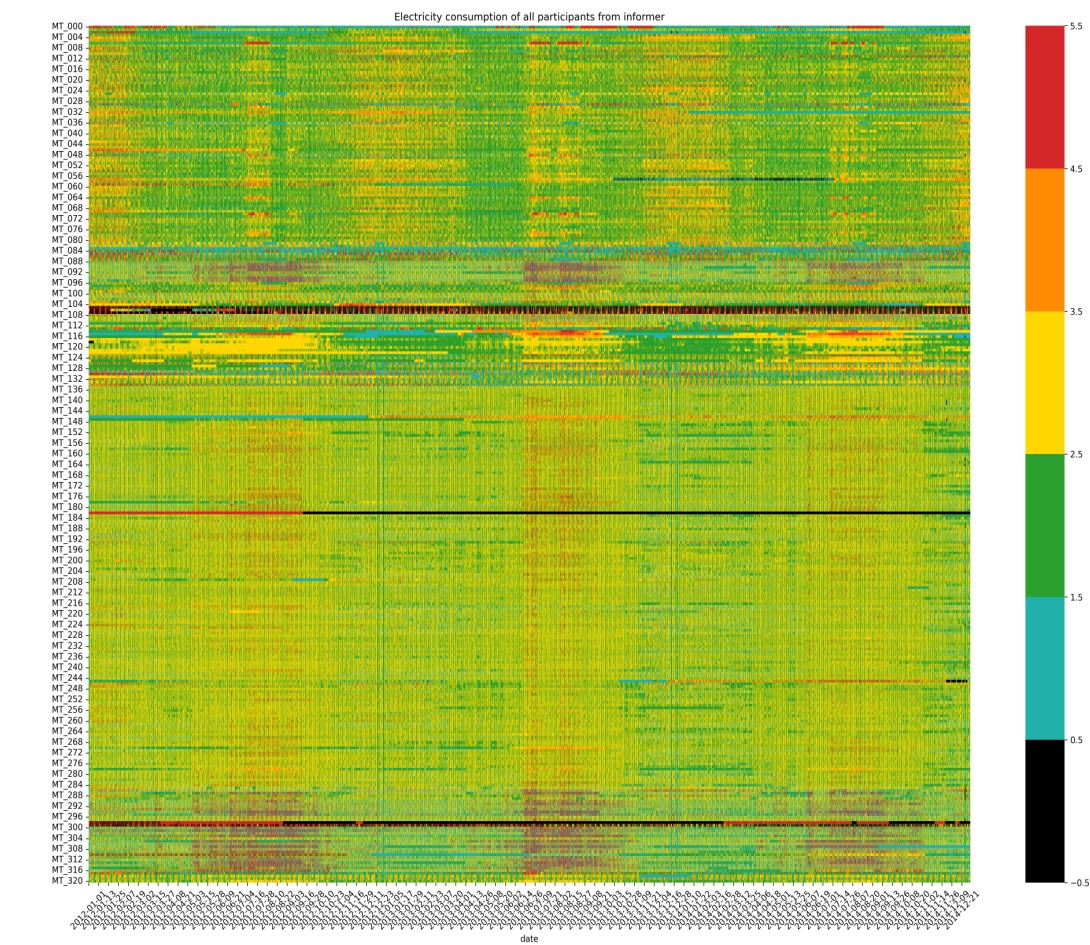

Figure 18: Overview of the normalized electricity consumption patterns of clients from the ECL dataset (derived from the UCI ELD dataset). The heatmap visualization simplifies the identification of inconsistent consumption patterns among clients.

## L   ELECTRICITY LOAD DIAGRAMS DATASET

### L.1   DESCRIPTION

The ELD [10] dataset consists of the electricity consumption data of 370 clients from what it appears to be a Portuguese electricity provider as timestamps report to Portuguese hours. Measurements were originally recorded every 15 minutes. The raw dataset covers the period from "2011-01-01 00:15:00" to "2015-01-01 00:00:00" (included). By aggregating four consecutive measurements (i.e., HH:15, HH:30, HH:45 and HH+1:00, an hourly version of the dataset can be obtained. Although the dataset description in UCI indicates having no missing data, some profiles depicted long and constant consumption equal to zero, as shown in the following sections, probably suggesting late arrival or early departure when occurring at the beginning or the end of the covered period, respectively.

### L.2   ANALYSIS

We consider the ELD dataset as spatiotemporal, where each channel represents the electricity consumption of clients across different locations in Portugal. These clients may belong to various categories such as *Residential*, *Commercial*, or *Industrial*, resulting in diverse consumption patterns

---

[10]https://archive.ics.uci.edu/dataset/321/electricityloaddiagrams20112014

and variation in volume, as evidenced in this document. While the dataset lacks specific information on the location and type of clients, it presents a rich tapestry of cycles closely tied to date, time, and human behavior. In addition, the variability in consumption patterns among clients poses a significant challenge for models, especially without external information, requiring them to decipher these underlying characteristics and correlations to accurately predict electricity consumption. Overall, predicting electricity consumption with this dataset presents a challenging task.

### L.3 ORIGINAL DATASET

ECL is an hourly dataset first introduced by (Li et al., 2019), derived from the ELD dataset available on UCI. This dataset provides electricity consumption data from 321 clients in Portugal, each identified as "MT_XXX", with 'XXX' representing a unique identifier.

All timestamps report to Portuguese hours. The dataset covers the period from "2012-01-01 00:00:00" to "2014-12-31 23:00:00" (included).

#### L.3.1 OVERALL ANALYSIS

Figure 18 plots the normalized consumption of the considered clients as a heatmap, aiding in the identification of distinctive patterns. These include clients with constant consumption values over time or those with unusual consumption patterns not typically observed in electricity usage. This figure reveals that most clients exhibit similar patterns, with noticeable summer peaks recurring annually in the bottom section of the figure. Conversely, clients in the upper section depict less pronounced peaks.

Notably, certain clients exhibit anomalies, such as the client displaying a continuous period of zero consumption (indicated by a black region).

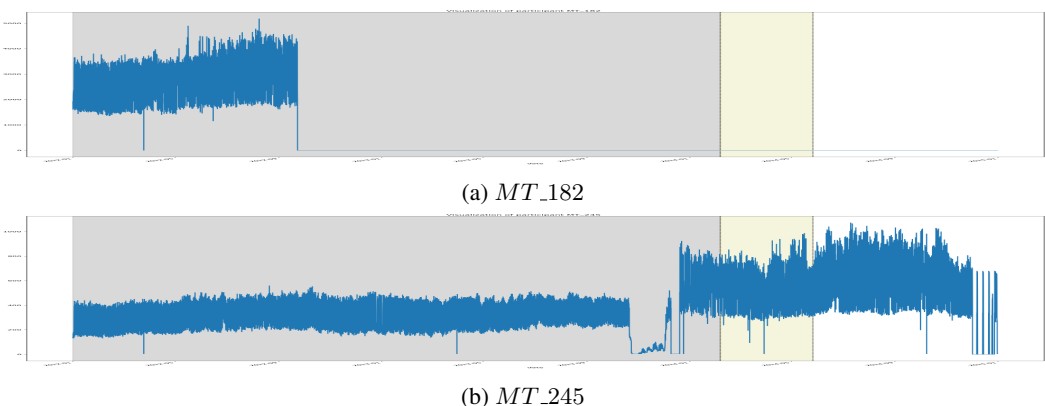

(a) $MT\_182$

(b) $MT\_245$

Figure 19: Overview of the electricity consumption profiles of two clients showing "early departure". The gray background area represents the training period, while the yellow area represents the validation period as defined in the ratio splitting. $MT\_245$ also exhibits sudden changes in consumption patterns.

#### L.3.2 INCONSISTENCIES PRESENTATION

In the following figures, the gray [resp. yellow] area represents the training [resp. validation] period as defined in the ratio splitting. In the raw UCI dataset, clients who began participating after the dataset's starting date showed constant consumption equal to zero before their participation started. These clients, that we refer to as "late arrival" clients, were removed in the ECL dataset version. However, as shown in Figure 19, two clients in the ECL dataset (particularly $MT\_182$) exhibit prolonged zero consumption after a certain date, suggesting an "early departure". We believe that these clients should have likely been removed as well to avoid impacting model evaluation in MTS forecasting.

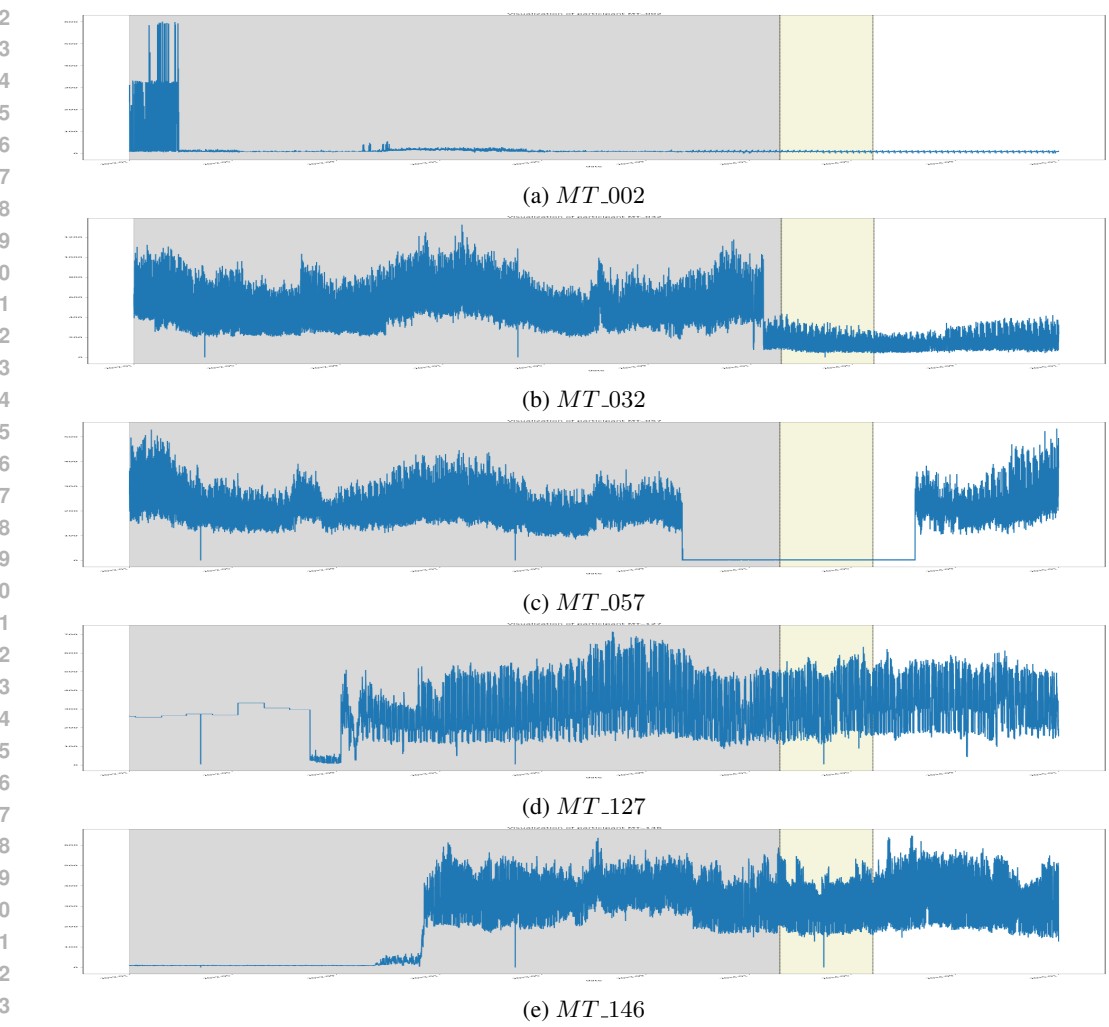

(a) $MT\_002$

(b) $MT\_032$

(c) $MT\_057$

(d) $MT\_127$

(e) $MT\_146$

Figure 20: Overview of the electricity consumption profiles of five clients displaying sudden changes in their overall patterns. The gray background area represents the training period, while the yellow area represents the validation period as defined in the ratio splitting. $MT\_127$ and $MT\_146$ also exhibit unusual consumption patterns at the beginning of the monitored period.

In addition, our analysis unveiled some clients with unusual and significant changes in their consumption patterns, such as the one shown in Figure 20. Without external information explaining such sudden changes, it becomes challenging for models to accurately learn consumption patterns and potential inter-variable relations.

Finally, similar to other datasets, we believe that the ratio splitting may not be optimal for conducting a fair comparison between models. This approach may favor models that perform well in the evaluation period but could potentially perform poorly elsewhere.

### L.3.3 Frequency Analysis

| | Fundamental | 2nd | 3rd |
|---|---|---|---|
| MT_001 | 12.0 (0.50) | 24.0 (1.00) | 8768.0 (365.33) |
| MT_002 | 26304.0 (1096.00) | 8768.0 (365.33) | 13152.0 (548.00) |
| MT_003 | 24.0 (1.00) | 12.0 (0.50) | 8768.0 (365.33) |
| MT_004 | 8768.0 (365.33) | 24.0 (1.00) | 4384.0 (182.67) |
| MT_005 | 24.0 (1.00) | 12.0 (0.50) | 8768.0 (365.33) |
| MT_006 | 4384.0 (182.67) | 8768.0 (365.33) | 2922.7 (121.78) |
| MT_007 | 24.0 (1.00) | 12.0 (0.50) | 8.0 (0.33) |
| MT_008 | 12.0 (0.50) | 8768.0 (365.33) | 24.0 (1.00) |
| MT_009 | 24.0 (1.00) | 8768.0 (365.33) | 84.0 (3.50) |
| MT_010 | 24.0 (1.00) | 8768.0 (365.33) | 12.0 (0.50) |
| MT_011 | 24.0 (1.00) | 167.5 (6.98) | 84.0 (3.50) |
| MT_012 | 24.0 (1.00) | 8768.0 (365.33) | 167.5 (6.98) |
| MT_013 | 24.0 (1.00) | 12.0 (0.50) | 4384.0 (182.67) |
| MT_014 | 24.0 (1.00) | 12.0 (0.50) | 8768.0 (365.33) |
| MT_015 | 24.0 (1.00) | 12.0 (0.50) | 4384.0 (182.67) |
| MT_016 | 24.0 (1.00) | 12.0 (0.50) | 4384.0 (182.67) |
| MT_017 | 24.0 (1.00) | 12.0 (0.50) | 8768.0 (365.33) |
| MT_018 | 24.0 (1.00) | 12.0 (0.50) | 8768.0 (365.33) |
| MT_019 | 24.0 (1.00) | 12.0 (0.50) | 8.0 (0.33) |
| MT_020 | 12.0 (0.50) | 24.0 (1.00) | 6.0 (0.25) |
| MT_021 | 12.0 (0.50) | 24.0 (1.00) | 8768.0 (365.33) |
| MT_022 | 24.0 (1.00) | 12.0 (0.50) | 8768.0 (365.33) |
| MT_023 | 24.0 (1.00) | 8768.0 (365.33) | 12.0 (0.50) |
| MT_024 | 24.0 (1.00) | 8768.0 (365.33) | 12.0 (0.50) |
| MT_025 | 167.5 (6.98) | 84.0 (3.50) | 168.6 (7.03) |
| MT_026 | 24.0 (1.00) | 12.0 (0.50) | 8768.0 (365.33) |
| MT_027 | 24.0 (1.00) | 12.0 (0.50) | 8.0 (0.33) |
| MT_028 | 12.0 (0.50) | 24.0 (1.00) | 4384.0 (182.67) |

Table 25: ECL Dataset - Frequency analysis of the first channels. The first value is the period in number of time steps the value in parentheses is the equivalent in days.

From Table 25 we can observed that some channels have their dominant periods significantly over one year. But the majority exhibit a longest cycle of one year.

### L.3.4 DATA DISTRIBUTION ANALYSIS

Figure 21 provides the distribution plots for the *ECL* dataset, revealing that channels are sensitive to seasonal variations.

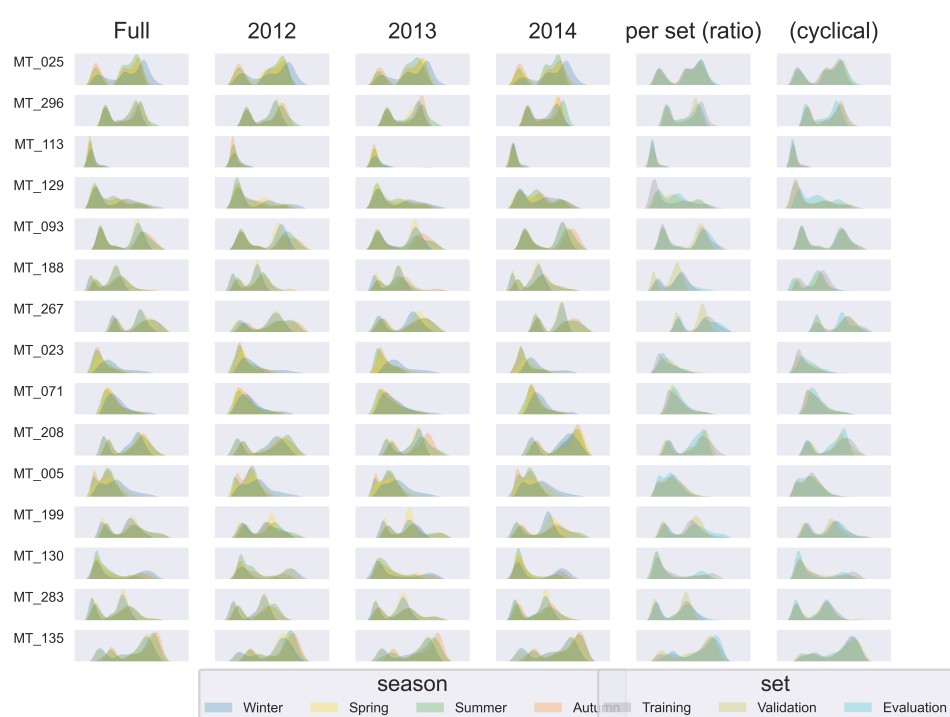

Figure 21: ECL Dataset - Distribution plots per channel. The last two columns illustrate data distribution per splitting strategy: ratio and our proposal cycle-inclusive. The other columns illustrate the data distribution for the whole datasets and per year, with a differentiation per season.

### L.4 PROPOSED CORRECTION

Based on our observations, we propose removing the following 13 clients from the ECL dataset:

- **Early departure**: $MT\_182$ and $MT\_245$
- **Significant changes in consumption patterns**: $MT\_032$, $MT\_057$, $MT\_127$, $MT\_146$ and $MT\_307$
- **No clear cyclical patterns**: $MT\_002$, $MT\_106$, $MT\_114$, $MT\_122$, $MT\_298$ and $MT\_310$

The overall visualization of our proposed dataset is depicted in Figure 22.

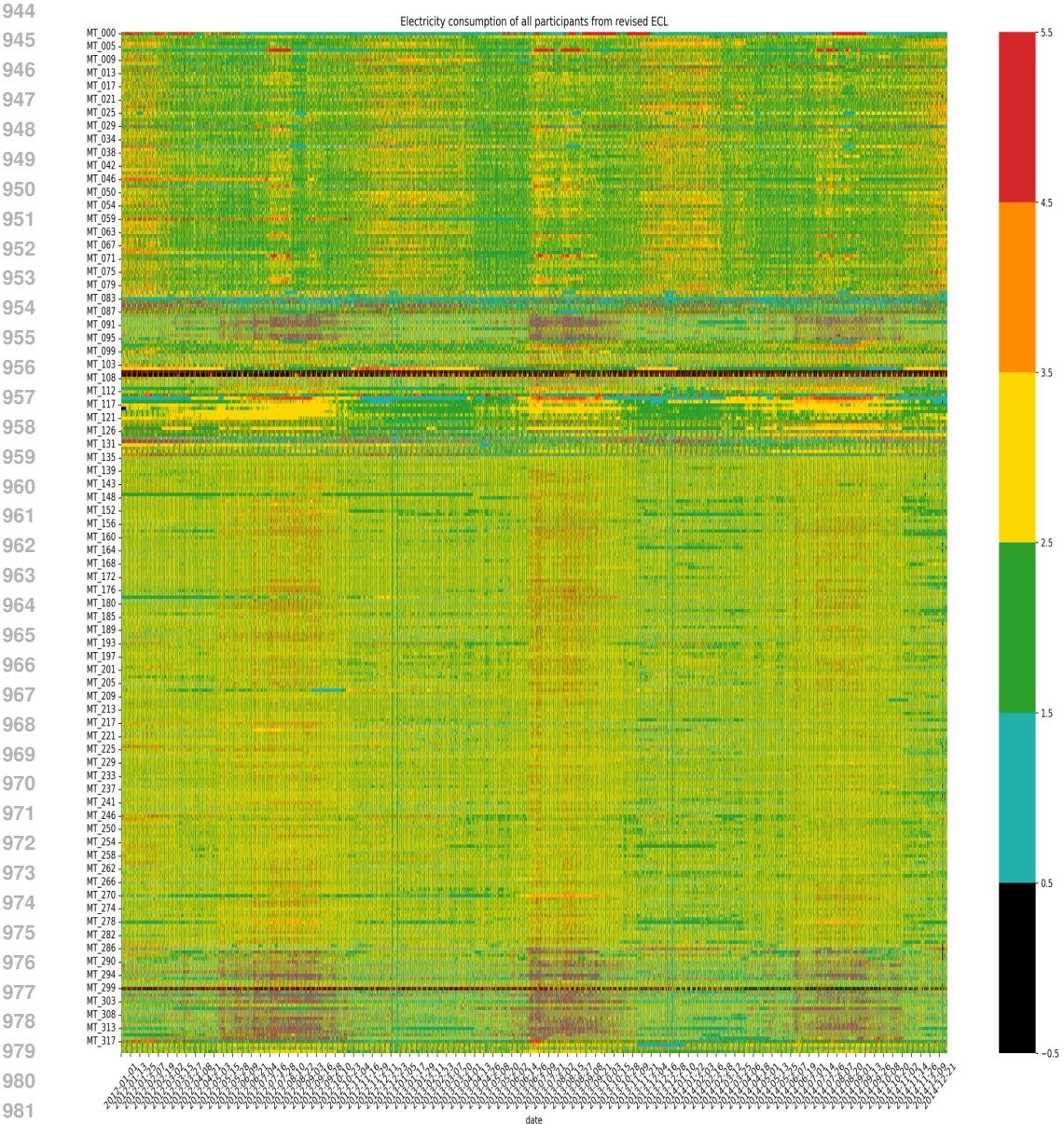

Figure 22: Overview of the normalized electricity consumption patterns of clients from our revised version of ECL dataset. The heatmap visualization simplifies the identification of inconsistent consumption patterns among clients.

### L.4.1 FREQUENCY ANALYSIS

From Table 26 we can observed that some channels have their dominant periods significantly over one year, but also significantly less than with the ECL dataset. But the majority exhibit a longest cycle of one year.

|          | Fundamental       | 2$^{nd}$          | 3$^{rd}$          |
| -------- | ----------------- | ----------------- | ----------------- |
| MT_000   | 13152.0 (548.00)  | 6576.0 (274.00)   | 3757.7 (156.57)   |
| MT_001   | 12.0 (0.50)       | 24.0 (1.00)       | 8768.0 (365.33)   |
| MT_003   | 24.0 (1.00)       | 12.0 (0.50)       | 8768.0 (365.33)   |
| MT_004   | 8768.0 (365.33)   | 24.0 (1.00)       | 4384.0 (182.67)   |
| MT_005   | 24.0 (1.00)       | 12.0 (0.50)       | 8768.0 (365.33)   |
| MT_006   | 4384.0 (182.67)   | 8768.0 (365.33)   | 2922.7 (121.78)   |
| MT_007   | 24.0 (1.00)       | 12.0 (0.50)       | 8.0 (0.33)        |
| MT_008   | 12.0 (0.50)       | 8768.0 (365.33)   | 24.0 (1.00)       |
| MT_009   | 24.0 (1.00)       | 8768.0 (365.33)   | 84.0 (3.50)       |
| MT_010   | 24.0 (1.00)       | 8768.0 (365.33)   | 12.0 (0.50)       |
| MT_011   | 24.0 (1.00)       | 167.5 (6.98)      | 84.0 (3.50)       |
| MT_012   | 24.0 (1.00)       | 8768.0 (365.33)   | 167.5 (6.98)      |
| MT_013   | 24.0 (1.00)       | 12.0 (0.50)       | 4384.0 (182.67)   |
| MT_014   | 24.0 (1.00)       | 12.0 (0.50)       | 8768.0 (365.33)   |
| MT_015   | 24.0 (1.00)       | 12.0 (0.50)       | 4384.0 (182.67)   |
| MT_016   | 24.0 (1.00)       | 12.0 (0.50)       | 4384.0 (182.67)   |
| MT_017   | 24.0 (1.00)       | 12.0 (0.50)       | 8768.0 (365.33)   |
| MT_018   | 24.0 (1.00)       | 12.0 (0.50)       | 8768.0 (365.33)   |
| MT_019   | 24.0 (1.00)       | 12.0 (0.50)       | 8.0 (0.33)        |
| MT_020   | 12.0 (0.50)       | 24.0 (1.00)       | 6.0 (0.25)        |
| MT_021   | 12.0 (0.50)       | 24.0 (1.00)       | 8768.0 (365.33)   |
| MT_022   | 24.0 (1.00)       | 12.0 (0.50)       | 8768.0 (365.33)   |
| MT_023   | 24.0 (1.00)       | 8768.0 (365.33)   | 12.0 (0.50)       |
| MT_024   | 24.0 (1.00)       | 8768.0 (365.33)   | 12.0 (0.50)       |
| MT_025   | 167.5 (6.98)      | 84.0 (3.50)       | 168.6 (7.03)      |
| MT_026   | 24.0 (1.00)       | 12.0 (0.50)       | 8768.0 (365.33)   |
| MT_027   | 24.0 (1.00)       | 12.0 (0.50)       | 8.0 (0.33)        |
| MT_028   | 12.0 (0.50)       | 24.0 (1.00)       | 4384.0 (182.67)   |

Table 26: **PELD_1H_3Y_308** - Frequency analysis of the first channels. The first value is the period in number of time steps the value in parentheses is the equivalent in days.

### L.4.2    DATA DISTRIBUTION ANALYSIS

Figure 23 provides the same distribution plots for the revised dataset: **PELD_1H_3Y_308**. The modified inconsistencies and errors did not altered the properties of the datasets. Data distribution vary significantly per season, but our cycle-inclusive strategy ensure better distribution similarity between sets, making such dataset more suitable for benchmarking.

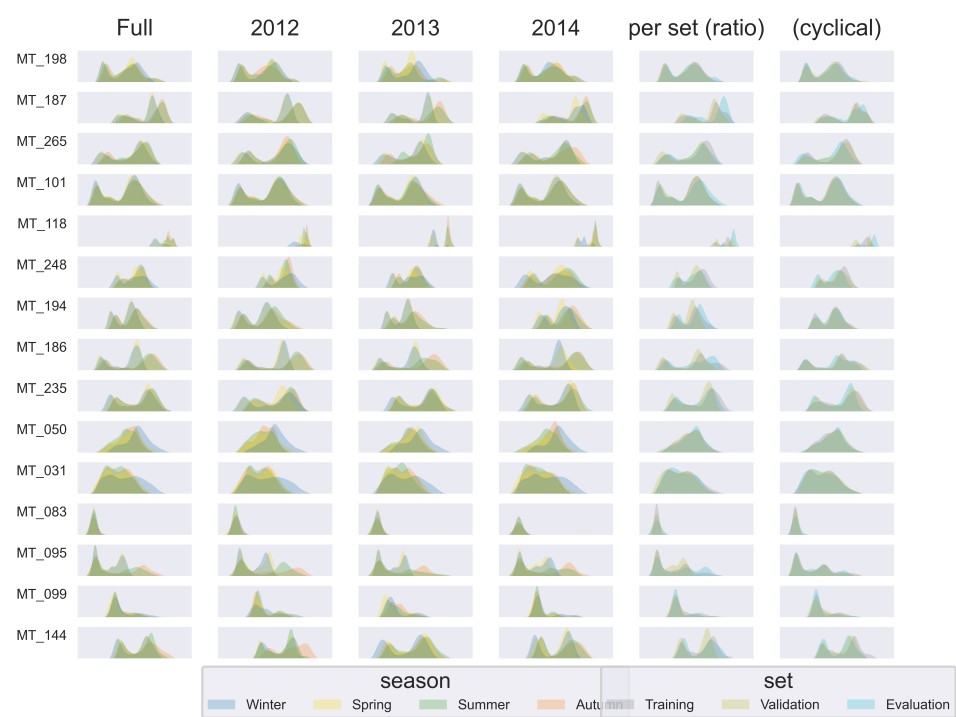

Figure 23: **PELD_1H_3Y_308** - Distribution plots per channel. The last two columns illustrate data distribution per splitting strategy: ratio and our proposal cycle-inclusive. The other columns illustrate the data distribution for the whole datasets and per year, with a differentiation per season.

### L.5 FUTURE VERSION

In the future, it may be necessary to remove or better identify clients exhibiting "short" periods of unusual consumption patterns or specific trends (either upward or downward consumption trends over years). This approach would allow for the segmentation of typical metrics (MAE, MSE, etc.) into three categories: an overall metric, metrics for clients with "usual" cyclical patterns, and metrics specifically for clients with these specific characteristics. Such a categorization would provide a clearer understanding of model performance and enable researchers to refine architectures more effectively.