# OpenReview forum: "Ensuring Fair Comparisons in Time Series Forecasting: Addressing Quality Issues in Three Benchmark Datasets"
_ICLR.cc/2025/Conference — Submitted to ICLR 2025_

### Official Review · Reviewer_FoY1 · 2024-10-18

**Soundness:** 1
**Presentation:** 3
**Contribution:** 2
**Rating:** 3
**Confidence:** 5

**Summary:**

The paper advocates for standardizing multivariate time-series forecasting evaluation, by removing inconsistencies and missing values, and by splitting data with cycle-preserving ratios. It further manually analyzes and releases cleaned versions of three commonly used datasets, and compares three recent models (NLinear, Informer and iTransformer) on them.

**Strengths:**

The paper tackles an **important problem**, the problem of ensuring a reliable, reproducible and consistent evaluation of time-series models.
The contributions are clearly stated, and I agree that the time-series community would benefit from standardized pre-processing and evaluation.

**Weaknesses:**

**The paper falls short in its analysis** of the impact of lack of standardization, for the following reasons:
1. It considers only three datasets, while evaluations are commonly carried out with many more datasets, as listed in Table 1 in the supplementary material. The same analysis and proprocessing should be carried out at least on the 10 most commonly used dataset for the work to have a real impact.
2. The paper’s evaluation of Section 6 does not highlight any worrying problems with using the publicly available versions of the datasets. For instance, model rankings do not change when comparing models on the old dataset versions and when comparing them on the newly proposed versions, and in general model's performance is not significantly impacted by the standardization. More strikingly, models look equally impacted by the changes which hinders the claim that current evaluation practices are not fair.
3. The paper’s evaluation of Section 6 does not provide new insights into model performance, e.g., their weaknesses and strengths wrt capturing cycles, multivariate relationships, robustness to noise, and other factors that can impact performance.
4. Only three and very recent models are considered, while there exists an extensive literature on the time-series forecasting, see e.g., survey [1].

**The proposed methodology is not original**, as it encompasses commonly used techniques, such as interpolation for missing value imputation and spectral analysis for cycle detection.
It is also heavily relies on expert knowledge and visual inspection, which hinders its applicability to new datasets.

**The work is not well positioned with respect to existing literature on standardizing time-series forecasting evaluation**. There exist at least two works, and related libraries [3][4], that provide access to standardized model implementations and datasets to facilitate a reliable, reproducible and consistent evaluation. How does the paper differ and improve upon these works?


[1] Dama, Fatoumata, and Christine Sinoquet. "Time series analysis and modeling to forecast: A survey." arXiv preprint arXiv:2104.00164 (2021).
[2] Wang, Yuxuan, et al. "Deep time series models: A comprehensive survey and benchmark." arXiv preprint arXiv:2407.13278 (2024).
[3] Alexandrov, Alexander, et al. "Gluonts: Probabilistic and neural time series modeling in python." Journal of Machine Learning Research 21.116 (2020).

**Questions:**

1. Two cleaned versions of the datasets are proposed. Which ones should the community use? In the effort of standardizing evaluation, it is confusing to release more than a single version.
2. Cycle-preserving splits are motivated to ensure train, validation and test sets are representative of the data distribution. This however does not account for shifts, which also create discrepancies. How should they be handled?
3. Releasing a library is a convenient way of ensuring that pre-processing and evaluation are automatically standardized. Is there a particular reason why the authors decided not to provide one?

---

> ### Author Response · Authors · 2024-11-25
> **Rebuttal to review by Reviewer FoY1**
>
> We would like to thank _Reviewer FoY1_ for the thorough review and valuable feedback on our submission.
> Below, we provide detailed responses to the questions and the limitations raised.
> Although the discussion phase is nearing completion, we welcome further feedback on our rebuttal.
>
> ## W1
> > The same analysis and proprocessing should be carried out at least on the 10 most commonly used dataset for the work to have a real impact.
>
> We agree that extending this work to additional datasets is essential, as mentioned in the paper.
> However, the extensive effort required—particularly for datasets with more than $30$ channels—poses a significant challenge for a single study.
> Our goal is to raise awareness of these issues, foster discussions, and encourage joint efforts from the research community.
> Contributions from others would provide diverse perspectives, enhancing the final outputs.
> Expanding the scope to additional datasets would also increase the complexity of the analysis (producing results table for each dataset before/after correction and with/without cycle-inclusive splitting), delaying the awareness and potentially reducing chances to be published in a conference.
>
> ## W2
> > which hinders the claim that current evaluation practices are not fair.
>
> We encourage the _Reviewer FoY1_ to refer to Table 16 in the Supplementary Material (and also provided in a separated comment above), which demonstrates a significant impact of dataset inconsistencies on evaluation metrics.
> Metrics directly influence rankings, which underscores the importance of addressing these issues.
> In addition, our detailed dataset analyses aim to help the community better understand and mitigate specificities, enabling the development of more appropriate solutions.
>
> ## W3
> > The paper’s evaluation of Section 6 does not provide new insights into model performance, e.g., their weaknesses and strengths wrt capturing cycles, multivariate relationships, robustness to noise, and other factors that can impact performance.
>
> We respectfully disagree.
> Tables 1 and 2 in the paper show that applying cycle-inclusive splitting to the **LCDWf_1H_4Y_USUNK** dataset significantly deteriorates the performance of all models.
> This highlights that models do not perform consistently across longer cycles.
> Moreover, the cycle-inclusive evaluation slightly challenges the superiority of recent models like iTransformer, suggesting that it may not be the best model for all prediction horizons.
>
>
> ## W4
> > Only three and very recent models are considered, while there exists an extensive literature on the time-series forecasting, see e.g., survey [1].
>
> We acknowledge the broader scope of TSF literature and understand that our model and dataset selection is limited.
> However, our focus was to raise awareness about dataset inconsistencies and splitting issues, rather than exhaustively benchmarking models.
> By concentrating on a limited set of datasets and models, we aim to encourage discussion at the conference and inspire collaborative efforts to extend this work.
>
> ## W5
> > The proposed methodology is not original, as it encompasses commonly used techniques, such as interpolation for missing value imputation and spectral analysis for cycle detection. It is also heavily relies on expert knowledge and visual inspection, which hinders its applicability to new datasets.
>
> As stated in the main rebuttal, a universal solution for identifying errors and inconsistencies across all TSF domains is unlikely.
> Each domain has unique characteristics—such as inter-channel relationships and value ranges—that require domain-specific knowledge.
> For example, imputing missing values with 0 (a common practice) can lead to contradictions, such as relative humidity being 0% on Earth or simultaneous zero readings in Fahrenheit and Celsius.
>
> While we do not claim to introduce a novel methodology, our goal is to provide transparency and a starting point for further refinement.
> Our corrections are intentionally basic to facilitate reproducibility and foster dialogue.
> We plan to extend this work (on both datasets and techniques), but we hope this paper will inspire future research to develop more sophisticated and generalizable solutions or join us in this task.
>
> ## W6
> > How does the paper differ and improve upon these works?
>
> May we ask the _Reviewer FoY1_ to clarify reference [4]?
> Regarding GluonTS, our work specifically addresses underlying errors, inconsistencies, and splitting issues, which are not tackled in [3].
> According to previous tests, while GluonTS ensures consistent data loading across models, it does not incorporate pre-processing to correct errors.
> Moreover, to the best of our knowledge, no prior work has investigated the impact of different splitting strategies on evaluation outcomes.

---

> > ### Comment · Reviewer_FoY1 · 2024-11-27
> > **positioning with respect to literature**
> >
> > In my review, I wrongly indexed the related works. "There exist at least two works, and related libraries [3][4]" was supposed to be "There exist at least two works, and related libraries [2][3]".
> >
> > Please clarify how your work positions with respect to this literature, that is the most related to your work, and provide evidence in the paper that these existing and commonly used libraries' "... effectiveness can be compromised if the underlying datasets contain errors or inconsistencies" (from the revised paper). Evidence could be, but is not limited to, changes in model rankings and significant degradation/improvement in model performance when using the datasets from these libraries wrt when using your dataset versions.

---

> > > ### Author Response · Authors · 2024-11-29
> > >
> > > Thank you for your feedback and for clarifying the indexing of your related works.
> > >
> > > > Please clarify how your work positions with respect to this literature, that is the most related to your work, and provide evidence in the paper that these existing and commonly used libraries' "... effectiveness can be compromised if the underlying datasets contain errors or inconsistencies" (from the revised paper). Evidence could be, but is not limited to, changes in model rankings and significant degradation/improvement in model performance when using the datasets from these libraries wrt when using your dataset versions.
> > >
> > > ## Regarding survey and benchmark paper
> > >
> > > To the best of our knowledge, survey and benchmark papers such as the one mentioned by _Reviewer FoY1_ primarily focus on model architectures (e.g., MLPs, CNNs, RNNs, Transformers) across various time series tasks, including imputation, short-term forecasting, long-term forecasting, and classification. In most cases, the datasets used for each task vary (e.g., reference [2]). However, our work highlights a critical gap: datasets specifically used for time series forecasting (TSF) and multivariate TSF (MTSF) often contain errors and inconsistencies, which can compromise the fairness and reliability of benchmarks for these architectures.
> > >
> > > We have revised the paper to make this point more explicit.
> > >
> > > ## Regarding existing time series libraries
> > >
> > > **GluonTS**: According to its Git repository and documentation, the library's preprocessing focuses on ensuring the dataset has the expected resolution (via the $resample()$ function) and filling missing time steps with a fixed, user-specified value ("desired value"). While this accounts for missing time steps, it is limited in its approach. For multivariate data, filling all missing values with the same fixed value introduces inconsistencies across channels. In contrast, our use of linear interpolation, though simple, maintains relation across channels. Furthermore, GluonTS does not address inherent errors or inconsistencies in the datasets, as highlighted in our work (and there are not mentioned of such processing in their documentation or code).
> > >
> > > **TFB** (suggested by _Reviewer D6qR_): According to the pipeline description, TFB includes steps for loading data, building models, evaluating models, and generating reports. The data_processing method primarily separates datasets into training and evaluation sets. Like GluonTS, there is no indication that TFB performs preprocessing to account for inherent errors or inconsistencies in datasets. This limitation is not explicitly addressed in their documentation or code.
> > >
> > > In light of these points, we argue that our corrected versions of the datasets are both legitimate and necessary. Additionally, libraries like GluonTS and TFB could benefit significantly from integrating our corrected datasets. We have revised the paper to clearly highlight the limitations of GluonTS and TFB regarding their handling of errors and inconsistencies in current TSF datasets.
> > >
> > > ## Regarding evidence
> > >
> > > As a consequence, under identical experimental settings, our results on the original datasets align with those obtained using these libraries. However, our corrected datasets yield the following insights:
> > >  * Significant impact of failure values in the MPI dataset, substantially affecting evaluation metrics.
> > >  * Model ranking changes:
> > >    * DLinear outperforms Autoformer, for univariate-to-univariate forecasting, on our corrected MPI dataset, whereas this was not the case with the original dataset.
> > >    * Informer outperforms iTransformer, a recognized SOTA model for MTSF (cf. reference [2]), with the cycle-inclusive splitting.
> > >
> > > These observations suggest that the best-performing model may depend on the evaluation period, the quality and nature of the dataset.
> > >
> > > In our humble opinion, such findings underscore the critical importance of creating standardized/cleaned datasets to enable fairer benchmarking. While our proposed correction methods are relatively basic, they highlight the need for researchers to develop more robust solutions for errors and inconsistencies identification and correction or to create new datasets free from such errors and inconsistencies (which will require time and efforts).
> > >
> > > Furthermore, developing a universal approach for identifying and correcting errors or inconsistencies (beyond anomalies) remains a highly challenging task, even with the power of large language models (LLMs). For context, zero-shot anomaly detection--a related but distinct problem--remains challenging even for LLMs, as demonstrated in the MIT study [4], which shows that domain-specific knowledge was necessary to reach satisfactory performance.
> > >
> > > Thank you once again for your valuable feedback. We sincerely hope that our revisions and clarifications address your concerns, and we remain open to further discussion to ensure that our submission is both clear and impactful.
> > >
> > > [4] https://arxiv.org/pdf/2405.14755

---

> ### Author Response · Authors · 2024-11-25
> **Rebuttal to review by Reviewer FoY1 (part 2)**
>
> ## Q1
> > In the effort of standardizing evaluation, it is confusing to release more than a single version.
>
> If this comment refers to LCD, as described in the Supplementary Material, the dataset includes both integer (input) and float (target) temperature data.
> To accommodate various user needs, especially for univariate scenarios, we provided two variants.
> As long as users clearly specify the version they employ, multiple versions should not cause confusion.
> In fact, when considering proper naming of different dataset variants (unlike the current situation), any benchmark performed on one variant in one paper cannot be compared to results from another paper using a different variant.
>
> ## Q2
> > This however does not account for shifts, which also create discrepancies. How should they be handled?
>
> In our opinion, handling shifts should be addressed within the model architecture, as it depends on the experimental setup (e.g., input length).
> Techniques such as RevIN, widely used in recent models, are well-suited for this purpose.
> While chronological data splitting aligns with TSF practices, alternative splitting strategies to address shifts could be envisioned but would require additional effort and domain expertise to ensure appropriateness.
>
> ## Q3
> > Releasing a library is a convenient way of ensuring that pre-processing and evaluation are automatically standardized. Is there a particular reason why the authors decided not to provide one?
>
> We believe that releasing a new library would unnecessarily complicate adoption.
> Instead, upon acceptance, we plan to make our corrected datasets available on established platforms like Hugging Face, as mentioned in the paper.
> Additionally, integrating pre-processing code into existing frameworks like GluonTS is another viable option.
> Ultimately, we believe that directly using corrected datasets should streamline usability for the community.

---

### Official Review · Reviewer_D6qR · 2024-10-28

**Soundness:** 2
**Presentation:** 2
**Contribution:** 2
**Rating:** 5
**Confidence:** 4

**Summary:**

This paper proposes a solution to address inconsistencies and unreasonable data splits in time series forecasting evaluation, facilitating more accurate assessments.

**Strengths:**

As far as I know, data inconsistency is a common issue. The use of imputation in this paper is reasonable. The paper provides comprehensive experiments comparing the original datasets with the revised ones.

**Weaknesses:**

1. **Generality of the problem:** The paper discusses inconsistencies and data splitting issues using only three datasets. Are these issues commonly observed across other datasets?
2. **Real-world significance:** While I agree that data inconsistency is relevant in practice, I disagree with the periodic splitting approach. As far as I know, most real-world time series applications [1-2] do not align with periodic splitting settings.
3. **Lack of related work discussion:** Many studies have discussed time series forecasting (TSF) benchmarking. At least, [3] also aim s to fair evaluation. The authors should discuss the difference in motivation and in evaluation results.
4. **Limited novelty and dataset contribution:** Overall, I have concerns about the novelty of this work. Imputation is a common preprocessing technique, and the rationale for periodic splitting requires further discussion. From a dataset perspective, only three datasets are processed, and the benchmarking uses very few methods.

**References:**
[1] Dugas, Andrea Freyer, et al. "Influenza forecasting with Google Flu Trends." *PLoS One* 8.2 (2013): e56176.
[2] Singh, Ritika, and Shashi Srivastava. "Stock prediction using deep learning." *Multimedia Tools and Applications* 76 (2017): 18569-18584.
[3] Qiu, Xiangfei, et al. "Tfb: Towards comprehensive and fair benchmarking of time series forecasting methods." VLDB 2024

**Questions:**

Please check limitations.

---

> ### Author Response · Authors · 2024-11-25
> **Rebuttal to review by Reviewer D6qR**
>
> We would like to thank _Reviewer D6qR_ for the thorough review of our submission.
> Below, we provide detailed responses to the limitations raised.
> Although the discussion phase is nearing completion, we are eager to receive further feedback on our rebuttal.
>
> ## W1
> > Generality of the problem
>
> We have found errors/inconsistencies on several major Time Series datasets commonly used in the literature. However, as
> mentioned in the main rebuttal, due to the time required to provide extensive analysis of datasets, our aim is to raise
> awareness, foster discussion and future works on these issues through collaborations.
>
> Thank you for this observation.
> We have identified errors and inconsistencies in several major time series datasets commonly used in the literature.
> However, as stated in our rebuttal, due to the time-intensive nature of conducting extensive analyses across multiple datasets, our current focus is on raising awareness of these issues and fostering discussions and future collaborative efforts to address them more comprehensively.
>
> ## W2
> > Real-world significance
>
> We appreciate the _Reviewer D6qR_’s perspective.
> In our view, research efforts, particularly those focused on benchmarking, aim to evaluate models to determine their relative efficiency rather than replicating specific real-world scenarios.
>
> Real-world applications often require probabilistic forecasting models that provide a range of potential values for each time step rather than single-point predictions.
> For instance, in scenarios such as photovoltaic (PV) + battery + grid optimization, where PV generation depends heavily on weather, the exact predicted values are less critical than the actionable decisions informed by a range of potential predictions.
>
> As such, the metrics and evaluation approaches for real-world scenarios differ significantly from those used in benchmarking.
> Benchmarking contributes to advancing knowledge and understanding, which ultimately supports the development of more robust and relevant models for practical applications.
>
> ## W3
> > Lack of related work discussion
>
> We are aware of [3], which, as we understand, focuses on ranking existing models but does not address errors, inconsistencies, or splitting issues in datasets.
> Nevertheless, we acknowledge that we overlooked its inclusion in our related works section.
> Thank you for pointing this out; we have revised the section to incorporate and discuss the contributions and distinctions of [3].
>
>
> ## W4
> > Limited novelty and dataset contribution
>
> As mentioned in the main rebuttal, our work does not aim to present a novel solution for identifying and correcting dataset inconsistencies.
> Instead, our primary goal is to raise awareness of these underlying issues and provide a transparent methodology that allows users to:
>
> 1. Clearly identify the modified time steps (via newly added columns).
> 2. Compare the modified data with the raw values.
> 3. Reproduce our correction process.
>
> The diversity of domains within time series forecasting (e.g., traffic, weather, electricity, finance) makes it challenging--if not unrealistic--to propose a universal solution for identifying and addressing any dataset errors.
> As mentioned in the paper, leveraging large language models (LLMs) could be an ideal solution for such tasks.
> However, current limitations, including issues like hallucinations, hinder their effectiveness.
> LLMs may either miss inconsistencies or identify non-existent ones, and their results often lack robustness due to variability between runs.
>
> Our intention is to initiate a dialogue within the community, highlighting these challenges and encouraging collaborative efforts to perform extensive analyses and provide clean versions of widely used TS datasets.
> While our current work focuses on three datasets, we hope to expand this effort in the future with the support of the broader research community.

---

### Official Review · Reviewer_YxjH · 2024-11-01

**Soundness:** 3
**Presentation:** 2
**Contribution:** 3
**Rating:** 6
**Confidence:** 2

**Summary:**

This paper motivates the need for high quality datasets in benchmarks of Time Series Forecasting models, and through careful data analysis, provide cleaned versions of 3 datasets. By comparing existing models on the new datasets, they establish the importance of the dataset version on method relative performances.

**Strengths:**

1) Extensive data analysis is systematically presented in Appendix, analyzing seasonality in a multivariate context. This is precious for practitioners looking to analyze models on such datasets.
2) The task of providing clean datasets to the forecasting community is both significant and difficult, as currently each new paper benchmarks previous work in its own fashion. Standardizing training-validation-test split via data analysis is valuable in such a context.
3) Code, datasets, and figures are provided and well structured.

**Weaknesses:**

1) There is overall very few citations in introduction, section 2.1, section 2.3 and section 3.
The paper makes claims on the general approaches to data processing (eg line 147). Additionally, the paper tracks variants of the three datasets through different papers. Overall, the first four pages are about existing works, concerns and approaches, yet few citations and no paper collection methodology are presented.
2) The main paper version is very vague on the proposed data correction methods. The protocol for FFT application was described in Appendix, but this would be better included in the main paper. Imputation on the other hand appears simplistic. It would be for the best if the paper could try or discuss other methods and why specifically linear interpolation and neighbourhood average should be chosen. Surveys: https://arxiv.org/abs/2402.04059, https://arxiv.org/abs/2011.11347
3) Three different methods might not be enough to see strong changes on method ranking. Additionally, 3 runs is insufficient to provide confidence on results significance.

**Questions:**

Seeing the correlation analysis results, would the authors say that correlation analysis is useful to measure dataset quality? I could not come to a conclusion, but I did not spend much time comparing the original and corrected plots.

(Appendix, 1870, "we can observe" and not "we can observed", this one caught my eye). As most of the paper/appendix is text without formulas, I suggest going through a spell and grammar checker.

Is there any way to measure seasonal differences on the ECL dataset?

---

> ### Author Response · Authors · 2024-11-25
> **Rebuttal to review by Reviewer YxjH**
>
> We would like to thank _Reviewer YxjH_ for their thorough review of our submission.
> Below, we provide detailed responses to the questions and limitations raised.
> Although the discussion phase is nearing completion, we are eager to receive further feedback on our rebuttal.
>
> ## W1
> > There is overall very few citations in introduction, section 2.1, section 2.3 and section 3. The paper makes claims on the general approaches to data processing (eg line 147). Additionally, the paper tracks variants of the three datasets through different papers. Overall, the first four pages are about existing works, concerns and approaches, yet few citations and no paper collection methodology are presented.
>
> Thank you for highlighting this issue.
> We have added additional references in the specified sections to address this concern.
>
> Regarding Section 3, to the best of our knowledge, there is no prior research that addresses dataset splitting issues in TSF or identifies and corrects underlying errors and inconsistencies.
> If _Reviewer YxjH_ is aware of relevant references, we would be grateful to include and discuss them in our paper.
>
> ## W2
> > The main paper version is very vague on the proposed data correction methods. The protocol for FFT application was described in Appendix, but this would be better included in the main paper. Imputation on the other hand appears simplistic. It would be for the best if the paper could try or discuss other methods and why specifically linear interpolation and neighbourhood average should be chosen.
>
>
> The data correction and imputation are described in section 4.2.
> 1. **Isolated** time steps with inconsistencies.
>      - Both partial (affecting a subset of channels) and complete (affecting all channels) inconsistencies are corrected using linear interpolation.
> 2. **Consecutive** time steps with inconsistencies.
>      - **Case (a)**: For partial inconsistencies (not all channels have inconsistencies), context-aware imputation is applied:
>        - A “correct data” subset is created using consistent time steps.
>        - Values are imputed by averaging similar entries from the “correct data” based on shared temporal features (e.g., time, month).
>        - If no similar data exists, linear interpolation is used.
>      - **Case (b)**: For complete inconsistencies, linear interpolation is applied to avoid introducing out-of-context information.
>
> Does this explanation remain unclear? We are happy to revise it further in the paper.
>
> It is worth noting that while we added the references mentioned, the objective of this work is not to propose novel detection or imputation techniques but rather to illustrate the impact of correcting errors on existing datasets.
>
> ## W3
> > Three different methods might not be enough to see strong changes on method ranking.
>
> As stated in the paper, our goal is not to provide an exhaustive benchmark but to demonstrate the impact of correcting errors.
> For example, Table 16 in the Supplementary Material (and also provided in a separated comment above) shows a significant performance gap in standard metrics between the uncorrected MPI dataset and its corrected version.
>
> > Additionally, 3 runs is insufficient to provide confidence on results significance.
>
> While many studies in the literature rely on 1 to 3 runs, we acknowledge this concern.
> Could _Reviewer YxjH_ suggest the number of runs required to ensure statistical significance?
> If there are established guidelines or references, we would be glad to follow and cite them.
>
> ## Q1
> > Would the authors say that correlation analysis is useful to measure dataset quality?
>
> We believe correlation analysis is not a reliable metric for dataset quality.
> For instance, datasets with large blocks of missing data (filled with a fixed value) could exhibit artificially high correlation between channels, potentially even higher than datasets without missing blocks.
>
> However, correlation analysis is useful for understanding inter-channel relationships, which can inform model design and behavior.
> It is an important tool for identifying properties that models should leverage.
>
> ## Q2
> > As most of the paper/appendix is text without formulas, I suggest going through a spell and grammar checker.
>
> Thank you for pointing this out.
> We have carefully proofread the paper and corrected all spelling and grammatical errors.
>
> ## Q3
> > Is there any way to measure seasonal differences on the ECL dataset?
>
> Could _Reviewer YxjH_ clarify this question? If the goal is to measure seasonal differences between ECL channels, the large number of channels ($308$) made it impractical to include a full frequency analysis table (the same applies to correlation analysis).
>
> We can, however, provide a histogram showing the most represented frequencies in the ECL dataset if this would address the question.

---

> > ### Comment · Reviewer_YxjH · 2024-11-27
> >
> > Thank you for the answer.
> >
> > I would appreciate a mathematical formulation of the interpolation in case 2b, naming the quantities used to do the interpolation.
> >
> > In my opinion, the problem with non-standardized predictive performance is not that reported performances are different between papers, but that the method ranking / behavior could be dependent on data processing choices differing from papers to papers. AutoFormer performances on MPI highlight this issue, but the analysis in the paper focuses most on the absolute change of predictive performance.
> >
> > Regarding number of runs, statistical significance is ideally computed with proper tests, such as Independent Samples T Test for independent populations, or Wilcoxon signed rank tests or the autorank package in python for paired data. Strong tendencies are generally distinguishable at 10 runs.

---

> > > ### Author Response · Authors · 2024-11-29
> > >
> > > Thank you for your feedback.
> > >
> > > > I would appreciate a mathematical formulation of the interpolation in case 2b,
> > >
> > > For a single channel having $n$ consecutive errors or inconsistencies from $t+k+1$ to $t+k+n$, where the values at time steps $t+k$ and $t+k+n+1$ are consistent, we applied linear interpolation as follows:
> > > For $i \in {t+k+1,\cdots,t+k+n}$, the interpolated value $y_i$ is computed as:
> > > $y_i = y_{t+k} + (x_i−x_{t+k})*(y_{t+k+n+1}−y_{t+k}) / (x_{t+k+n+1}−x_{t+k})$
> > >
> > > While not perfect, this approach is preferable to filling missing values with a fixed value.
> > > Specifically used in case 2b, where contextual information from other channels is unavailable, it avoids filling all channels with the same value, which would be inconsistent with the normal data patterns.
> > >
> > >
> > > > Regarding number of runs, statistical significance is ideally computed with proper tests [...]. Strong tendencies are generally distinguishable at 10 runs.
> > >
> > > In our understanding, the reviewer raises two distinct points: (i) the statistical significance of the results and (ii) the number of runs required to infer such significance. We interpret the latter as being more related to the robustness of the results and the variability in model predictions. Please correct us if this understanding is incorrect. While $10$ runs may help reveal strong trends, they do not inherently guarantee statistical significance without proper testing. To address this, we are planning to perform a statistical significance analysis of our 3-run results using a _paired t-test_ and will include these findings in the next revision. Unfortunately, the deadline for uploading revisions to OpenReview has already passed. Nevertheless, once the analysis is complete, we will provide a summary here.
> > >
> > > Thank you once again for your valuable feedback. We sincerely hope that our revisions and clarifications address your concerns, and we remain open to further discussion to ensure that our contributions are both clear and impactful.

---

> ### Author Response · Authors · 2024-12-03
> **Paired t-test results**
>
> In these tests, we are considering whether the results for a given experiment (prediction horizon and model) obtained with the original datasets were drawn from the same population as the results obtained with the cleaned version (otherwise they are from two different populations).
> We set the threshold at $5\\%$ as it is often done in the literature.
>
> The t-test quantifies the difference between the arithmetic means of the two results. The p-value quantifies the probability of observing as or more extreme values assuming the null hypothesis, that the samples are drawn from populations with the same population means, is true. A p-value larger than the chosen threshold (depicted in bold in Tables S1 to S8) indicates that our results is not so unlikely to have occurred by chance. Therefore, we do not reject the null hypothesis of equal population means. If the p-value is smaller than our threshold, then we have evidence **against** the null hypothesis of equal population means.
>
> ## UCI ELD
>
> Table S1 - pvalue for UCI ELD in Multivariate-to-Multivariate scenario with ratio splitting
> |              |     | ECL vs. PELD_1H_3Y_308 | ECL vs. PELD_1H_3Y_308 |
> |-------------:|----:|-----------------------:|-----------------------:|
> |              |     |                    MSE |                    MAE |
> |              |  96 |            3.87692e-10 |              0.0262972 |
> |      NLinear | 192 |            4.47191e-20 |            2.08074e-11 |
> |              | 336 |            6.89816e-08 |             0.00964533 |
> |              | 720 |            1.29662e-06 |            0.000285956 |
> |              |     |                        |                        |
> |              |  96 |             0.00025635 |            0.000276501 |
> |     Informer | 192 |            0.000266339 |             0.00129099 |
> |              | 336 |               0.022167 |              0.0267096 |
> |              | 720 |              0.0124594 |              0.0338038 |
> |              |     |                        |                        |
> |              |  96 |            0.000428919 |              0.0108611 |
> | iTransformer | 192 |            3.68511e-05 |             0.00134274 |
> |              | 336 |            3.77962e-05 |              0.0657208 |
> |              | 720 |            0.000240438 |               0.070916 |
>
> In this scenario, we can reject the null hypothesis and therefore conclude that the results with the clean dataset are significant.
>
>
> Table S2 - pvalue for UCI ELD in Multivariate-to-Multivariate scenario with cycle-inclusive splitting
> |              |     | ECL vs. PELD_1H_3Y_308 | ECL vs. PELD_1H_3Y_308 |
> |-------------:|----:|-----------------------:|-----------------------:|
> |              |     |                    MSE |                    MAE |
> |              |  96 |            2.18485e-06 |            0.000311399 |
> |      NLinear | 192 |            1.98664e-06 |            0.000397757 |
> |              | 336 |            5.96994e-08 |            7.61006e-07 |
> |              | 720 |              5.221e-05 |            0.000886912 |
> |              |     |                        |                        |
> |              |  96 |             0.00244866 |               **0.100637** |
> |     Informer | 192 |              **0.0687518** |               **0.850804** |
> |              | 336 |              0.0173096 |               **0.100636** |
> |              | 720 |              0.0171914 |              **0.0633262** |
> |              |     |                        |                        |
> |              |  96 |            9.18341e-06 |            0.000126319 |
> | iTransformer | 192 |            1.50853e-06 |            1.84953e-05 |
> |              | 336 |              9.836e-08 |            0.000139968 |
> |              | 720 |            7.20129e-08 |            0.000474033 |
>
> In this scenario, we can reject the null hypothesis and therefore conclude that the results with the clean dataset are significant, only for iTransformer and NLinear. Informer results are not conclusive, and more runs should be performed.

---

> > ### Author Response · Authors · 2024-12-03
> > **Paired t-test results (MPI)**
> >
> > ## MPI
> >
> > Table S3 - pvalue for MPI in Multivariate-to-Multivariate scenario with ratio splitting
> > |              |     | Original vs. Simple | Original vs. Simple | Original vs. MPIW_10T_1Y_R | Original vs. MPIW_10T_1Y_R |
> > |-------------:|----:|--------------------:|--------------------:|---------------------------:|---------------------------:|
> > |              |     |                 MSE |                 MAE |                        MSE |                        MAE |
> > |              |  96 |         2.57e-08 |         1.19e-05 |                2.6e-08 |                6.1e-06 |
> > |      DLinear | 192 |         8.36e-05 |         0.000795 |                3.59e-05 |                0.000773 |
> > |              | 336 |         2.32e-06 |         3.4e-05 |                9.1e-06 |                2.8e-05 |
> > |              | 720 |         6.45e-06 |         5.5e-05 |                6.29e-06 |                8.9e-05 |
> > |              |     |                     |                     |                            |                            |
> > |              |  96 |                   / |                   / |                 0.0028 |                 0.0097 |
> > |   Autoformer | 192 |                   / |                   / |                  0.019 |                  0.037 |
> > |              | 336 |                   / |                   / |                  0.026 |                  **0.070** |
> > |              | 720 |                   / |                   / |                   **0.11** |                  **0.096** |
> > |              |     |                     |                     |                            |                            |
> > |              |     |                 MSE |                 MAE |                        MSE |                        MAE |
> > |              |  96 |         9.03e-06 |         8.66e-06 |                1.23e-05 |                1.40e-05 |
> > | iTransformer | 192 |         8.28e-07 |         1.45e-07 |                1.00e-06 |                2.29e-07 |
> > |              | 336 |         1.03e-06 |         1.04e-06 |                6.60e-07 |                3.84e-07 |
> > |              | 720 |         6.78e-09 |         4.71e-08 |                1.25e-06 |                2.89e-07 |
> >
> > In this scenario, we can reject the null hypothesis and therefore conclude that the results with the clean dataset are significant, only for iTransformer and DLinear. Autoformer results are not conclusive, and more runs should be performed.
> >
> > Table S4 - pvalue for MPI in Univariate-to-Univariate scenario with ratio splitting
> > |              |     | Original vs. Simple |      Original vs. Simple | Original vs. MPIW_10T_1Y_R | Original vs. MPIW_10T_1Y_R |
> > |-------------:|----:|--------------------:|-------------------------:|---------------------------:|---------------------------:|
> > |              |     |                 MSE |                      MAE |                        MSE |                        MAE |
> > |              |  96 |         3.07e-06 |              2.51e-07 |                1.00821e-07 |                8.32997e-09 |
> > |      DLinear | 192 |         3.54e-09 |              1.15e-09 |                4.64674e-10 |                6.40108e-11 |
> > |              | 336 |         4.95e-09 |              3.88e-09 |                1.28886e-12 |                2.54358e-10 |
> > |              | 720 |         4.56e-10 |              9.39e-10 |                4.75962e-07 |                8.19331e-08 |
> > |              |     |                     |                          |                            |                            |
> > |              |  96 |                   / |                        / |                1.10716e-05 |                2.08054e-06 |
> > |   Autoformer | 192 |                   / |                        / |                6.38358e-06 |                1.31835e-06 |
> > |              | 336 |                   / |                        / |                7.65624e-05 |                 9.3937e-06 |
> > |              | 720 |                   / |                        / |                1.44131e-05 |                1.62557e-06 |
> > |              |     |                     |                          |                            |                            |
> > |              |     |                 MSE |                      MAE |                        MSE |                        MAE |
> > |              |  96 |         4.68165e-09 |              8.06821e-10 |                4.60671e-07 |                3.11748e-09 |
> > | iTransformer | 192 |         3.33496e-11 |              5.95629e-12 |                 4.7499e-09 |                9.31546e-10 |
> > |              | 336 |         4.81668e-09 |              1.87197e-09 |                5.40087e-07 |                1.51031e-08 |
> > |              | 720 |         1.11264e-08 |               1.2731e-12 |                6.55467e-08 |                5.81892e-09 |
> >
> > In this scenario, we can reject the null hypothesis and therefore conclude that the results with the clean dataset are significant.

---

> > > ### Author Response · Authors · 2024-12-03
> > > **Paired t-test results (LCD 1/4)**
> > >
> > > ## LCD Univariate-to-Univariate scenario
> > > Table S5 - pvalue for LCD with cycle-inclusive splitting
> > > |              |     | Original vs. LCDWf_1H_4Y_USUN | Original vs. LCDWf_1H_4Y_USUN | Original vs. LCDWi_1H_4Y_USUN | Original vs. LCDWi_1H_4Y_USUN |
> > > |-------------:|----:|------------------------------:|------------------------------:|------------------------------:|------------------------------:|
> > > |              |     |                           MSE |                           MAE |                           MSE |                           MAE |
> > > |              |  96 |                   2.97e-07 |                   7.57e-07 |                    1.34e-06 |                   8.19e-06 |
> > > |      NLinear | 192 |                    7.53e-09 |                   2.60e-08 |                   1.65e-08 |                    9.52e-08 |
> > > |              | 336 |                   6.22e-13 |                   2.15e-10 |                   3.35e-12 |                   1.26e-09 |
> > > |              | 720 |                   5.40e-13 |                   2.67e-12 |                   1.20e-10 |                   4.93e-10 |
> > > |              |     |                               |                               |                               |                               |
> > > |              |  96 |                    0.00159 |                    0.00357 |                   0.000143 |                     0.0015 |
> > > |     Informer | 192 |                    0.005 |                    0.00260 |                     **0.0799** |                     **0.0953** |
> > > |              | 336 |                     **0.0947** |                      **0.173** |                     **0.0562** |                      **0.166** |
> > > |              | 720 |                      **0.190** |                      **0.318** |                      **0.326** |                      **0.493** |
> > > |              |     |                               |                               |                               |                               |
> > > |              |  96 |                    0.00219 |                    0.00512 |                    0.00294 |                    0.00915 |
> > > | iTransformer | 192 |                    0.00200 |                    0.00498 |                    0.00329 |                    0.00876 |
> > > |              | 336 |                    0.00305 |                       **0.357** |                   0.000831 |                       **0.103** |
> > > |              | 720 |                    0.00642 |                     0.0163 |                     0.0169 |                      0.046 |
> > >
> > > In this scenario, we can reject the null hypothesis and therefore conclude that the results with the clean dataset are significant, only for NLinear and almost all iTransformer results. Informer results are not conclusive, and more runs should be performed.

---

> > > > ### Author Response · Authors · 2024-12-03
> > > > **Paired t-test results (LCD 2/4)**
> > > >
> > > > ## LCD Univariate-to-Univariate scenario
> > > >
> > > > Table S6 - pvalue for LCD with ratio splitting
> > > > |              |     | Original vs. LCDWf_1H_4Y_USUN | Original vs. LCDWf_1H_4Y_USUN | Original vs. LCDWi_1H_4Y_USUN | Original vs. LCDWi_1H_4Y_USUN |
> > > > |-------------:|----:|------------------------------:|------------------------------:|------------------------------:|------------------------------:|
> > > > |              |     |                           MSE |                           MAE |                           MSE |                           MAE |
> > > > |              |  96 |                   8.79e-07 |                   5.21e-06 |                   5e-06 |                   5.09e-05 |
> > > > |      NLinear | 192 |                   3.39e-08 |                   7.40e-08 |                   2.18e-07 |                   9.78e-07 |
> > > > |              | 336 |                   2.53e-07 |                   7.51e-07 |                   6.90e-07 |                   4.50e-06 |
> > > > |              | 720 |                   5.20e-08 |                   1.17e-07 |                   7.48e-08 |                   3.07e-07 |
> > > > |              |     |                               |                               |                               |                               |
> > > > |              |  96 |                      **0.107** |                      **0.176** |                      **0.379** |                      **0.536** |
> > > > |     Informer | 192 |                     **0.0949** |                      **0.145** |                      **0.165** |                      **0.214** |
> > > > |              | 336 |                      **0.629** |                      **0.623** |                      **0.683** |                      **0.668** |
> > > > |              | 720 |                      **0.249** |                      **0.310** |                      **0.553** |                       **0.6** |
> > > > |              |     |                               |                               |                               |                               |
> > > > |              |  96 |                    0.00068 |                   0.000655 |                   0.000786 |                     0.0018 |
> > > > | iTransformer | 192 |                   0.000287 |                   0.000863 |                    0.00292 |                     0.0104 |
> > > > |              | 336 |                    0.00596 |                    0.00925 |                   0.000132 |                   0.000728 |
> > > > |              | 720 |                    0.00140 |                     0.0025 |                    0.00199 |                    0.00196 |
> > > >
> > > > In this scenario, we can reject the null hypothesis and therefore conclude that the results with the clean dataset are significant, only for iTransformer and NLinear. Informer results are not conclusive, and more runs should be performed.

---

> > > > > ### Author Response · Authors · 2024-12-03
> > > > > **Paired t-test results (LCD 3/4)**
> > > > >
> > > > > ## LCD Multivariate-to-Multivariate scenario
> > > > > Table S7 - pvalue for LCD with cycle-inclusive splitting
> > > > > |              |     | Original vs. LCDWf_1H_4Y_USUN | Original vs. LCDWf_1H_4Y_USUN | Original vs. LCDWi_1H_4Y_USUN | Original vs. LCDWi_1H_4Y_USUN |
> > > > > |-------------:|----:|------------------------------:|------------------------------:|------------------------------:|------------------------------:|
> > > > > |              |     |                           MSE |                           MAE |                           MSE |                           MAE |
> > > > > |              |  96 |                   5.46817e-10 |                    6.0136e-10 |                   9.85883e-09 |                   1.21153e-08 |
> > > > > |      NLinear | 192 |                   3.46465e-08 |                   3.71661e-08 |                   3.24509e-07 |                   3.85172e-07 |
> > > > > |              | 336 |                   5.93273e-09 |                   6.46812e-09 |                   6.18841e-08 |                   7.31146e-08 |
> > > > > |              | 720 |                   9.23155e-11 |                   1.03567e-10 |                   1.71372e-09 |                     1.824e-09 |
> > > > > |              |     |                               |                               |                               |                               |
> > > > > |              |  96 |                      **0.276562** |                      **0.903942** |                       **0.58757** |                      **0.635984** |
> > > > > |     Informer | 192 |                     **0.614086** |                     0.0157754 |                      **0.810838** |                      **0.704586** |
> > > > > |              | 336 |                   0.000468817 |                   0.000665168 |                       **0.11099** |                      **0.123031** |
> > > > > |              | 720 |                       **0.10507** |                     **0.0946786** |                      **0.337283** |                      **0.309824** |
> > > > > |              |     |                               |                               |                               |                               |
> > > > > |              |  96 |                   6.92896e-07 |                   6.52834e-07 |                    0.00737257 |                    0.00872735 |
> > > > > | iTransformer | 192 |                   0.000354362 |                    0.00034317 |                     **0.0570613** |                      **0.061995** |
> > > > > |              | 336 |                   9.71845e-05 |                   0.000157824 |                    0.00250668 |                     0.0029688 |
> > > > > |              | 720 |                    0.00230772 |                    0.00216448 |                     0.0166755 |                     0.0178841 |
> > > > >
> > > > > In this scenario, we can reject the null hypothesis and therefore conclude that the results with the clean dataset are significant, only for NLinear and almost all iTransformer results (some results with the integer version are not significant). Informer results are not conclusive, and more runs should be performed.

---

> ### Author Response · Authors · 2024-12-03
> **Paired t-test results (LCD 4/4)**
>
> ## LCD Multivariate-to-Multivariate scenario
> Table S8 - pvalue for LCD with ratio splitting
> |              |     | Original vs. LCDWf_1H_4Y_USUN | Original vs. LCDWf_1H_4Y_USUN | Original vs. LCDWi_1H_4Y_USUN | Original vs. LCDWi_1H_4Y_USUN |
> |-------------:|----:|------------------------------:|------------------------------:|------------------------------:|------------------------------:|
> |              |     |                           MSE |                           MAE |                           MSE |                           MAE |
> |              |  96 |                   0.000258533 |                   0.000273424 |                   0.000936839 |                    0.00112018 |
> |      NLinear | 192 |                   5.63472e-05 |                   6.00389e-05 |                   9.60816e-05 |                   0.000115361 |
> |              | 336 |                   6.01829e-06 |                   6.34247e-06 |                   4.07154e-08 |                    5.0511e-08 |
> |              | 720 |                   9.00731e-08 |                   9.34643e-08 |                   2.45338e-08 |                   2.85148e-08 |
> |              |     |                               |                               |                               |                               |
> |              |  96 |                     0.0329336 |                    0.00312674 |                      **0.129197** |                       **0.11172** |
> |     Informer | 192 |                     **0.0619376** |                     **0.0578092** |                      **0.423858** |                      **0.171492** |
> |              | 336 |                      **0.118386** |                      **0.134625** |                      **0.529327** |                      **0.592045** |
> |              | 720 |                     **0.109531** |                      **0.114738** |                      **0.220532** |                      **0.234282** |
> |              |     |                               |                               |                               |                               |
> |              |  96 |                     0.0154452 |                     0.0152869 |                     0.0316815 |                     0.0352083 |
> | iTransformer | 192 |                    0.00179572 |                    0.00192744 |                   0.000395793 |                   0.000450916 |
> |              | 336 |                    0.00266389 |                    0.00262952 |                      **0.097294** |                      **0.102792** |
> |              | 720 |                    0.00635411 |                    0.00744612 |                     **0.0556079** |                     **0.0645383** |
>
> In this scenario, we can reject the null hypothesis and therefore conclude that the results with the clean dataset are significant, only for NLinear and iTransformer results (except for the results with the integer version). Informer results are not conclusive, and more runs should be performed.
>
> These results suggest that the integer version of the dataset might not be suitable as-is for benchmarking models.

---

> > ### Author Response · Authors · 2024-12-03
> > **Paired t-test results (conclusion)**
> >
> > We have included the paired t-test results in the Supplementary Material and updated the text to reflect the associated observations.
> >
> > Thank you again for your thoughtful feedback. We hope these results further emphasize the importance of exploring these topics in greater depth within the community.

---

### Official Review · Reviewer_VTuf · 2024-11-04

**Soundness:** 2
**Presentation:** 1
**Contribution:** 1
**Rating:** 3
**Confidence:** 5

**Summary:**

The paper proposes three new benchmark datasets for time series forecasting. The proposed datasets aim to address data inconsistencies, missing values and improper temporal splits, facilitating fair evaluations for methods designed for forecasting task.

**Strengths:**

1)  Creating comprehensive and accurate benchmark dataset, especially for time series analysis, which is of high real-world importance is a critical and important task.

2) The paper conducts extensive experiments

**Weaknesses:**

1) The contribution of the paper is incremental, as it applies existing methods to three established datasets to create new benchmarks. The limitations of current benchmark datasets are well-known, and various solutions have already been developed to address these issues.

2) The paper lacks a systematic method for creating benchmark datasets, which limits its applicability. This raises the question: can the approach presented in the paper be applied to any existing dataset to establish it as a fair benchmark?

3) As noted in the paper, the approach relies heavily on domain expert knowledge, which limits its broader applicability. Additionally, if domain knowledge is to play a significant role, it could also be used to design new benchmark datasets.

4) Given the limited novelty and the real-world application of the paper, I believe the paper may be a better fit for a benchmark track.

**Questions:**

1) Can the approach presented in the paper be applied to any existing dataset to establish it as a fair benchmark?

---

> ### Author Response · Authors · 2024-11-25
> **Rebuttal to review by Reviewer VTuf**
>
> We would like to thank _Reviewer VTuf_ for their review and address the identified limitations below.
> Although the discussion phase is nearing its conclusion, we look forward to further feedback on our rebuttal.
>
> ## W1
> > The limitations of current benchmark datasets are well-known, and various solutions have already been developed to address these issues.
>
> To the best of our knowledge, no existing papers explicitly tackle the identification of errors and inconsistencies in commonly used Time Series Datasets for benchmarking or propose cycle-inclusive splitting.
> If _Reviewer VTuf_ has references supporting this claim, we would greatly appreciate them.
>
> Additionally, many widely recognized transformer variants (e.g., Informer, Autoformer, FEDformer, Crossformer, Pathformer, iTransformer, PatchTST) rely on standard dataloaders that do not include pre-processing steps to address such inconsistencies and errors.
> This highlights a significant gap in the field that our work seeks to address.
>
> ## W2 & Q1
> > This raises the question: can the approach presented in the paper be applied to any existing dataset to establish it as a fair benchmark?
>
> > Can the approach presented in the paper be applied to any existing dataset to establish it as a fair benchmark?
>
> As stated in the main rebuttal, a one-size-fits-all solution for identifying errors and inconsistencies across all domains of TSF datasets is highly unlikely.
> Each domain has unique characteristics—such as inter-channel relationships and value ranges—that require domain-specific knowledge.
> For example: Replacing missing data with 0 (a common practice) can lead to contradictions, such as relative humidity on Earth being zero or temperatures in Fahrenheit and Celsius both being zero at the same time.
>
> While our paper does not propose a novel method for universally addressing inconsistencies, we see this as a future research direction.
>
> On the other hand, the correction method we use is applicable to any dataset, though it is intentionally basic.
> We do not claim it to be novel but rather a starting point for discussion and further refinement.
> Our primary goal is to raise awareness, provide corrected dataset versions, and foster dialogue around these critical issues, as recognized by _Reviewer VTuf_.
>
> ## W3
> > Additionally, if domain knowledge is to play a significant role, it could also be used to design new benchmark datasets.
>
> Designing new benchmark datasets is a challenging and resource-intensive endeavor. It requires:
> 1. Clearly defining the dataset's purpose.
> 2. Setting up appropriate devices for raw data collection.
> 3. Allowing sufficient time for data validation and relevance.
>
> Typically, several years are needed for a benchmark dataset to achieve widespread recognition and relevance.
> This underscores the importance of preserving and improving existing TS datasets.
> By conducting extensive analyses and providing cleaned versions, we aim to enhance the usability and reliability of these valuable resources.
>
>
> ## W4
> > I believe the paper may be a better fit for a benchmark track.
>
> This paper was submitted under the primary area of "Datasets and Benchmarks" for ICLR 2025, aligning with its focus and contributions.

---

### Author Response · Authors · 2024-11-25
**Main rebuttal**

We sincerely thank the reviewers for their comments and for recognizing the significance of our work.
Below, we summarize the key points highlighted in the reviews:

1. Creation of a comprehensive benchmark dataset: Our work addresses a critical real-world challenge, particularly in time series analysis.
2. Extensive experiments and multivariate data analysis: The systematic exploration of seasonality provides valuable insights for both practitioners and researchers.
3. Standardizing data splits and cleaning datasets: This effort is vital for fostering reliable, reproducible, and consistent evaluation in the time-series community.
4. Well-structured resources: The provided code, datasets, and visualizations are highly accessible and support reproducibility.

We also appreciate the reviewers’ insightful questions, which we will address separately, but we summarize some key points regarding our submission below.

The primary goal of this paper is to raise awareness about underlying issues in commonly used time-series forecasting (TSF) datasets. These issues, often overlooked, can significantly impact the validity of published results. Our contributions include:

1. Corrected versions of three widely used datasets.
2. Cycle-inclusive splitting methodologies.
3. Extensive analyses to promote fairer comparisons and deeper understanding in TSF.

Our aim is not to propose a universal solution for identifying and correcting dataset issues. Instead, we provide transparency through our methodology, enabling future users to trace modifications, compare them with raw data, and replicate our processes.
Addressing Concerns

 * **Limited Number of Dataset**: While some reviewers suggested incorporating more datasets, our choice to focus on three datasets was intentional. Adding more datasets would not necessarily enhance the demonstration of the core problems but would complicate the analysis and make the paper overly dense for conference formats. Extensive benchmarks are better suited for future work.
 * **Diverse Dataset Challenges**: TSF datasets span various domains (e.g., traffic, weather, electricity, finance), making a universal error-detection approach un-realistic. Although large language models (LLMs) offer potential, current limitations such as hallucinations and inconsistent outputs render them currently unreliable for this task.
 * **Collaboration for Broader Impact**: Cleaning and analyzing datasets with high-dimensional channels (e.g., >$30$ variables) requires substantial effort. We hope this paper sparks discussion and collaboration within the community to address these challenges comprehensively. Joint contributions would ensure diverse perspectives and more robust outcomes.

As a conclusion, we envision this work more as a starting point to raise awareness, foster dialogue, and inspire collective efforts to improve dataset quality in TSF research. By identifying current gaps and challenges, we aim to galvanize the community toward cleaner, more reliable benchmarks for future advancements.

---

### Author Response · Authors · 2024-11-25
**Results with MPI dataset and univariate-to-univariate scenario**

|         | F   | MSE w/ Original         | MAE w/ Original         | MSE w/ Simple          | MAE w/ Simple          | MSE w/ Corrected           | MAE w/ Corrected          |
|---------|-----|--------------|--------------|--------------|--------------|--------------|--------------|
| DLinear | 96  | 0.005±0.0003 | 0.056±0.0027 | 0.387±0.0145 | 0.429±0.0071 | 0.555±0.0089 | 0.514±0.0029 |   |
|         | 192 | 0.006±0.0001 | 0.064±0.0006 | 0.476±0.0033 | 0.484±0.0021 | 0.651±0.0027 | 0.567±0.0012 |   |
|         | 336 | 0.006±0.0002 | 0.064±0.0019 | 0.527±0.0039 | 0.510±0.0025 | 0.743±0.0007 | 0.604±0.0002 |   |
|         | 720 | 0.006±0.0002 | 0.066±0.0021 | 0.595±0.0024 | 0.548±0.0012 | 0.947±0.0223 | 0.690±0.0093 |   |
| Auto.   | 96  | 0.003±0.0002 | 0.041±0.0017 | NA           | NA           | 0.767±0.0347 | 0.674±0.0182 |   |
|         | 192 | 0.004±0.0009 | 0.047±0.0063 | NA           | NA           | 0.767±0.0347 | 0.674±0.0182 |   |
|         | 336 | 0.004±0.0002 | 0.050±0.0016 | NA           | NA           | 0.940±0.0796 | 0.756±0.0353 |   |
|         | 720 | 0.004±0.0005 | 0.052±0.0030 | NA           | NA           | 1.205±0.0670 | 0.861±0.0259 |   |
| iTrans. | 96  | 0.001±0.0000 | 0.027±0.0002 | 0.266±0.0020 | 0.360±0.0016 | 0.440±0.0103 | 0.456±0.0029 |   |
|         | 192 | 0.002±0.0000 | 0.029±0.0002 | 0.339±0.0007 | 0.414±0.0005 | 0.571±0.0043 | 0.532±0.0025 |   |
|         | 336 | 0.002±0.0000 | 0.031±0.0002 | 0.377±0.0028 | 0.444±0.0024 | 0.641±0.0157 | 0.573±0.0054 |   |
|         | 720 | 0.002±0.0000 | 0.035±0.0001 | 0.499±0.0046 | 0.516±0.0005 | 0.857±0.0124 | 0.671±0.0050 |   |

Results with MPI (autoformer weather dataset) for univariate-to-univariate predictions and a ratio splitting (7:1:2). Our experiments are run three times, both the average error and standard deviation are reported in this table. Original are results obtained when using dataset as define in Autoformer paper; Simple are results obtained when replacing failure values by 0; Corrected are results obtained with our corrected version.

---

### Meta-Review · Area_Chair_Wr5A · 2024-12-23

**Metareview:**

The paper highlights current challenges with inconsistency in time-series evaluations due to a variety of issues like missingness, inconsistencies, and practices such as temporal splits. The authors remedy these issues by creating a benchmark with three datasets and comparison of recent methods for these tasks.

Reviewers have acknowledged the importance of the problem, some comprehensiveness of the evaluation, availability of code, data processing pipelines and broad intent of the contribution.

Several reviewers have cited that the empirical evaluation is somewhat weak due to lack of sufficient comparison of models, insufficient literature review for the benchmarking paper, including missing prior literature review highlighting many existing contributions, and insufficient insights given that model ranking does not change before applying the methods proposed in this benchmark versus after. Therefore it is unclear what problems the contribution is trying to surface and whether it is able to do so successfully.

Authors have addressed several of these concerns, including additional empirical evaluations as part of the rebuttal. However, overall reviewers have finally mentioned that the contribution still is not at par to justify an acceptance. Therefore, I recommend a reject.

**Additional Comments On Reviewer Discussion:**

No additional concerns were raised by reviewers during discussion.

---

### Decision · Program_Chairs · 2025-01-22

Reject